Technical Report

# Urinary detection of therapy-induced senescence and fibrosis using an injectable albumin-based nanoprobe

Muhamad Hartono[1,2], Jianfeng Ge[2], Mary Denholm [2,3], Matthew G. Ellis [4], Joaquín Araos Henríquez [2], Andrew G. Baker [1], Robert C. Rintoul [2,5], Tijmen Euser[4], Ljiljana Fruk [1] ✉ & Daniel Muñoz-Espín [2] ✉

Cellular senescence is a hallmark of age-related disorders, including cancer, in which senescence contributes to tumor progression and treatment resistance. Targeting senescent cells therapeutically requires noninvasive methods to longitudinally monitor senescence burden. Here, we present an injectable nanoprobe for noninvasive detection of therapy-induced senescence in lung cancer and pulmonary fibrosis via urine testing. Using human biopsy samples, clinical transcriptomic datasets and mouse models, we identify matrix metalloproteinase-7 (MMP-7) as a specific biomarker of senescence in lung cancer and bleomycin-induced fibrosis. We develop ALBANC, a nanoprobe composed of human serum albumin linked to gold nanoclusters (AuNCs) through MMP-7-cleavable peptide linkers. MMP-7-mediated cleavage releases AuNCs that are renally excreted, enabling rapid and sensitive colorimetric urine detection via a nanoparticle growth-based assay, enabling longitudinal tracking of cisplatin-induced senescence and senolysis in mouse lung tumors and fibrosis. This approach offers a noninvasive and sensitive precision tool for monitoring senescence burden in lung cancer.

Sensitive diagnostic tools are crucial for early disease detection, enabling more effective and less invasive treatment and improving long-term survival[1]. Detecting cellular senescence in patients remains challenging, despite its implication in multiple age-related diseases, including cancer[2]. Cells undergo senescence in response to irreparable stress or damage, whereby they stably halt their cell cycle. Senescent cells are characterized by different features, including increased levels of cyclin-dependent kinase inhibitors (for example, p16 and p21), reduced hyperphosphorylated retinoblastoma (pRb) expression and elevated lysosomal β-galactosidase activity, also known as senescence-associated β-galactosidase (SA-β-gal), which is commonly used in a colorimetric-based senescence detection assay. Senescence limits proliferation of damaged cells while promoting tissue repair via the senescence-associated secretory phenotype (SASP). SASP factors includes cytokines, chemokines and matrix remodeling proteases[3–5]. In physiological conditions, SASP factors facilitate the repair and regeneration of damaged tissue[2]. However, with persistent damage or aging, senescent cells can accumulate, partially due to inefficient immune clearance. This accumulation can cause chronic inflammation, tissue dysfunction and fibrosis, contributing to diseases such as cancer[2,6].

In the context of cancer, senescence plays a dual role. It can serve as a cell-autonomous tumor-suppressive mechanism against

[1]Department of Chemical Engineering and Biotechnology, University of Cambridge, Cambridge, UK. [2]Early Cancer Institute, Department of Oncology, University of Cambridge, Cambridge, UK. [3]Department of Oncology, Addenbrooke's Hospital, Cambridge, UK. [4]Nanophotonics Centre, Department of Physics, Cavendish Laboratory, University of Cambridge, Cambridge, UK. [5]Royal Papworth Hospital NHS Foundation Trust, Cambridge Biomedical Campus, Cambridge, UK. ✉e-mail: lf389@cam.ac.uk; dm742@cam.ac.uk

oncogene-driven tumorigenesis[7–11]. However, chemotherapy and radiotherapy can often lead to therapy-induced senescence (TIS) in cancer and stromal cells, which can promote metastasis[12–15], stemness[16], and a tumor-promoting microenvironment[17–19]. TIS has also been suggested to suppress anti-tumor immunity after treatment[20]. Accordingly, senolytic therapies that selectively eliminate senescent cells have been extensively explored[21–24]. Multiple senolytics have been validated in pre-clinical models and can synergize with existing senescence-inducing cancer chemotherapies; early-phase clinical trials are also reporting promising results[23,25,26]. This 'one-two punch' combination strategy combining pro-senescence therapy (for example, chemotherapy) with senolytics has shown promise. However, its clinical translation is hampered by a lack of tools that can sensitively and accurately detect senescence in vivo[27].

Recent advancements in artificial urinary nanoprobes show that enzyme- or protease-responsive reporters in urine can enable noninvasive monitoring of disease progression and therapeutic response[28–40] by generating so-called synthetic urinary biomarkers[28,29,41]. Although these nanoprobes have advanced early cancer detection, they have not been leveraged to detect cellular senescence or fibrosis. Current senescence detection in patients rely on histology from biopsies or surgery, which may not represent whole tissues and is unsuitable for longitudinal monitoring[7,27,42,43]. Noninvasive detection could improve cancer therapy response evaluation, enable earlier relapse detection and support risk stratification and prognostication. Herein, we propose a urinary nano-scale sensing platform (nanoprobe) that exploits the activity of a specific SASP protease to identify pro-inflammatory cells associated with chemotherapy-induced senescence. SASP profiles are dynamic and vary by tissue type and senescence trigger; therefore, enabling context-specific senescence detection and characterization of senescent cells[44,45]. Herein, we focused our efforts on senescent cells in lung tissue, specifically in the context of lung cancer emergence and development, and pulmonary fibrosis[46,47]. We profiled proteases secreted by chemotherapy-induced senescent lung cancer cells and identified matrix metalloproteinase-7 (MMP-7) as a context-specific senescence biomarker for in vivo sensor design.

To monitor MMP-7 activity, we developed an albumin-linked Au nanocluster (ALBANC) nanoprobe (Extended Data Fig. 1) comprising gold nanoclusters (AuNCs) as urinary reporters, human serum albumin as a protein carrier and an MMP-7-cleavable peptide linker. The ~10 nm nanoprobe is cleaved by MMP-7, releasing <2 nm AuNCs that pass the renal filtration cutoff (5–6 nm)[48]. Following renal clearance, AuNCs accumulate in urine, where they can be quantified by colorimetric and/or spectroscopic assays and linked to MMP-7 activity/level associated with chemotherapy-induced senescence. Urinary AuNC-based nanoprobes have been reported for bacterial implant infection[49] and colorectal cancer[50] detection. However, they relied on peroxidase-mimicking signal transduction and were not designed to report TIS states. Additionally, unlike prior AuNC-Neutravidin systems that detected colorectal cancer via MMP-9 activity and employed non-covalent biotin-avidin assembly[50], ALBANC is engineered to report TIS and uses a covalent albumin-linked architecture assembled via azide–DBCO click chemistry. Crucially, we demonstrate an alternative approach to conventional peroxidase assays for colorimetric detection by integrating a nanoparticle growth-based silver amplification strategy, yielding ~250-fold improved analytical sensitivity. Together, this enables a sensitive urinary readout of senescence-associated protease activity during chemotherapy, complementing histological assays and supporting longitudinal monitoring in our disease models.

MMP-7 has been implicated in multiple cancers and inflammatory conditions, and MMP-7-triggered release mechanisms for disease detection have been explored[51–54]. However, these systems did not enable urinary detection of in vivo MMP-7 level. Here, we use MMP-7 to specifically monitor chemotherapy-induced senescence (instead of pan-senescence). Furthermore, although prior research has investigated the use of serum proteins to delay renal clearance of nanoparticles[55], our approach uses proteolytic cleavage to regulate the specific release of AuNCs, enabling protease-dependent release and renal clearance of AuNCs. Additionally, although signal amplification through Au/Ag growth has been widely used to enhance detection sensitivity in vitro[56,57], we apply this technique to colorimetric in vivo detection via on absorbance readout.

In addition to the detection of senescence, ALBANC nanoprobes were used to monitor senolytic efficacy, supporting potential application for longitudinal monitoring. We also utilized ALBANC to detect pulmonary fibrosis, which is associated with elevated levels of MMP-7[46,47,58]. Across both disease models, baseline MMP-7 was low, but ALBANC detected elevated MMP-7 activity in senescent and fibrotic states, producing visible urinary signals. These results support sensitive urinary detection of senescence-associated MMP-7 activity in TIS, and suggest that the platform could be adapted to other disease contexts characterized by elevated protease levels.

## Results

### Chemotherapy-induced senescent lung cancer cells increased the secretion of MMP-7 in vitro and in vivo

To identify secreted proteases as potential TIS biomarkers in lung cancer, we developed an in vitro senescence model by treating A549 cells with four clinically relevant drugs: cisplatin, pemetrexed, docetaxel and palbociclib (Fig. 1a). Senescence was validated by increased SA-β-gal and p21 levels, reduced pRb expression and arrested growth (Fig. 1b and Supplementary Fig. 1a–d). Protease array analysis of the conditioned media revealed eight proteases were increased in secretion upon senescence induction, with MMP-7 showing the highest increase compared with non-senescent (untreated) A549 cells (Fig. 1c and Extended Data Fig. 2a). We selected MMP-7 for validation due to its most abundant secretion by senescent cells and because it has not been associated with age-related senescence[59]. Supporting the protease array results, enzyme-linked immunosorbent assay (ELISA) confirmed increased MMP-7 secretion in conditioned media (Fig. 1d), accompanied by increased intracellular MMP-7 protein and mRNA levels (Extended Data Fig. 2d,e). Similarly, elevated MMP-7 secretion was observed in murine L1475 lung cancer cells after cisplatin-induced senescence (Supplementary Figs. 1e–i and 2b and Extended Data Fig. 2f). However, fibroblast (HPF-a cells) did not secrete detectable levels of MMP-7 in either normal or senescent states (Supplementary Fig. 1j–n and Extended Data Fig. 2b, c, g). This observation is consistent with low physiological MMP-7 expression reported previously[60]. MMP-7 secretion also increased over time as more A549 cells entered senescence after cisplatin treatment (Supplementary Fig. 2).

The increase in MMP-7 expression was further validated in vivo in mice bearing flank A549 subcutaneous xenografts treated with vehicle, cisplatin or pemetrexed (Fig. 1e). Histology confirmed increased levels of senescence markers (SA-β-gal, p21) and decreased pRb in cisplatin- and pemetrexed-treated tumors (Fig. 1f,g), alongside reduced tumor growth, compared with the vehicle group (Supplementary Fig. 3). Drug-treated tumors also showed elevated MMP-7 expression, enriched in regions positive for senescence markers (Fig. 1f,g). Moreover, intratumoral p21+ levels correlated with circulating serum MMP-7 level (Extended Data Fig. 3a). When normalized to tumor volume, mice with higher senescence burden showed higher circulating MMP-7 before and after treatment (Extended Data Fig. 3b–f), whereas vehicle controls showed a slight, nonsignificant decrease. Together, these data indicate that TIS is associated with elevated tumor MMP-7 expression and increased circulating MMP-7, supporting the in vitro experimental results with conditioned media. Secreted MMP-7 levels also correlated with higher proteolytic activity relative to proliferating cells (Extended Data Fig. 4a–c). This finding was partly explained by higher MMP-7 levels and reduced levels of MMP-7 inhibitor, TIMP-1 (Extended Data Fig. 4d,e). Thus, this elevated secretion and

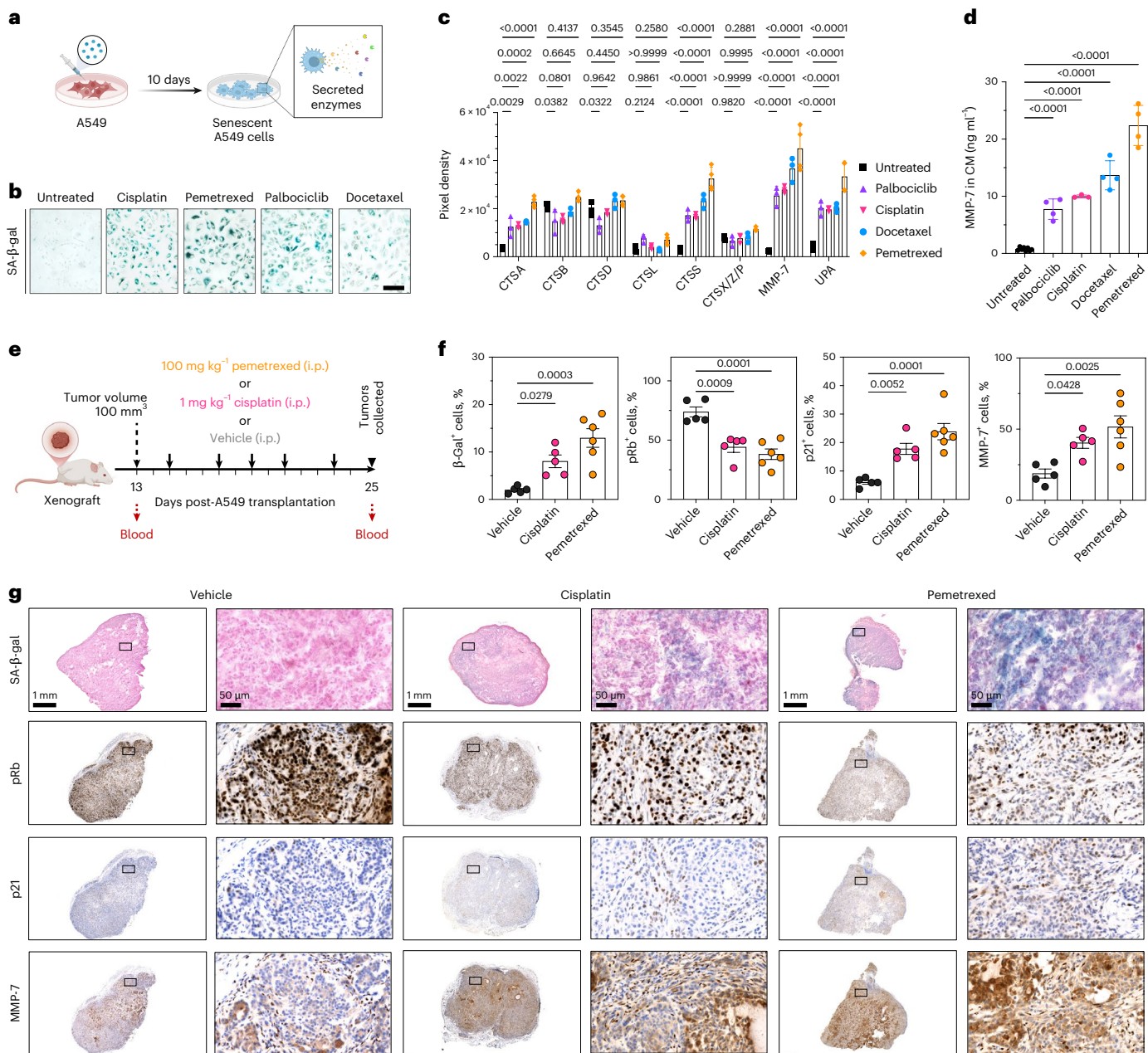

**Fig. 1 | Chemotherapy-induced senescent lung cancer cells abundantly secrete MMP-7 as part of their SASP. a**, A549 lung adenocarcinoma cells were treated with chemotherapy drugs for 10 days. **b**, Senescence induction was assessed by SA-β-gal activity staining (scale bars = 100 μm). Conditioned media of these senescent cells was screened for protease biomarkers. Figure created in BioRender. Fruk, L. (2026) https://BioRender.com/afyvgy3. **c**, Relative level of proteases secreted by chemotherapy-induced senescent cells and untreated A549 cells ($n = 3$ independent biological replicates per group except for $n = 4$ independent biological replicates for pemetrexed group, mean ± standard deviation (s.d.), two-way analysis of variance (ANOVA) with Dunnett's multiple comparisons). **d**, Concentration of MMP-7 in the conditioned media from senescent and non-senescent A549 cells ($n = 8$ independent biological replicates for the untreated group, $n = 4$ independent biological replicates for the palbociclib/docetaxel/pemetrexed group, $n = 3$ independent biological replicates for the cisplatin group; mean ± s.d., ordinary one-way ANOVA with

Dunnett's multiple comparisons). **e**, Schematic representation of experimental layout. Briefly, mice, transplanted subcutaneously with A549 cells, were subjected to pemetrexed treatment, cisplatin treatment or vehicle at the timings depicted. Peripheral blood was collected before treatment and at the endpoint (day 13). Tumors were resected at day 13 after initiation of treatment ($n = 5$ independent mice for the cisplatin and vehicle groups, $n = 6$ independent mice for the pemetrexed group). Mouse figure was created in BioRender. Fruk, L. (2026) https://BioRender.com/ruw7v3q. **f**, Quantification of SA-β-gal⁺, pRb⁺, p21⁺ and MMP-7⁺ cells per total cells in vehicle-treated or drug-treated specimens (mean ± s.d., ordinary one-way ANOVA with Dunnett's multiple comparisons). **g**, Representative histology images of tumor specimens resected from vehicle-treated or drug-treated mice and subjected to SA-β-gal, pRb, p21 and MMP-7 staining (performed on consecutive sections). Scale bars = 50 μm and 1 mm, as depicted.

activity supports the design of an MMP-7-responsive nanoprobe for TIS detection.

To corroborate these findings in a more physiologically relevant lung cancer model, we orthotopically transplanted the primary lung

cancer KRas^G12D/WT;p53^−/− (KP) L1475(luc) cell line in mice. Tumor-bearing mice received vehicle or one cycle of cisplatin (Supplementary Fig. 4a). Histology revealed increased levels of the senescence markers SA-β-gal, p16 and p21 and reduced levels of Ki-67 (a proliferation marker) in

cisplatin-treated tumors, whereas the senescence markers were largely absent in vehicle controls (Supplementary Fig. 4b,c). Importantly, the induction of p16 and p21 expression in the cisplatin group coincided with elevated MMP-7 staining, supporting the association between TIS and MMP-7 overexpression in vivo.

After identifying elevated MMP-7 secretion in senescent A549 cells, we asked whether this extends to other cancer types. We established TIS models in melanoma (SK-MEL-103), prostate adenocarcinoma (PC-3) and breast cancer (MDA-MB-231) cell lines. PC-3 cells were treated with docetaxel and SK-MEL-103/MDA-MB-231 with palbociclib (drugs selected for clinical relevance). As shown in Supplementary Fig. 5, all cell lines exhibited senescence after treatment. Conditioned media from senescent and control cells were profiled by protease arrays and compared with A549 profiles (Supplementary Fig. 6). MMP-7 elevation seen in senescent A549 cells was not observed in the other lines. Instead, senescent PC-3 cells increased MMP-13 secretion, SK-MEL-103 upregulated cathepsin A secretion and MDA-MB-231 increased MMP-3 secretion. These data support context-dependent SASP protease composition shaped by lineage, tissue of origin and treatment.

To further evaluate MMP-7 specificity in the context of chemotherapy-induced senescence, we performed two additional in vivo studies. First, we assessed age-associated senescence by staining lung, kidney, and heart from young (2 months) and aged (19 months) C57BL/6 mice for p16 (a marker of age-associated senescence)[2] and MMP-7 (Supplementary Fig. 7). As expected, p16 staining increased with age across organs, whereas MMP-7 levels did not differ between young and aged mice. This finding suggests that MMP-7 does not broadly increase in aged tissues. Second, we also tested whether cisplatin induces MMP-7 in healthy (non-tumor) tissues by treating C57BL/6 mice with two cycles of cisplatin or vehicle (Supplementary Fig. 8). Lung histology showed increased p21 levels after cisplatin, consistent with a senescence response. However, MMP-7 levels in the lungs remained comparable between cisplatin and vehicle groups. Similar results were obtained in kidney and heart tissues, where MMP-7 expression did not significantly differ following chemotherapy. Together, these results indicate that MMP-7 upregulation is not a general feature of age-associated or damage-induced (non-tumor) tissue senescence but is preferentially elevated in chemotherapy-treated lung tumors in our models, supporting MMP-7 as a context-specific biomarker of TIS in lung cancer.

## ALBANC nanoprobes enable detection of MMP-7 activity by colorimetric assays

To target the proteolytic activity of MMP-7, the ALBANC nanoprobe was assembled by click chemistry conjugating azidopeptide-functionalized

AuNCs to DBCO-functionalized human serum albumin (Fig. 2a). AuNCs were synthesized in a one-pot reaction using chloroauric acid ($HAuCl_4$), glutathione (GSH) and an MMP-7-cleavable peptide (Supplementary Fig. 9a). The peptide included an MMP-7-cleavable sequence, a C-terminal cysteine for Au binding and an N-terminal azide for albumin conjugation. The resulting AuNCs were 1.6 ± 0.3 nm in diameter with an ~3 nm hydrodynamic size, which is below the glomerular filtration cutoff ( ~ 6 nm) (Fig. 2b and Supplementary Fig. 9b,c). These AuNCs were negatively charged and showed small-molecule-like absorption and fluorescence (Supplementary Fig. 9e–g). AuNCs with varying peptide loading were prepared and characterized (Supplementary Fig. 10-12). Human serum albumin was selected as the protein carrier for its biocompatibility and prolonged circulation time. In addition, DBCO functionalization did not affect albumin structural integrity (Supplementary Figs. 13a–g and 14), and albumin was not cleaved by MMP-7 or other relevant proteases (Supplementary Fig. 13h).

Successful nanoprobe assembly was verified by gel electrophoresis and dynamic light scattering (DLS), which showed an increased size after conjugation, and by transmission electron microscopy (TEM), which confirmed albumin-bound AuNCs (ALBANC nanoprobes) (Fig. 2c,d and Supplementary Fig. 15). Optimization of conjugation (Supplementary Figs. 16 and 17) yielded negatively charged ALBANC with ~1 AuNC per albumin and an ~11 nm hydrodynamic size, above the glomerular filtration limit (Supplementary Fig. 15c-e). AuNC and ALBANC nanoprobe remained stable in vitro after incubation in urine and physiological concentrations of GSH (Supplementary Fig. 18). Additionally, control nanoprobes (containing non-cleavable linker) and a fluorophore-labeled variant were also successfully prepared (Supplementary Figs. 19-20), supporting design adaptability.

The ALBANC nanoprobe functions via size-dependent biodistribution. Upon cleavage by MMP-7, the AuNC reporters are released from albumin, allowing circulation and renal excretion due to their small size. AuNCs were chosen as urinary reporters due to their biocompatibility, efficient renal clearance and peroxidase-like activity, enabling detection via simple colorimetric assays (Fig. 2f and Supplementary Fig. 21). Here, AuNCs catalyze 3,3′,5,5′-tetramethylbenzidine (TMB) oxidation into a blue oxidation product in the presence of hydrogen peroxide ($H_2O_2$), enabling a colorimetric readout as the rate of absorbance at 652 nm increase over time ($A_{652nm}$/s) (Supplementary Fig. 22). In synthetic urine, the peroxidase assay yielded a limit of detection (LoD) of ~100 nM (19 pmol AuNC) with ~15 min detection without sample pretreatment (Fig. 2g).

To improve sensitivity, we developed a colorimetric assay in which AuNCs seed nanoparticle growth rather than relying on peroxidase-like activity. Inspired by Au/Ag signal amplification methods used in optical

**Fig. 2 | Preparation of ALBANC nanoprobe and its colorimetric detection assays. a**, Schematic illustration of the two-step synthesis of MMP-7 nanoprobe. First, AuNC bearing azide-functionalized peptides is synthesized using a one-pot protocol. Subsequently, AuNC is conjugated to DBCO-functionalized albumin via click chemistry reaction, resulting in the formation of the nanoprobe. Created in BioRender. Fruk, L. (2026) https://BioRender.com/r46tl5v. **b,c**, TEM image of AuNCs (**b**; mean diameter ± s.d., 1.6 ± 0.3 nm, n = 200 particles) and nanoprobe (**c**). Scale bars, 5, 10 or 50 nm. Experiment was repeated independently three times with similar results. **d**, Gel electrophoresis showing the assembly of nanoprobe with Coomassie stain (top) and luminol (bottom, stains for the AuNC). After conjugation, the AuNC is attached to albumin, as indicated by the presence of a single band around 70 kDa. NP, nanoprobe. Three independent experiments (three biological replicates) were performed, yielding similar results. **e**, Schematic illustration of two colorimetric assays for the detection of AuNC in synthetic urine: (1) peroxidase assay and (2) alloy formation assay, which involves buffer exchange of the AuNC into water via ultrafiltration. Created in BioRender. Fruk, L. (2026) https://BioRender.com/2id8x31. **f**, Representative UV-Vis spectra showing increase in absorbance at both 370 nm and 652 nm correlated to TMB oxidation in the presence of AuNCs. Inset shows photographs of substrate without AuNC (left) and substrate with AuNCs yielding blue color

development (right). Three independent experiments were performed with similar results. **g**, The limit of detection (LoD) of AuNC in synthetic urine was measured by peroxidase assay, by plotting the initial velocity of TMB oxidation ($A_{652nm}$/s, n = 9 independent experiments, mean ± s.d., log-log fit). Peroxidase activity of AuNC is linear over 100 to 10,000 nM of AuNC concentration, with a log scale regression $R^2$ = 0.9981. The dashed line indicates the LoD, calculated as 3 s.d. (3σ) above the mean background signal. **h**, TEM image of formed from AuNC-Ag alloy nanoparticles upon reaction with the substrate mix ($AgNO_3$, CTAC and ascorbic acid). Three independent experiments were performed, yielding similar results. **i**, Elemental map photographs of Au-Ag alloy nanoparticles showing the presence of both Au and Ag. Three independent experiments were performed, yielding similar results. **j**, The formed Au-Ag alloy nanoparticles produced yellow color (λ peak = 414 nm). Three independent experiments were performed with similar results. **k**, The LoD of AuNC in synthetic urine measured by the alloy formation assay (n = 6 independent experimental replicates, involving independently synthesized AuNC batches, mean ± standard error of the mean (s.e.m.), log-log fit). Alloy formation assay is linear over 0.4 to 1,000 nM of AuNC concentration, with a log scale regression $R^2$ = 0.9309. The dashed line indicates the LoD.

microscopy[61], we developed a protocol that produces yellow-colored AuNC-Ag alloy nanoparticles ($\lambda_{max}$ = 414 nm) using ascorbic acid and CTAC, with renally excreted AuNCs acting as seeds (Fig. 2h–j and Supplementary Fig. 23). TEM and elemental mapping confirmed AuNC-Ag alloy formation with a size of 30.9 ± 7.8 nm and uniform Au/Ag distribution (Fig. 2i and Supplementary Fig. 24). We hypothesize that carboxylate groups on AuNCs play a key role in facilitating alloy formation. Consistent with Ag[+] affinity for carboxylates, AuNC-Ag formation was reduced when GSH was replaced with cysteine methyl ester that lacks carboxylate groups (Supplementary Fig. 25)[62].

Before the alloy formation assay, a buffer exchange step is required to transfer AuNCs from urine to deionized water. This step eliminates absorbance overlap with urine color, minimizes

differences in salt content that could influence alloy formation, reduces matrix effects and improves assay consistency. Assay optimization (Supplementary Fig. 26) achieved an LoD of 0.4 nM (0.076 pmol) for AuNCs (Fig. 2k), representing a 250-fold improvement over the previous peroxidase assay. Overall, the integration of the ALBANC nanoprobe with these two assays provides a colorimetric platform for detecting cleaved AuNCs in response to MMP-7 activity.

We then evaluated nanoprobe cleavage and assay readouts using recombinant MMP-7 (Fig. 3a–c). Gel electrophoresis confirmed cleavage, with ~80% AuNC released within 2 h and ~90% by 20 h (Fig. 3b; Supplementary Fig. 27). We also assessed AuNC cleavage by MMP-7 using peroxidase and alloy assays across MMP-7 concentrations (Fig. 3c). Both assays detected MMP-7 activity at ~1 nM, comparable

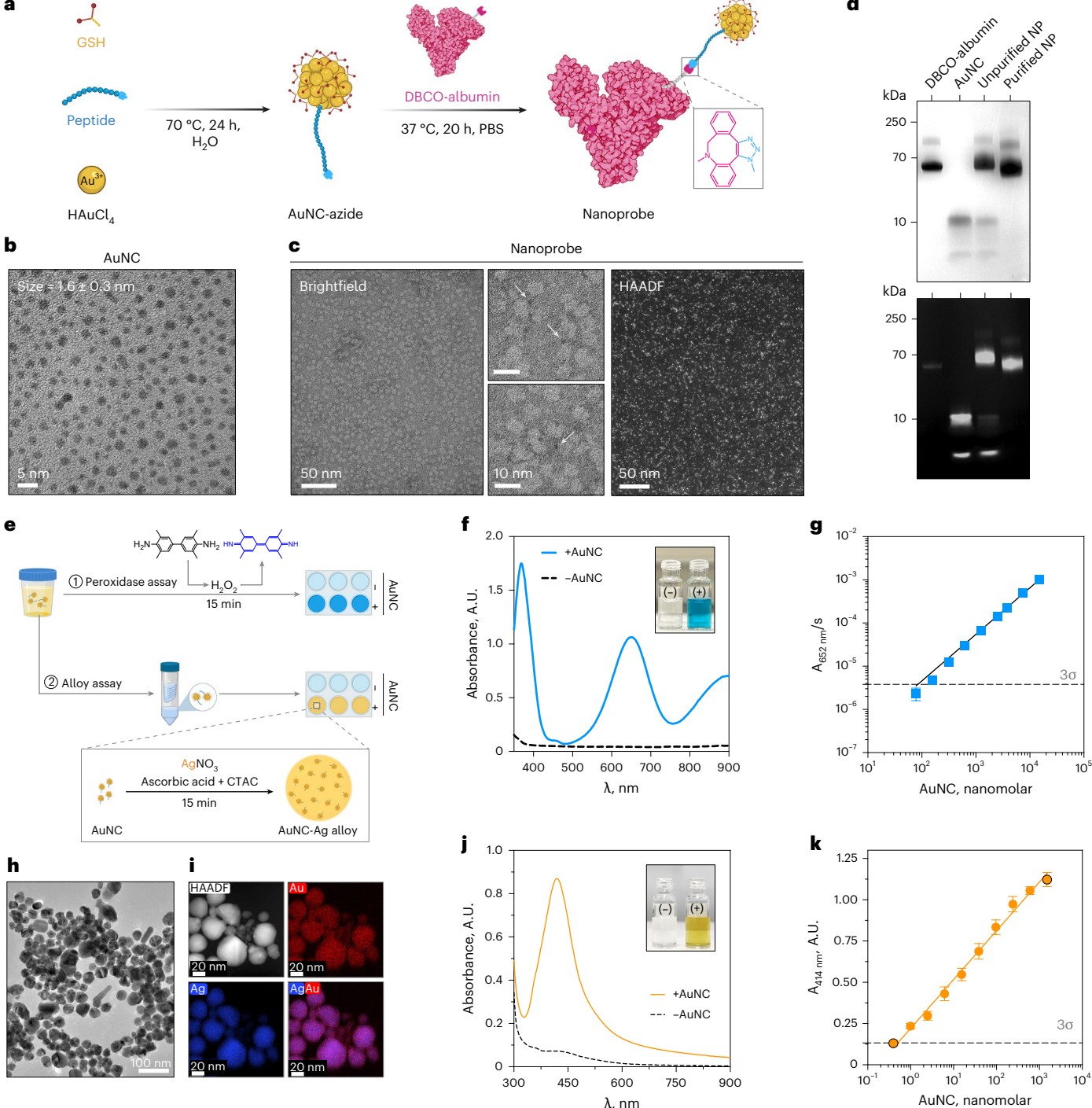

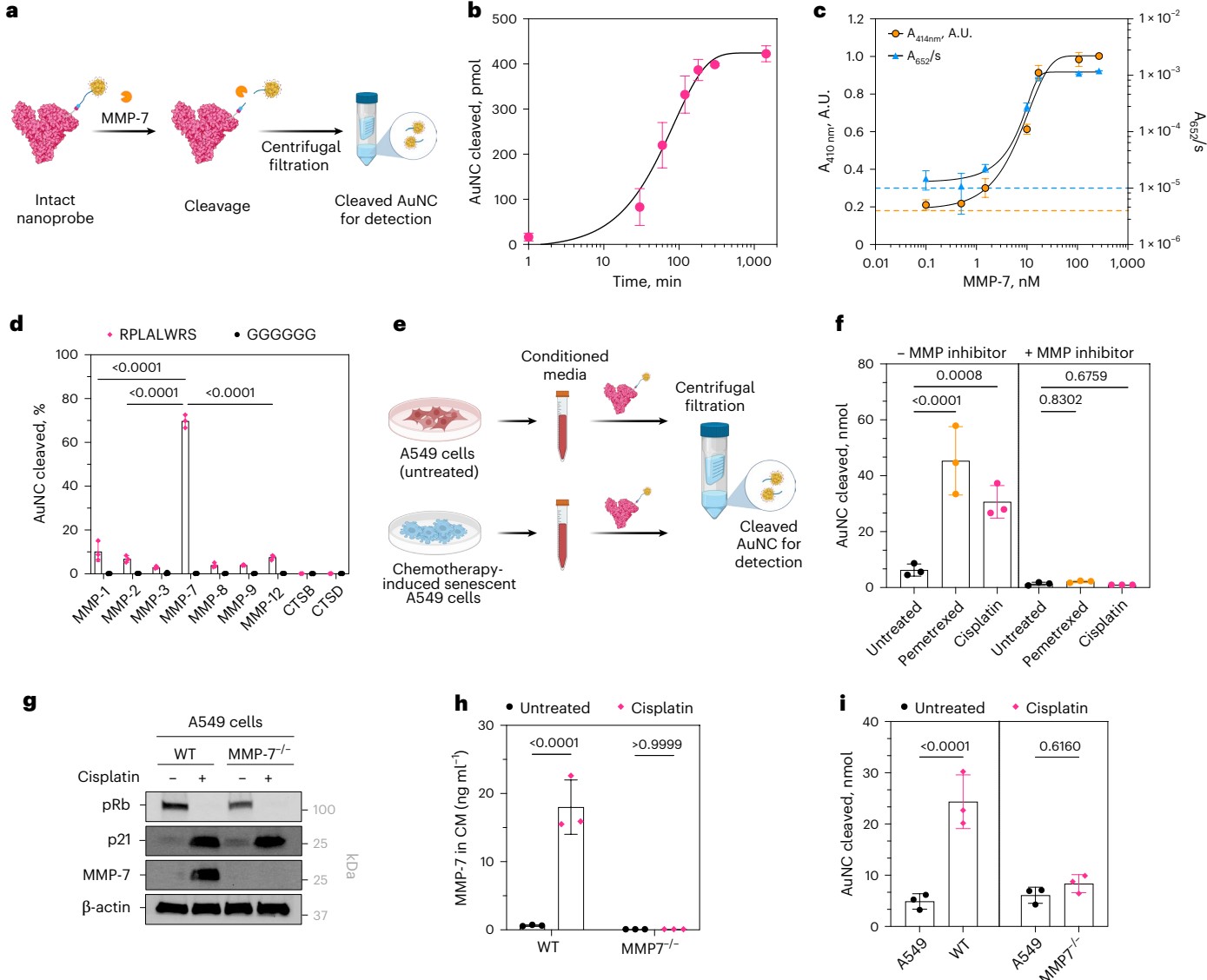

**Fig. 3 | ALBANC nanoprobe responds to protease activity of MMP-7 in vitro.**
**a**, Recombinant MMP-7 detection using the nanoprobe and alloy formation assay. Upon incubation with MMP-7, the nanoprobe suspension is separated using ultracentrifugal filters to concentrate the cleaved AuNC in the filtrate for detection. Created in BioRender. Fruk, L. (2026) https://BioRender.com/nn2d1dp. **b**, Plot of moles of cleaved AuNC from the nanoprobe after incubation with recombinant MMP-7 for various durations ($n = 3$ independent experiments, mean ± s.d., nonlinear fit), as detected using alloy formation assay. **c**, Plot of absorbance (from the alloy formation assay) and $A_{652nm}/s$ (from the peroxidase assay), proportional to the cleaved AuNC from the nanoprobe after incubation with different concentrations of recombinant MMP-7 ($n = 3$ independent experimental replicates/independent experiments per assays, mean ± s.d., nonlinear fit). The dashed lines indicate the threshold for detection, determined to be three standard deviations higher than the average background signal. **d**, Plot of cleaved AuNC from the nanoprobe after incubation with various relevant enzymes ($n = 3$ independent experimental replicates/independent experiments, mean ± s.d., ordinary two-way ANOVA with Dunnett's multiple comparisons). **e**, Schematic illustration of MMP-7 detection in the conditioned media of A549 and senescent A549 cells using ALBANC nanoprobe. Created in BioRender. Fruk, L. (2026) https://BioRender.com/lbzhy3k. **f**, Plot of AuNC cleaved from the nanoprobe (mole), as detected using alloy formation assay, after incubation in conditioned media from either A549 or senescent A549 cells, with and without EDTA ($n = 3$ biological replicates per group, mean ± s.d., ordinary one-way ANOVA with Dunnett's multiple comparisons). **g**, Western blot for the expression of MMP-7 and relevant senescence markers (pRb and p21) in untreated and cisplatin-treated wild-type (WT) or MMP-7-deficient (MMP-7$^{-/-}$) A549 cells. Three independent experiments (three biological replicates) were performed yielding similar results. **h**, Concentration of MMP-7 in the conditioned media from senescent and non-senescent WT versus MMP-7$^{-/-}$ A549 cells ($n = 3$ biological replicates per group, mean ± s.d., ordinary one-way ANOVA with Dunnett's multiple comparisons). **i**, Plot of AuNC cleaved from the nanoprobe (mole), as detected using alloy formation assay, after incubation in conditioned media from either non-senescent versus cisplatin-induced senescent WT or MMP-7$^{-/-}$ A549 cells ($n = 3$ biological replicates per group, mean ± s.d., ordinary one-way ANOVA with Šidák multiple comparisons test).

to fluorescence-based commercial assays. Furthermore, incubation with different proteases showed that the nanoprobe was preferentially cleaved by MMP-7, with minor cleavage by MMP-1, MMP-2 and MMP-12 (Fig. 3d). This is consistent with overlapping substrate specificities within the MMP family[63,64]. Importantly, the control

nanoprobe (non-cleavable linker) showed no cleavage, supporting MMP-7-dependent AuNC release (Fig. 3d).

We further validated nanoprobe activation using conditioned media from senescent A549 cells (Fig. 3e,f). Ethylenediaminetetraacetic acid (EDTA; a protease inhibitor) treatment minimized cleavage and

eliminated differences between senescent and nonsenescent media. To further confirm MMP-7 specificity, we generated MMP-7-deficient A549 cells (MMP-7$^{-/-}$ A549) and induced senescence with cisplatin (Supplementary Fig. 28). Western blot confirmed loss of MMP-7 in MMP-7$^{-/-}$ A549 cells, even when these MMP-7$^{-/-}$ A549 cells are senescent (Fig. 3g). Moreover, ELISA showed negligible MMP-7 secretion from MMP-7$^{-/-}$ A549 cells, whereas WT A549 cells secreted higher MMP-7 upon senescence induction, as previously shown (Fig. 3h). Accordingly, conditioned media from MMP-7$^{-/-}$ A549 cells did not produce senescence-dependent cleavage of the nanoprobe, whereas media from senescent WT cells did (Fig. 3i). By contrast, significant nanoprobe cleavage was detected in conditioned media from senescent WT A549 cells, consistent with their elevated MMP-7 levels. Together, these results support specific detection of MMP-7 activity in conditioned media using the nanoprobe.

To evaluate biocompatibility, cell viability assays were first performed across multiple cell lines using AuNC and ALBANC nanoprobes at concentrations up to 15 μM, with no observed reduction in cell viability (Supplementary Fig. 29a–c). For in vivo toxicity assessment, 3 nmol nanoprobes (based on AuNC content) was intravenously injected into healthy C57BL/6 mice. Over 15 days, body weight, clinical observation and histology showed no overt toxicity in major organs (Supplementary Fig. 29d). Histological analysis also revealed no signs of toxicity or necrosis in the heart, lungs, liver, spleen or kidneys after injection (Supplementary Fig. 29e). To evaluate biodistribution, fluorescence imaging was performed, showing kidney accumulation of free AuNCs at 2 h, and inductively coupled plasma mass spectrometry (ICP-MS) indicated ~60% urinary excretion by 24 h (Extended Data Fig. 5a,b,e), consistent with prior reports[50,65]. In contrast, ALBANC was not detected in urine 2 h after injection (Extended Data Fig. 5c,d). Instead, ALBANC accumulated in reticuloendothelial system organs like the liver and spleen. This finding important, because both assays can detect uncleaved, albumin-bound AuNC or intact ALBANC (Supplementary Fig. 30), which would otherwise interfere with urine detection. By day 15, signals were near background in major organs (Extended Data Fig. 5f,h), suggesting substantial clearance over the tested period. Additionally, pharmacokinetic analysis indicated that ALBANC exhibited a circulation half-life of approximately 50.3 min (Extended Data Fig. 5i).

The colorimetric assays semi-quantitatively detected renally cleared AuNCs in urine from healthy C57BL/6 mice. Urinary signals correlated with Au content by ICP-MS (Pearson $r = 0.67$ and $0.65$) (Extended Data Fig. 6a–c). TEM/EDX confirmed the renally cleared AuNCs as elemental gold (Extended Data Fig. 6d–f). The nanoclusters retained their original shape and size distribution, consistent with preserved peroxidase activity and alloy-forming behavior after in vivo administration.

## Platinum TIS correlates with increased MMP-7 expression in human NSCLC

After validating MMP-7 overexpression in senescence and developing an MMP-7-responsive nanoprobe, we evaluated NSCLC patient specimens after neoadjuvant platinum chemotherapy for MMP-7 expression (Fig. 4). We assessed proliferation and senescence markers in stage III lung adenocarcinoma (LUAD) tumors collected 1 month after neoadjuvant chemotherapy and compared them with treatment-naïve stage I–IV tumors. Ki-67 staining was variable in treatment-naïve tumors but reduced in those after chemotherapy (Fig. 4b,c and Supplementary Fig. 31). Compared with treatment-naïve patients, tumors from cisplatin-treated patients had significantly more cells expressing p21, mainly found at the tumor periphery and more scattered toward the tumor core, indicating a higher senescence burden. The p21 level was inversely correlated with Ki-67, consistent with senescence induction in platinum-treated group. Treatment-naïve specimens showed minimal MMP-7 staining, whereas platinum-treated tumors showed increased MMP-7 in cytoplasmic and extracellular matrix (ECM) regions (Fig. 4b,d). Furthermore, consecutive staining sections showed co-localization of high MMP-7 and p21 staining (Fig. 4b and Extended Data Fig. 7). This elevated MMP-7 expression in areas densely populated with p21$^+$ cells supports an association between p21 positivity and MMP-7 upregulation after platinum chemotherapy.

To further assess this in patient data, we reanalyzed a publicly available single cell RNA sequencing (scRNA-seq) dataset of naïve lung adenocarcinoma (LUAD) patients (Control) and LUAD patients treated with comparing treatment-naïve LUAD with neoadjuvant chemotherapy/NCT (pemetrexed/cisplatin)[66]. The analysis shows that CDKN1A (p21) was expressed across several cell types, whereas MMP7 and CDKN2A (p16) expression was restricted to malignant epithelial cells from LUAD tumors (Fig. 4e). Within malignant epithelial cells, re-clustering showed higher CDKN1A, CDKN2A and MMP7 expression in NCT-treated tumors (Fig. 4f,g). In agreement with our previous results, our analyses revealed that NCT induces a higher expression

**Fig. 4 | Platinum chemotherapy induces senescence in human NSCLC and is associated with elevated MMP-7 expression across histological and single-cell analyses. a**, Platinum-based chemotherapy administered before surgery (neoadjuvant) induces senescence in NSCLC patients and this correlates with increased MMP-7 expression in surrounding cells. Schematic representation of human NSCLC samples analyzed in this study. Created in BioRender. Fruk, L. (2026) https://BioRender.com/65hu5ir. **b**, Representative histological images of NSCLC biopsy samples from treatment-naïve patients and platinum chemotherapy, analyzed for Ki-67, p21 and MMP-7. Scale bars = 200 μm (top row) or 50 μm (bottom row). **c**, Quantification of Ki-67$^+$, p21$^+$ and MMP-7$^+$ cells per total cells in treatment-naïve or platinum-treated specimens ($n = 3$ independent patients per group; biological replicates). Data points represent seven representative images quantified per patient sample (mean ± s.d., unpaired t-test, two tailed). **d**, Quantification of MMP-7$^+$ cells per total cells in p21-low and p21-high areas in platinum-treated specimens. Here, $n = 3$ independent patients per group (biological replicates). Data are presented as mean ± s.d. Statistical significance was determined using an unpaired two-tailed t-test. High-p21 regions were defined as areas with >10% p21$^+$ cells (relative to total cells), and low-p21 regions as areas with <10% p21$^+$ cells. Each dot represents an independent tumor area (5 areas per patient sample for high-p21 regions; 2 areas per patient sample for low-p21 regions). **e**, Dot plot showing the scaled expression of CDKN1A, CDKN2A and MMP7 across all cell types from a single-cell RNA-seq dataset of naïve lung adenocarcinoma patients (control, $n = 4$) and lung adenocarcinoma patients treated with neoadjuvant chemotherapy (NCT) ($n = 5$) (Huang et al., 2024 PMID: 39729352). The color shows the average scaled expression of each gene, and the size of the dots represents the percentage of cells expressing a given gene as measured by single-cell RNA-seq. **f**, Color-coded uniform manifold approximation and projection (UMAP) plot showing the clustering of epithelial cells according to the sample ID (left) or according to the treatment group (right) from a single-cell RNA-seq lung adenocarcinoma dataset of naïve patients (control, $n = 4$) and treated with neoadjuvant chemotherapy (NCT) ($n = 5$) (Huang et al., 2024 PMID: 39729352)[66]. **g**, UMAP plot showing the average scaled expression of EPCAM in epithelial from a single-cell RNA-seq dataset of naïve lung adenocarcinoma patients (control, $n = 4$) and lung adenocarcinoma patients treated with neoadjuvant chemotherapy (NCT) ($n = 5$) (Huang et al., 2024 PMID: 39729352). **h**, Dot plot showing the scaled expression of CDKN1A, CDKN2A and MMP7 epithelial cells from a single-cell RNA-seq dataset of naïve lung adenocarcinoma patients (control, $n = 4$) and lung adenocarcinoma patients treated with neoadjuvant chemotherapy (NCT) ($n = 5$) (Huang et al., 2024 PMID: 39729352). The color shows the average scaled expression of each gene, and the size of the dots represents the percentage of cells expressing a given gene as measured by single-cell RNA-seq. **i**, Enrichment of senescence and drug treatment related pathways within upregulated genes (log$_2$ fold change [31] and false discovery rate (FDR) < 0.05) between lung adenocarcinoma patients treated with neoadjuvant chemotherapy (NCT) ($n = 5$) and naïve lung adenocarcinoma patients (control, $n = 4$) (Huang et al., 2024 PMID: 39729352). The bar size represents the number of genes enriched on each pathway. The FDR value of each enrichment is located next to each bar.

of *CDKN1A*, *CDKN2A* and *MMP7* (Fig. 4h). Furthermore, differential expression analysis of malignant epithelial cells between NCT-treated and control LUAD showed a significant enrichment of pathways related to senescence, ECM and cytokine-to-receptor interactions and drug metabolism (Fig. 4i). Collectively, these data support an association between chemotherapy-induced senescence in human lung tumors (NSCLC and LUAD) and increased MMP-7 levels, motivating MMP-7 activity as a marker of TIS in lung cancer.

## Urinary detection of chemotherapy-induced senescence in lung cancer

We evaluated ALBANC detection of TIS burden in a human lung cancer xenograft model in athymic nude mice. Mice were treated with either vehicle or cisplatin (Fig. 5a), a third group received both cisplatin and ABT-737 (a senolytic drug), and a fourth group received ABT-737 alone[67,68]. Treatment commenced once tumors reached an average volume of 100 mm³, and administered thrice weekly with cisplatin and ABT-737 on alternating days for 15 days. At endpoint, nanoprobes were

injected intravenously (i.v.) and urine collected 2 h later. Histology showed increased SA-β-gal and p21, reduced pRb, and elevated MMP-7 in cisplatin-treated tumors (Fig. 5b,d). In contrast, the combination treatment with senolytic ABT-737 resulted in reduced levels of SA-β-gal, p21 and MMP-7, consistent with reduced senescence burden. ABT-737 treatment alone produced minimal changes versus vehicle. Consistent with these findings, cisplatin treatment (without a senolytic) led to reduced tumor growth (Fig. 5c), whereas ABT-737 alone had no significant impact on tumor growth. The combination treatment further inhibited tumor growth, consistent with prior reports that eliminating therapy-induced senescent cells can enhance anti-tumor responses[67–70]. Together, these data show that cisplatin induces senescence in xenografts and that ABT-737 co-treatment reduces this burden.

At the endpoint, urine was collected and analyzed by the peroxidase assay and the alloy formation assay (after buffer-exchanged into deionized water) to quantify cleaved AuNCs. Yellow color formation with a 414-nm absorbance peak indicated AuNC-Ag alloy nanoparticle formation from renally cleared AuNCs. Cisplatin-treated mice

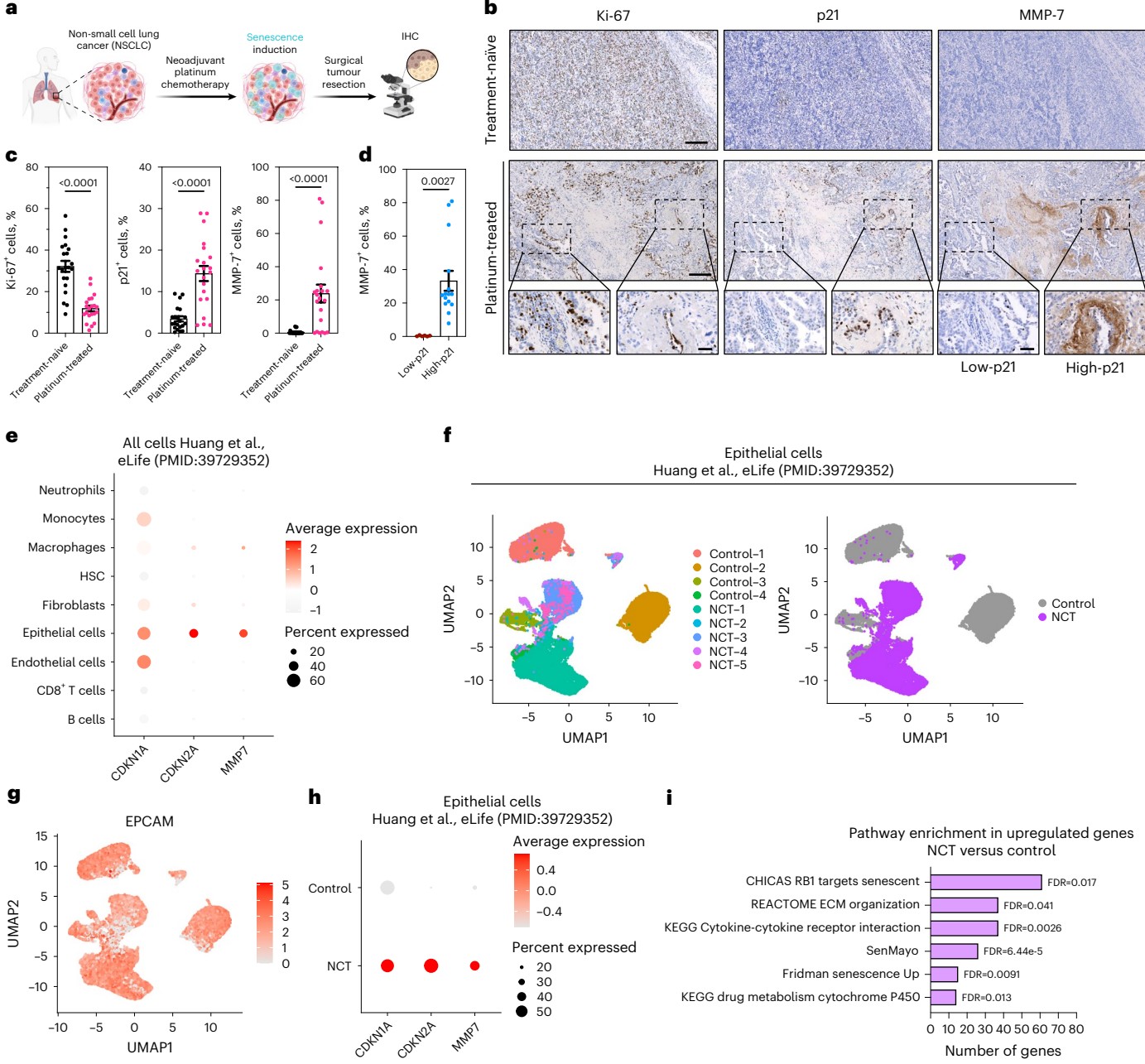

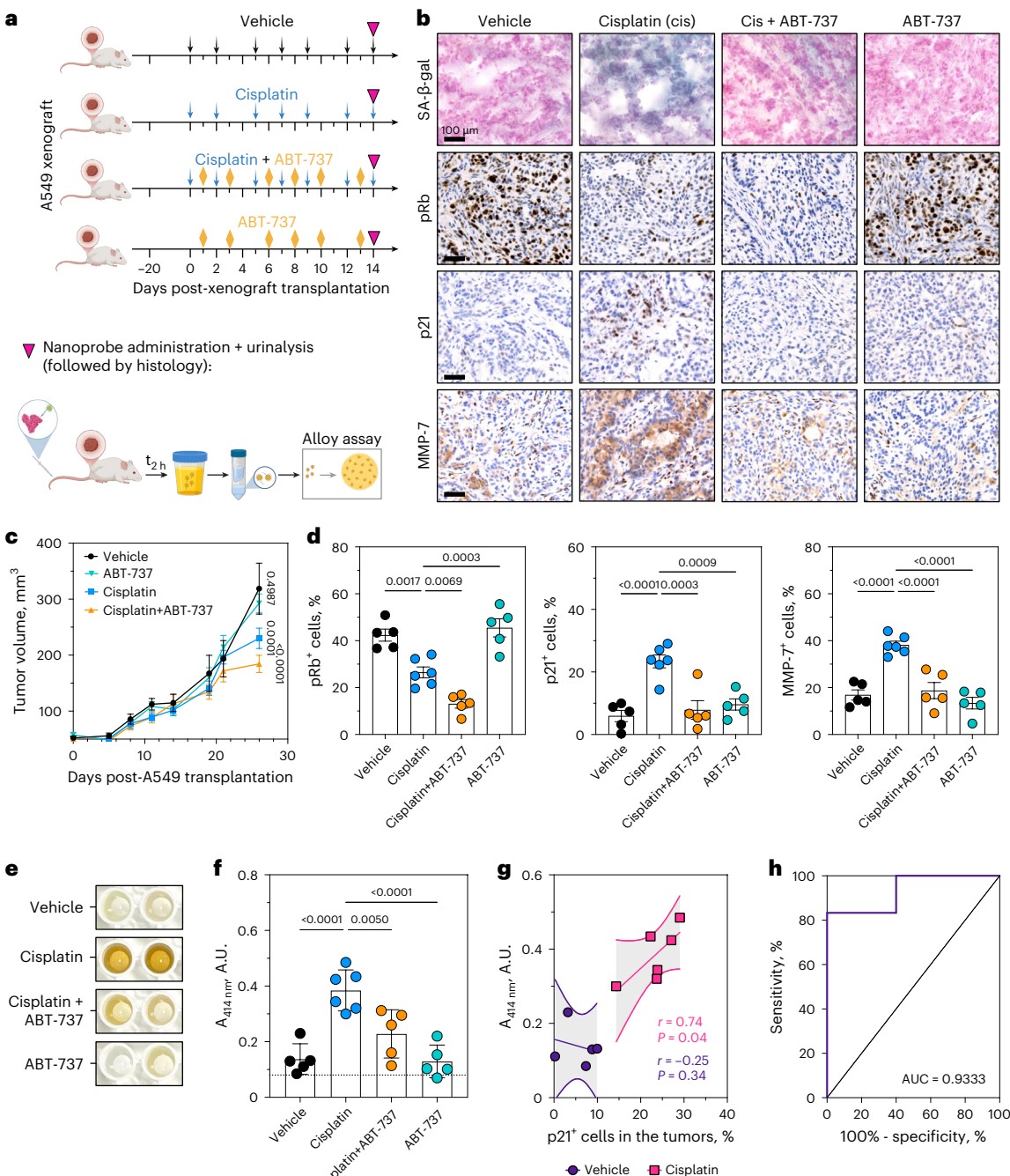

**Fig. 5 | Urinary detection of chemotherapy-induced senescence in lung cancer. a**, Schematic showing the treatment protocol for mice bearing A549 xenografts. At the endpoint, mice were administered with nanoprobes i.v. Urine was collected 2 h post-injection (p.i.) of nanoprobes, and the renally cleared AuNC was detected using colorimetric assays. Created in BioRender. Fruk, L. (2026) https://BioRender.com/8yex08h. **b**, Representative IHC images of tumors at the endpoint, stained for SA-β-gal, pRb, p21 and MMP-7 expression. Scale bar = 100 μm. **c**, Volume of tumors from cisplatin, cisplatin and ABT-737, ABT-737, or vehicle groups (n = 5 independent mice per group for vehicle, cisplatin + ABT-737, and ABT-737 groups, n = 6 independent mice for cisplatin group, mean ± s.e.m., two-way ANOVA with Dunnett's multiple comparisons test). **d**, Percentage of pRb+, p21+ and MMP-7+ cells in tumors from mice treated with vehicle; cisplatin; cisplatin and ABT-737; or ABT-737 (n = 5 independent mice per group for vehicle, cisplatin + ABT-737, and ABT-737 groups, n = 6 independent mice for cisplatin group, mean ± s.e.m, ordinary one-way ANOVA with Dunnett's multiple comparison test). **e**. Photograph of the alloy formation assay on urine samples from xenograft mice injected with nanoprobes. **f**. Absorbance

values ($A_{414 nm}$) from alloy formation assays of urine samples collected from xenograft mice 2 h p.i. with nanoprobes (n = 5 independent mice per group for vehicle, cisplatin + ABT-737, and ABT-737 groups, n = 6 independent mice for cisplatin group, mean ± s.e.m, ordinary one-way ANOVA with Dunnett's multiple comparison test). **g**, Correlation analysis between the percentage of p21+ cells in the tumors from the xenograft mice, either cisplatin-treated or vehicle-treated group, and intensity of signals from urine samples detected using the alloy formation assay ($A_{414 nm}$). Each symbol represents an individual mouse (vehicle, n = 5; Cisplatin, n = 6). Solid lines indicate linear regression fits (least-squares fit; estimated mean relationship between variables), and dashed lines denote the 95% confidence intervals of the regression line. One-tailed Pearson correlation coefficients (r) and corresponding one-tailed P values are shown. **h**. ROC analysis showing the diagnostic specificity and sensitivity of nanoprobes and alloy formation assay in detecting senescence between cisplatin-treated and vehicle-treated mice groups (AUC = 0.9333, 95% CI = 0.7797 – 1, P = 0.0176). The solid line represents the ROC curve, the dashed diagonal line represents the performance of a random classifier (AUC = 0.5).

showed ~3.5-fold higher urinary signal than vehicle by direct colorimetric readout (Fig. 5e,f). In contrast, cisplatin+ABT-737 co-treatment reduced the urinary signal to near vehicle levels, consistent with reduced senescence burden. The ABT-737 only group had urinary signals similar to the vehicle group. This pattern correlated well with the senescence burden observed in tumors through histology, as urinary signals in cisplatin-treated mice showed a positive correlation with the expression of senescence marker p21[+] in the tumors (Fig. 5g). Receiver-operating-characteristic (ROC) analysis yielded an area under the curve (AUC) of 0.9333 (95% confidence interval (CI) of 0.7797 - 1), indicating high sensitivity and specificity in detecting the senescence burden (Fig. 5h).

The same urine samples were analyzed by a peroxidase assay measuring the initial rate of TMB oxidation by AuNCs ($A_{652\,nm}$/s) present in the urine (Extended Data Fig. 8a). A similar trend was observed. Signals were highest in cisplatin-treated mice and reduced with cisplatin + ABT-737. Urinary signals also correlated positively with the percentage of p21[+] cells in the tumors (Extended Data Fig. 8c). However, no visible color differences were observed (Extended Data Fig. 8b), consistent with lower peroxidase assay sensitivity than the alloy assay. ROC analysis (Extended Data Fig. 8d) yielded an AUC of 0.9333 (95% CI: 0.7787 - 1), confirming the accuracy of the peroxidase assay in distinguishing senescence burden in cisplatin-treated versus vehicle-treated mice spectroscopically.

Serum MMP-7 level was measured by ELISA in individual mice before and after treatment (Extended Data Fig. 8e). This within-mouse comparison enabled longitudinal assessment across groups. Serum MMP-7 level increased modestly in all groups but rose most in cisplatin-treated mice, yielding the highest endpoint levels (Extended Data Fig. 8f). In cisplatin-treated mice, serum MMP-7 correlated with urinary AuNC signals measured by both alloy and peroxidase assays (Extended Data Fig. 8g,h). These findings are consistent with MMP-7-mediated nanoprobe cleavage contributing to the urinary AuNC signals. Together, these data show that systemic ALBANC administration in lung xenograft-bearing mice, coupled with urinary colorimetric/spectroscopic readouts, produces AuNC signals that track senescence burden. Urinary signals also decreased after ABT-737 co-treatment, consistent with reduced senescent cell burden due to the senolytic effect.

### Use of ALBANC nanoprobe for urinary detection of pulmonary fibrosis

Next, we asked whether ALBANC could detect pulmonary fibrosis, where MMP-7 levels are elevated and have been proposed as a prognostic marker[71,72]. Given links between senescence and fibrotic progression[73] we hypothesized that fibrosis-associated MMP-7 activity could activate ALBANC. We stained lung sections from patients with idiopathic pulmonary fibrosis (IPF) for fibrosis markers (α-SMA, Masson's trichrome), senescence markers (p21, p16, GL-13) and MMP-7 (Fig. 6). Fibrotic regions showed higher senescence markers and MMP-7 than background regions (<5% α-SMA positivity; <15% Masson's staining) (Fig. 6b). Moreover, these fibrotic areas overlapped spatially with high senescence marker expression and MMP-7 positivity (Fig. 6a,b). These findings suggest that MMP-7 expression is closely associated with both fibrosis and cellular senescence in human lung tissue, supporting MMP-7 as a fibrosis-associated marker linked to senescence-associated regions.

We next tested the ALBANC nanoprobe using a bleomycin-induced fibrosis model in C57BL/6 mice (14 days treatment; intratracheal bleomycin versus untreated controls) (Fig. 7a). Lungs were collected at endpoint for histology. Micro-computed tomography (micro-CT) of mice showed structural remodeling and parenchymal thickening in bleomycin-treated lungs (Fig. 7b). Moreover, Masson's trichrome and α-SMA staining of the collected lungs showed increased collagen and myofibroblasts (Fig. 7c,d and Extended Data Fig. 9). MMP-7 staining was

elevated in collagen-rich, α-SMA[+] areas, and serum MMP-7 increased in bleomycin-treated mice (Supplementary Fig. 32). Bleomycin-treated lungs also showed increased p16 and p21 staining (Fig. 7c), indicating senescence burden[2,73]. These p16[+]/p21[+] regions overlapped with fibrotic lesions and high MMP-7[+] staining. Quantitatively, senescence marker especially p21, fibrosis markers (collagen, α-SMA) and MMP-7 levels were also positively correlated (Supplementary Fig. 33). Together, these findings show that bleomycin-treated mice developed fibrosis accompanied by senescence induction and elevated MMP-7 expression.

At the endpoint, mice received ALBANC via i.v. Urine collected 2 h later showed a 5.6-fold higher alloy assay signal in bleomycin-treated vs untreated mice (Fig. 7e,f). ROC curve analysis (Fig. 7g) confirmed high nanoprobe sensitivity and specificity, with an AUC of 0.859 (95% CI, 0.39–0.97). On the other hand, the peroxidase assay showed a 4.4-fold higher signal in bleomycin-treated mice, although changes were not visible by eye (Extended Data Fig. 10a,b). ROC analysis also demonstrated accurate disease tracking with an AUC of 0.891 (95% CI, 0.721–1) (Extended Data Fig. 10c). Urinary signals from both the alloy formation and peroxidase assays showed a positive correlation with lung fibrotic markers, senescence markers, as well as with MMP-7 levels in the lungs and serum of fibrotic mice (Fig. 7h,i and Extended Data Fig. 10d-i), consistent with MMP-7-dependent cleavage contributing to nanoprobe cleavage. In contrast, untreated mice showed low/near-background signals, underscoring the specificity of the nanoprobe to MMP-7 activity associated with fibrotic phenotypes. Together, these results support ALBANC for noninvasive fibrosis monitoring via urinary readout of MMP-7 activity.

Next, we tested whether ALBANC coupled with the alloy assay can detect early/incipient fibrotic changes and associated senescence. C57BL/6 mice received intratracheal bleomycin for 7 days to induce incipient fibrosis, with untreated controls (Fig. 7a). At day 7, micro-CT showed parenchymal thickening in bleomycin-treated mice (Fig. 7b) but less severe than the 14-day bleomycin treatment. Histology confirmed signs of incipient fibrosis, with increased collagen (Masson's trichrome) and α-SMA[+] activated myofibroblasts in bleomycin-treated lungs (Fig. 7j,k). Senescence markers p16 and p21 were also elevated in regions of fibrotic tissue remodeling, with p21 enriched in α-SMA[+] areas, consistent with prior reports of p21 in early fibrosis[74].

At day 7, mice received ALBANC i.v., and urine was collected 2 h later. Alloy assays showed higher urinary signals spectroscopically in bleomycin-treated vs control mice (Fig. 7l), with ROC analysis demonstrating a good diagnostic performance (AUC = 0.90, 95% CI: 0.6809-1; Fig. 7m). Urinary signals correlated with α-SMA and p21 (Pearson's $r$ = 0.9268 and 0.8847, respectively; Fig. 7n,o), consistent with urinary readouts reflecting early remodeling and p21-associated changes. No significant correlation was observed with Masson's trichrome or p16. This is consistent with early fibrosis, where collagen deposition is less extensive and p21 can predominate over p16 in senescence-associated responses[74]. By comparison, the peroxidase assay did not distinguish the two treatment groups at day 7 (Supplementary Fig. 34). Together, these findings demonstrate that systemic ALBANC administration enables urinary AuNC readouts that spectroscopically track both early lung remodeling, as part of the fibrotic response, and established pulmonary fibrosis in mice. Furthermore, the enhanced sensitivity of the alloy assay improves the detection of early-stage fibrosis in the mouse model.

### Discussion

Despite its important role in cancer, detecting senescence, particularly in vivo, remains a challenge. Recent approaches use small-molecule or nanomaterial probes that either target lysosomal SA-β-gal[7] or exploit senescence-associated changes such as increased lysosomes and dysregulated endocytosis[75,76]. However, galactose-conjugated ON-OFF fluorescent SA-β-gal probes can have limited clinical translatability due to poor tissue light penetration and signal dispersion. In addition, SA-β-gal

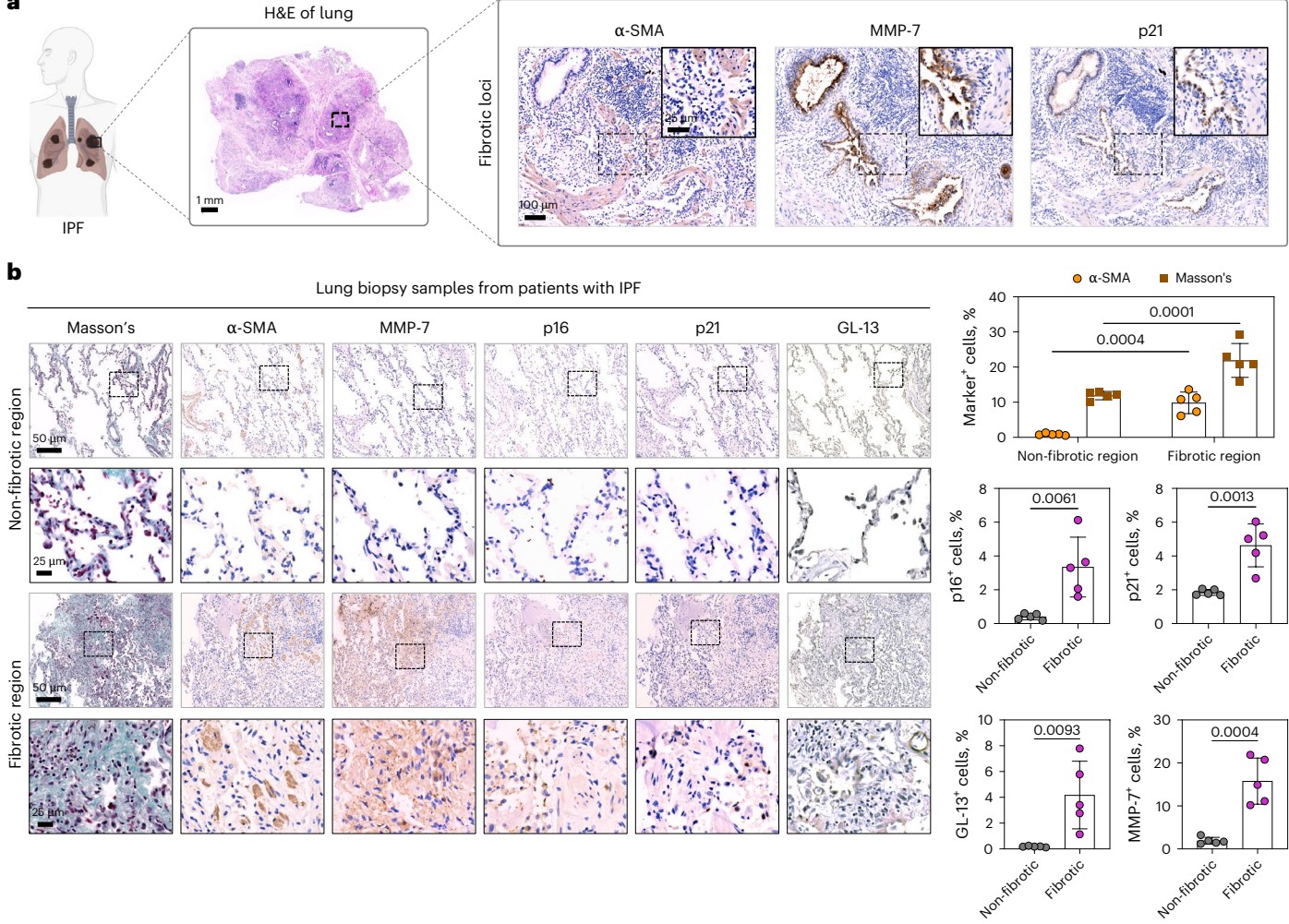

**Fig. 6 | MMP-7 levels are elevated in the lungs of patients with IPF.**
**a**, Representative histological images of lungs of IPF patients, stained for H&E, fibrosis marker (Masson's trichrome), senescence marker (p21) and MMP-7. Scale bar = 100 μm. Created in BioRender. Fruk, L. (2026) https://BioRender.com/whgh1jw. **b**, Representative histological images and staining quantification of lung biopsy samples from patients with IPF, analyzed for fibrosis markers (Masson's and α-SMA), senescence markers (p16, p21 and GL-13), and MMP-7. Scale bars = 50 μm or 25 μm. Non-fibrotic or background lung region is defined as those with <5% α-SMA⁺ cells and <15% Masson's⁺ total cells in the area. Fibrotic region is defined as those with >5% α-SMA⁺ cells and >15% Masson's⁺ total. Data are presented as mean ± s.d.; two-way ANOVA with Šidák multiple comparisons and unpaired two-tailed t-test were used for statistical analysis. Here, $n = 2$ independent patients with IPF (biological replicates); data points represent $n = 2$ (for patient 1) or 3 (for patient 2) independent areas in the lungs from each patient sample.

activity can be high in non-senescent cells (for example, macrophages and osteoclasts), limiting overall specificity[77–80]. Moreover, SA-β-gal elevation is shared across multiple senescence programs (oncogene, DNA damage and replicative senescence), complicating context-dependent assessment[7]. This limitation is particularly challenging in older individuals with heterogeneous burdens of aging-associated senescent cells, including populations with physiological roles. We previously developed NanoJAGGs, organic nanoprobes that enable in vivo photoacoustic assessment of senescence burden by leveraging increased lysosomal content and endocytosis[75]. However, NanoJAGGs cannot distinguish chemotherapy-induced from aging-associated senescence and lack tissue specificity.

Given these limitations, leveraging tissue- and inducer-specific SASP profiles offers a route to context-dependent senescence detection. This heterogeneity can enable detection and characterization of tissue-specific senescent cells based on secretory signatures. With a focus on lung disease, we sought SASP enzymes that could be leveraged for in vivo sensing. We first identified MMP-7 as a biomarker of chemotherapy-induced senescence in lung adenocarcinoma. We then

designed a three-component nanoprobe for noninvasive in vivo detection of MMP-7 via size-dependent renal clearance. Several lines of evidence from our study support MMP-7 as preferentially associated with TIS in lung tumors compared with age-associated senescence in our models. In an orthotopic model, cisplatin increased canonical senescence markers (p16/p21), reduced proliferation (Ki-67) and coincided with increased tumor MMP-7. This validates our xenograft observations in a physiologically relevant setting. Second, in naturally aged mice, whereas p16 level increased across organs, MMP-7 level did not differ from young controls. Third, in a model of damaged-induced senescence in healthy mice exposed to cisplatin in the absence of tumors, p21 level in the lungs increased consistently with a senescence response, yet MMP-7 levels in lung, kidney, and heart were not significantly altered. Together, these data suggest MMP-7 is not broadly elevated by aging or chemotherapy alone, but is increased in chemotherapy-treated lung tumors in our models.

We probed MMP-7 specificity of the nanoprobe by ablating MMP-7 in A549 cells. In MMP-7⁻/⁻ A549 cells, senescence did not increase intracellular/secreted MMP-7 and its senescent conditioned media did

not cleave the nanoprobe, unlike the WT A549 counterpart. These results support that nanoprobe activation is largely driven by MMP-7 activity in these settings, rather than off-target proteases or other SASP components. An essential component of our nanoprobe is the human serum albumin protein carrier, selected for biocompatibility, cost-effectiveness, and reported prolonged circulation[81]. Albumin was conjugated to 1.6-nm AuNCs via an MMP-7-cleavable peptide using click chemistry. MMP-7 cleavage (of the peptide linker) releases AuNCs that are renally cleared (below the glomerular filtration cutoff). Although AuNCs possess intrinsic fluorescence, fluorescence-based readout is insufficiently sensitive in urine. Instead, we initially used AuNC peroxidase-like activity to oxidize colorless TMB substrate into a blue-colored oxidized product in the presence of $H_2O_2$. However, this approach may be affected by interfering species present in urine, which can compromise sensitivity. To enhance the detection strategy, we developed an assay based on the ability of AuNC to act as seeds for the growth of larger Au-Ag alloys with a high extinction coefficient, resulting in 250-fold improvement in detection sensitivity compared with the peroxidase assay. Alloy formation completes in ~20 min with low concentrations of affordable reagents including $AgNO_3$, ascorbic acid, and CTAC. The size of the AuNC-Ag particles influences the optical absorption profile. Importantly, we used a standardized endpoint assay format and calibrated each batch using known AuNC concentrations. The initial AuNCs are monodisperse, and under the growth conditions the resulting alloy nanoparticles are 30.9 ± 7.8 nm in size. The alloy assay requires urine buffer exchange to reduce salt effects on growth, which can be done by routine centrifugation in a laboratory setting (in urine and blood analyses).

Biodistribution in healthy mice showed free AuNC renal excretion within 2 h, whereas uncleaved ALBANC accumulated in liver/spleen and was near background by day 15. No overt toxicity was observed over the study period, consistent with tolerability in these conditions. We then evaluated the nanoprobe in chemotherapy-induced senescence and pulmonary fibrosis mouse models. Fibrosis model was included because MMP-7 is known to be elevated in lung fibrosis. It is worth noting that the negligible urinary signals in the control groups in the two disease models (that is healthy, non-fibrotic mice and untreated xenograft mice with a low senescence burden) demonstrated the stability of the nanoprobe in vivo. That is, the nanoprobe was not significantly cleaved when MMP-7 level was relatively low.

SASP composition was lineage- and context-dependent. Senescence induction in melanoma, prostate, and breast cancer cells yielded distinct protease signatures (for example, MMP-13, cathepsin A and MMP-3) rather than uniform MMP-7 upregulation. These findings support MMP-7 as a lung cancer-associated TIS marker in our mouse models and human samples, and suggest protease-based diagnostics may need tailoring by lineage and treatment. Accordingly, ALBANC is modular and can be reconfigured to target other proteases identified (that is by changing the peptide linkage sequence) by SASP profiling or public transcriptomic data from drug-treated tumors, thereby enabling indication-specific detection of senescence. For example, our analyses of available scRNA-seq datasets from LUAD patients treated with neoadjuvant platinum-based chemotherapy revealed that elevated *MMP7* expression correlates with upregulation of canonical senescence markers (for example, *CDKN1A*, *CDKN2A*), and enrichment of senescence-associated transcriptional programs. Such orthogonal readouts could inform rational design of urinary nanosensors for context-specific senescence detection.

We show that ALBANCs with the alloy assay can detect chemotherapy-induced senescence and lung fibrosis in our models. Furthermore, urinary readouts also decreased with senolytic treatment, supporting use for treatment monitoring. This finding is particularly relevant given the number of senolytic agents currently in development and clinical trials, as well as the lack of effective tools to evaluate their efficacy. ALBANC readouts could then be paired with senescence-clearance strategies to inform dosing and response assessment. Additionally, across studies, the alloy assay produced visible urinary color changes due to better sensitivity, whereas the peroxidase assay required spectroscopy. Both methods distinguished higher versus lower senescence/fibrosis, and urinary signals correlated with

**Fig. 7 | Urinary detection of bleomycin-induced lung fibrosis. a**, Schematic representation of bleomycin-induced pulmonary fibrosis model. Briefly, C57BL/6 mice were either intratracheally administered with bleomycin for 7 days (incipient fibrosis) or 14 days (established fibrosis) or left untreated. At the endpoint, mice were administered with nanoprobes i.v. Urine was collected 2 h p.i. of nanoprobes, and the renally cleared AuNC was detected using alloy formation assay. Created in BioRender. Fruk, L. (2026) https://BioRender. com/6j83ai8. **b**, Representative CT scans of lungs from untreated- versus bleomycin-treated mice. **c,d**, Representative histological images (**c**) and staining quantification (**d**) of lungs at the treatment endpoint, stained for fibrosis markers (Masson's trichrome and α-SMA), senescence markers (p21 and p16) and MMP-7, from untreated mice and mice after undergoing 14-day bleomycin treatment (n = 8 independent mice per group, mean ± s.e.m, unpaired two-tailed t-test). Scale bar = 25 μm. **e**, Photograph of the alloy formation assay on urine samples from untreated or bleomycin-treated mice injected with nanoprobes. **f**, Absorbance values ($A_{414\,nm}$) from alloy formation assay of urine samples collected from untreated or bleomycin-treated mice 2 h p.i. with nanoprobes (n = 8 mice per group, mean ± s.e.m, unpaired two-tailed t-test). **g**, ROC analysis showing the diagnostic specificity and sensitivity of nanoprobes and alloy formation assay in detecting established fibrosis between bleomycin-treated (14 days) and untreated mice group. The solid line represents the ROC curve for distinguishing untreated controls (n = 8 independent mice) from bleomycin-treated patients (n = 8 independent mice). The dashed diagonal line represents the performance of a random classifier (AUC = 0.5). The area under the curve (AUC) was 0.86 (standard error = 0.1019; 95% confidence interval 0.6597–1.000; P = 0.0157). **h,i**, Correlation analysis between the percentage of intensity of signals from urine samples detected using the alloy formation assay with (**h**) fibrosis markers, Masson's⁺ area (r = 0.7485; P = 0.0327) and α-SMA⁺ cells (r = 0.701; P = 0.0321), and (**i**) senescence markers, p16 (r = 0.3910; P = 0.3382) and p21 (r = 0.7904; P = 0.0196), in the lungs of 14-day bleomycin-treated mice.

For (**h**) and (**i**), solid lines indicate linear regression fits (least-squares fit; estimated mean relationship between variables), and dashed lines denote the 95% confidence intervals of the regression line; n = 8 independent mice per group, two-tailed Pearson correlation; Pearson correlation coefficients (r) and corresponding one-tailed P values are listed. **j,k**, Representative histological images (**j**) and staining quantification (**k**) of lungs at the treatment endpoint, stained for fibrosis markers (Masson's trichrome and α-SMA), senescence markers (p21 and p16) and MMP-7, from untreated mice and mice after undergoing 7-day bleomycin treatment (n = 4 individual mice (untreated) and n = 5 individual mice (7-day bleomycin) (biological replicates); mean ± s.e.m, unpaired two-tailed t-test). Scale bar = 25 μm. **l**, Absorbance values ($A_{414\,nm}$) from alloy formation assay of urine samples collected from untreated or bleomycin-treated (7 days) mice 2 h p.i. with nanoprobes (mean ± s.d., n = 4 individual mice (untreated) and n = 5 individual mice (7-day bleomycin) (biological replicates), mean ± s.e.m, unpaired two-tailed t-test). **m**, ROC analysis showing the diagnostic specificity and sensitivity of nanoprobes and alloy formation assay in detecting incipient fibrosis between bleomycin-treated (7 days) and untreated mice groups. The solid line represents the ROC curve for distinguishing untreated controls (n = 4 independent mice) from bleomycin-treated mice (n = 5 independent mice). The dashed diagonal line represents the performance of a random classifier (AUC = 0.5). The area under the curve (AUC) was 0.90 (standard error = 0.118; 95% confidence interval 0.6809 – 1; P = 0.0432). **n,o**, Correlation analysis between the percentage of intensity of signals from urine samples detected using the alloy formation assay with (**n**) fibrosis markers, Masson's⁺ area (r = −0.3905; P = 0.5158) and α-SMA⁺ cells (r = 0.9268; P = 0.0235), and (**o**) senescence markers, p16 (r = 0.5264; P = 0.3622) and p21 (r = 0.8847; P = 0.0462), in the lungs of 7-day bleomycin-treated mice. For (**n**) and (**o**), solid lines indicate linear regression fits (least-squares fit; estimated mean relationship between variables), and dashed lines denote the 95% confidence intervals of the regression line; n = 5 individual mice (7-day bleomycin); two-tailed Pearson correlation.

histological markers. Importantly, the enhanced analytical sensitivity of the alloy assay enabled detection of senescence and fibrosis at earlier stages (for example, day 7 post-bleomycin), where conventional peroxidase assays failed to discriminate treated from control groups. It also captured reduced senescence-associated signals after senolytic treatment in tumors. Together, these results are consistent with improved in vivo performance of the alloy assay, particularly at low or early disease burden. This study also underscores the potential of ALBANC nanoprobe for early detection applications.

We introduce an in vivo approach for urinary senescence detection that leverages SASP-associated protease activity. ALBANC provides a noninvasive readout of TIS-associated protease activity that

may enable longitudinal monitoring during therapy. Such readouts could support treatment monitoring and may inform decisions such as additional evaluation for relapse risk or senolytic co-treatment. Because readout is urine-based and colorimetric/spectroscopic, the approach could be compatible with standard laboratory workflows. In this context, targeted senescence-clearance strategies with improved therapeutic windows could be paired with ALBANC readouts to help guide dosing and timing in combination strategies.

This study has limitations. MMP-7 was selected based on its upregulation in chemotherapy-induced senescence in lung cells, but MMP-7 can also be implicated in various cancers[82,83] and non-cancerous conditions such as infections and inflammation[84,85]. In our mouse

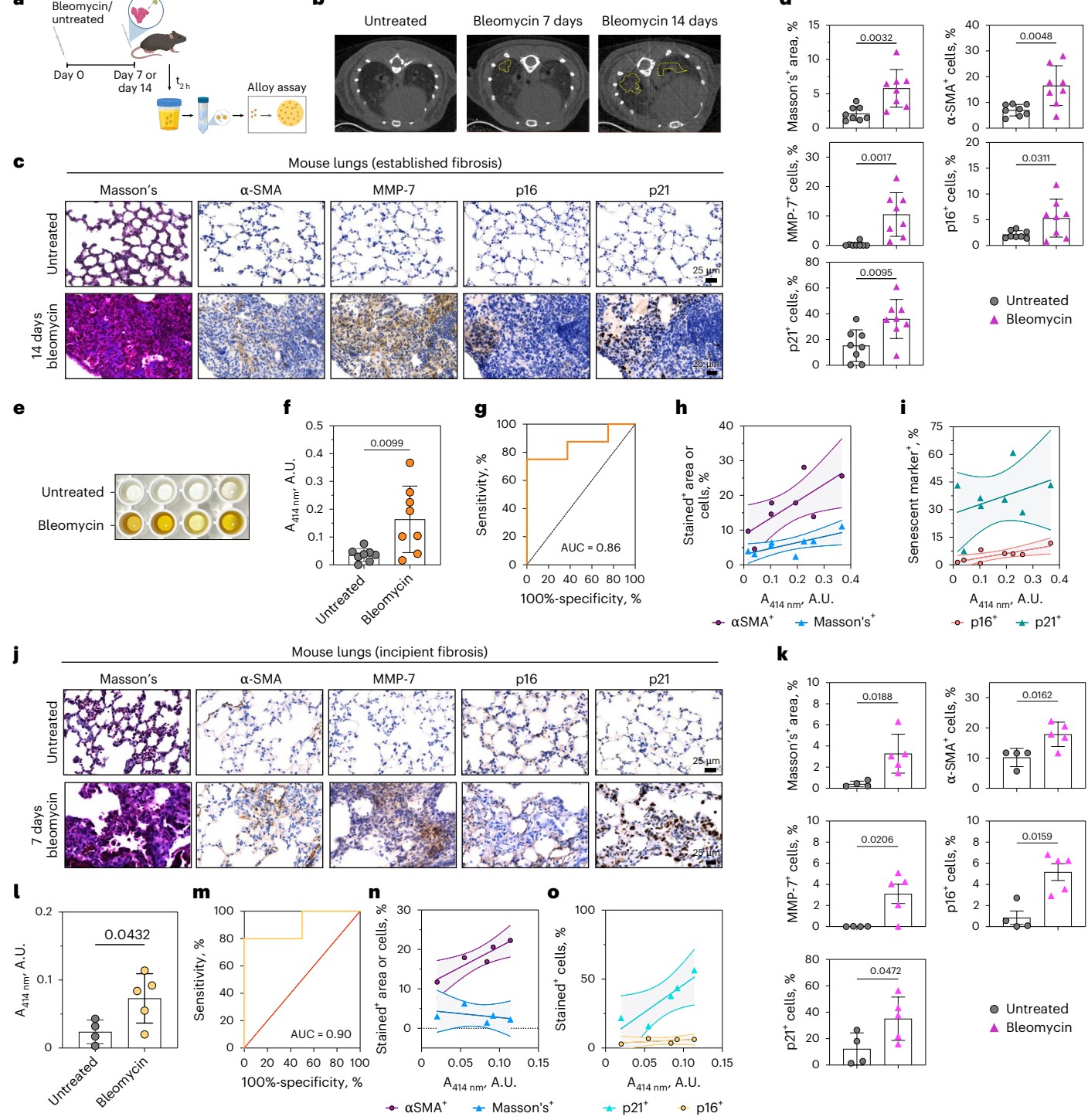

models, healthy mice and untreated xenografts with low senescence burden showed background MMP-7 in plasma and tissues. Despite this, it remains a valuable biomarker for senescence and fibrosis in this context, as urinary signals in non-senescent and healthy mice were significantly lower (and resulted in non-visible color change in the test) compared to the urinary signals in cisplatin-induced senescent and bleomycin-induced fibrotic mice due to the significantly higher levels of MMP-7. To further establish MMP-7 as a robust and specific biomarker of chemotherapy-induced senescence, several key areas warrant investigation. Comparisons across a broader spectrum of senescent and non-senescent pathologies will be required to define biomarker specificity. Longitudinal mapping is needed to resolve how MMP-7 levels correlate with the onset, progression, and resolution of senescence over time. In parallel, mechanistic studies could clarify the cellular sources and regulatory drivers of MMP-7 under chemotherapeutic stress, distinguishing senescence-associated expression from inflammatory or tumor-intrinsic signals. Finally, integrated cross-tissue transcriptomic and proteomic profiling of drug-treated tumors and their microenvironments will enable benchmarking of MMP-7 against other SASP-associated proteases, clarifying contexts in MMP-7 alone is sufficient and those that necessitate multiplexed detection to define a robust, context-aware senescence signature.

Looking ahead, the platform's modular design could allow reconfiguration for simultaneous detection of multiple proteases, thereby enhancing specificity and extending applicability to diseases driven by ECM remodeling, where early and accurate diagnosis is essential for effective intervention. The alloy amplification assay improves sensitivity versus peroxidase-based urinary readouts, but currently requires buffer exchange, which may limit point-of-care use. Future work could integrate amplification into lateral-flow formats for point-of-care or low-resource settings. Additionally, long-term cytotoxicity and biocompatibility will be required to support translational deployment of the ALBANC nanoprobe across diseases characterized by elevated protease activity. Together with cross-tissue transcriptomic analyses of drug-treated tumors to benchmark MMP-7 against other SASP proteases, these efforts will help define contexts in which MMP-7 alone is sufficient and those that require multiplexed detection to establish a robust, context-aware senescence signature.

## Methods

### Ethics declaration
Research complies with all relevant ethical regulations. The use of human tissues was approved by ethics review committees at Royal Papworth Hospital Research Tissue Bank (RPHRTB) Project Number T02722 for patients with lung adenocarcinoma, and Project Numbers T02147 and T02259 for patients with idiopathic pulmonary fibrosis. Written informed consents were obtained from all patients.

All animal experiments were approved for ethical conduct by the Home Office England and Central Biomedical Services, performed under PPL holder numbers P7EC604EE and PP7061972 and regulated under the Animals (Scientific Procedures) Act 1986, as stated in the International Guiding Principles for Biomedical Research involving Animals.

### Human biopsies
Human lung adenocarcinoma samples were obtained from the RPHRTB after review by the RPHRTB project review committee (project number T02722). Human idiopathic pulmonary fibrosis samples were obtained from the RPHRTB after review by the RPHRTB project review committee (project numbers T02147 and T02259). RPHRTB has a derogation under the UK Human Tissue Authority (HTA) to supply samples (HTA number 12212) that are surplus to therapeutic necessity and were acquired with Research Ethics Committee-approved RPHRTB permission. Patients signed the RPHRTB general consent form, approving the use of their biopsies for research purposes and sample transfer was covered by a valid Material Transfer Agreement. Written informed consent was obtained for all tissue samples using Papworth Hospital Research Tissue Bank's ethical approval (East of England - Cambridge East Research Ethics Committee). Further clinical information on the lung adenocarcinoma and IPF patient samples is available in Supplementary Tables 1 and 2.

### Cell cultures
Human pulmonary adenocarcinoma A549 (catalog no. CCL-185), MDA-MB-231 (catalog. no. HTB-26), and PC-3 (catalog no. CRL-1435) cell lines were obtained from the American Type Culture Collection (ATCC); SK-Mel-103 cell line was obtained from Sigma-Aldrich (catalog no. SCC439). These cell lines were cultured in Dulbecco's modified Eagle's medium (Gibco, 11965084) enriched with 10% fetal bovine serum. Human pulmonary fibroblasts-adult (HPF-a) were purchased from ScienceCell (catalog. no. 3310) and cultured in complete Fibroblast Medium (ScienceCell, 2301). Heterozygous KRas$^{G12D/WT}$ murine L1475 cell line, generated by C.P. Martins from KRas$^{LSL-12D/WT}$; p53$^{\{-/-\}}$ (KP) mice[86], was cultured in Dulbecco's modified Eagle's medium, enriched with 10% fetal bovine serum. All cells were cultured under a controlled environment of 37 °C and 5% $CO_2$ concentration and regularly screened for mycoplasma contamination using the universal Mycoplasma Detection Kit (ATCC, 30-1012 K). All cell lines were tested negative for mycoplasma contamination.

### Induction of cellular senescence in vitro
To generate in vitro chemotherapy-induced senescence model, dose-dependent experiments for four different drugs were performed to ascertain the optimal sub-lethal drug concentrations that would induce cell senescence with minimum cell death. A549 cells underwent treatment with 15 μM cisplatin, 100 nM docetaxel, 5 nM pemetrexed or 15 μM palbociclib for 10 days and were used for the described experimental objectives promptly after the drugs were removed. L1475 cells were subjected to treatment with 5 μM of cisplatin for 5 days. Lung fibroblast cells (HPF-a) were treated with 5 μM of cisplatin for 7 days. SK-MEL-103 were treated with 15 μM of palbociclib over a 7-day period. MDA-MB-231 cells were treated with 5 μM of palbociclib for 7 days. PC-3 were treated with 14 nM of docetaxel for 7 days.

### Histology
Samples for histological examination were carefully collected, preserved, dried, and prepared for paraffin embedding and slicing. Cryo-sectioning was performed, and the resulting slides were then stored at −80 °C until used for SA-β-gal staining. Meanwhile, immunohistochemistry (IHC) staining was conducted on the 5- to 7-μm paraffin sections. The tissue sections were positioned on Superfrost plus slides and allowed to dry overnight. IHC was facilitated through an automatic immunostaining platform (Autostainer Link by Dako and Bond by Leica). Antigen retrieval was the initial step and was performed using Tris-EDTA buffer at pH 9, followed by the blockage of innate peroxidase using 3% $H_2O_2$. The slides were subsequently subjected to the relevant primary antibodies. At post-primary antibody application, the slides were treated with matching secondary antibodies and necessary visualization systems (Bond Polymer Refine Detection, Bond, Leica; EnVision FLEX + , Dako), each of which was conjugated with horseradish peroxidase. The immunohistochemical reaction was enabled using 3,3′-diaminobenzidine tetrahydrochloride (Dako). The final stages involved dehydration, clarification, and application of a permanent mounting medium to prepare the slides for a microscopic study. Finally, the complete slides were scanned with an AxioScan Z1 from Zeiss, and images were captured with the Zen Blue Software from Zeiss (v2.6). Supplementary Table 3 provides a list of the antibodies used and the methods employed for antigen retrieval. All procedures were performed at the Early Cancer Institute, University of Cambridge.

## Generation of conditioned media

Conditioned media (CM) from senescent and non-senescent cells were generated as follows. Senescent cells were plated at 70% to 80% confluence on a 10 cm dish, whereas non-senescent cells were plated at 40% to 50% confluence in a complete medium. The following day, the adhered cells were rinsed twice with pre-warmed PBS (10 ml) to ensure no proteases from the serum used for culturing the cells were left. Then, as much as 8 ml fresh serum-free DMEM was added to the dish and left to condition for 24 h. After this period, the CM was gathered and transferred to Falcon tubes before being centrifuged for 10 min at 102 $g$ at 4 °C. The supernatant was then carefully transferred to new Falcon tubes and underwent another centrifugation for 10 min at 1,125 $g$ at 4 °C to remove any cell debris. This conditioned medium was immediately stored at −80 °C until it was used for analysis.

## Human protease array and human protease inhibitors array

The CM derived from senescent and non-senescent cells were analyzed for the relative levels of secreted proteases present using the Proteome Profiler Human Protease Array Kit (R&D Systems, ARY021B) and Proteome Profiler Human Protease Inhibitor Array Kit (R&D Systems, ARY023), as per the instructions provided by the manufacturer. A total of 1,000 µl CM was combined with 500 µl of blocking buffer supplied by the kit. The membranes were imaged with a ChemiDoc imager (Bio-Rad, v6.1) at different exposure times (from 30 s to 2 min). Following the subtraction of background noise, pixel density was quantified using the Image-J Software (v1.53k). Because the number of proteins secreted by the cells depends on the number of cells, the measured levels of secreted proteases need to be adjusted by cell numbers, which was set to be between 150,000 to 200,000 cells.

## A549 xenograft mice experiments

A549 cell suspension, prepared by combining Cultrex Basement Membrane Extract (Type 3, RnD Systems, 3632-010-02) in a 1:1 ratio with 3 million A549 cells, was injected subcutaneously into each flank of athymic nude mice (Crl:NU(NCr)-Foxn1nu, 10 weeks old, Charles River Laboratories). Mice were weighed twice a week and tumor volume (V) was longitudinally assessed via a caliper, using the formula: Volume = $(D \times d^2)/2$, where $D$ represents the longest diameter of the tumor, and $d$ indicates the shortest diameter (experiments were terminated at either pre-established time endpoints or when tumors reached an average diameter of 1.5 cm; maximal tumor size/burden was not exceeded at the established endpoints). Chemotherapy treatment started once the average tumor volume exceeded 100 mm³ (20 days after tumor injection). Immediately before the treatment started, about 100 µl blood from each mouse was collected. Five mice were treated with cisplatin (1 mg kg⁻¹ body weight), six mice were treated with pemetrexed (100 mg kg⁻¹ body weight) and five other mice were treated with saline as vehicle. All treatment was administered via intraperitoneal injection. Mice were treated with 6 doses of chemotherapy: thrice a week (on Monday, Wednesday and Friday) for 2 weeks. On the following day after the final treatment was administered, the mice were sacrificed at the same time. From each mouse, blood, both tumors, heart, lungs, kidney and liver were collected for analysis. Blood was collected into anticoagulating EDTA-treated tubes (Sarstedt, 16.444.100).

For nanoprobe studies, immediately before the treatment started, about 100 µl of blood from each mouse was collected. Six mice were treated with cisplatin (1 mg kg⁻¹ body weight), five mice were co-treated with cisplatin (1 mg/kg body weight) and ABT-737 (25 mg kg⁻¹ body weight), five mice were treated with ABT-737, and five other mice were treated with saline as vehicle. All treatment was administered via intraperitoneal injection. Mice were treated with 6 doses of chemotherapy or vehicle: three times a week (on Monday, Wednesday and Friday) for 2 weeks. Treatment with ABT-737 was given three times a week (alternating days with cisplatin). The day after the final treatment was administered, nanoprobes (15 µM, 200 µl) were injected to mice intravenously.

Urine (50–200 µl) was collected from mice 2 h after injection of the nanoprobes. The urine samples were then analyzed immediately using both peroxidase and alloy formation assay. Mice were then sacrificed, and the organs and blood were collected for analysis. Mice were then culled at the same time. From each mouse, blood, both tumors, heart, lungs, kidneys, spleen and liver were collected for analysis.

## Micro-CT imaging

For micro-CT imaging, mice were anesthetized with isoflurane and scanned on a microPET-CT system (Mediso Medical) using Nucline Nanoscan software (v2.0). Imaging was performed with an X-ray energy of 35 kVp, 450 ms exposure and 720 projections acquired in a semi-circular single field-of-view mode (one projection per step). Data were reconstructed with a Butterworth filter at high resolution (small voxel size).

## Bleomycin-induced pulmonary fibrosis

Female C57BL/6 mice, 11–12 weeks old (Charles River Laboratories), weighing 20 to 25 g, were used. Bleomycin sulphate (Scientific Laboratory Supplies, B8416-15UN) was dissolved in sterile 0.9% saline and given as a single dose of 0.5 U per animal (65 µl). Eight mice received intratracheally (i.t.) instillations of bleomycin on day 0 as previously described[87]. The control group (8 mice) did not receive any treatment. Right before the treatment started, about 100 µl blood from each mouse was collected. On day 14, following i.t. bleomycin administration, nanoprobes (15 µM, 200 µl) were injected into both treated and untreated groups intravenously. Urine (50–200 µl) was collected from mice 2 h after injection of the nanoprobes. The urine samples were then analyzed immediately using both peroxidase and alloy formation assay. Mice were then sacrificed, and the major organs and blood were collected for analysis. A similar protocol was followed for the early or incipient fibrosis experiment, with the only difference being the duration of bleomycin treatment, which was 7 days instead of 14.

## Serum extraction from blood samples

Blood samples were collected into tubes with anticoagulants (EDTA-treated). By spinning the tubes in a refrigerated centrifuge (4 °C) for 10 min at 2,500 $g$, cells and debris were separated from the serum as pellets. The serum was transferred into a new tube. An extended spin of 10 min at 16,200 $g$ further removes platelets from the serum. The clear fluid that remains on top after centrifugation (serum) was transferred into a new microcentrifuge tube using a pipette. Samples were kept at a temperature range of 2 °C to 8 °C during processing. Serum samples were aliquoted and stored at −80 °C until further use or analysis.

## AuNC synthesis

Preparation of peptide-functionalized AuNCs was modified from a published protocol[88]. In general, into a fresh solution of gold (III) chloride trihydrate solution (HAuCl₄, 20 mM, 400 µl) with deionized water (2,533 µl) in a 12-ml glass vial was quickly added reduced L-GSH (GSH, 15 mM, 667 µl) and peptide, azidoacetyl-KGRPLALWRSGGGC or azidoacetyl-KGGGGGGGC (5 mM, 400 µl). The mixture was left to stir at room temperature for 5 min, followed by heating at 70 °C while stirring (500 rpm) for 24 h. The resulting AuNCs were purified and washed with PBS four times using centrifugal ultrafiltration (Amicon Ultra centrifugal filter units Ultra-15, 10 kDa, Sigma-Aldrich, 3,195 $g$, 10 min) whereby the AuNC remains in the retentate. The synthesis produced AuNCs with a 77–85% yield based on Au content. The purified AuNCs are stored in PBS with a concentration of 0.3 mM AuNC. The suspension remained stable colloidally when stored at 4 °C for at least 6 months with negligible changes in optical absorbance and hydrodynamic size.

## Estimation of elemental content in AuNCs

The molecular weight of the AuNCs was determined using liquid chromatography-mass spectrometry (LC-MS). The content of ligands

(GSH and peptides) attached to the AuNCs was estimated by measuring the percentage of sulfur (S) and gold (Au) using ICP-MS analysis. To quantify the number of biotinylated ligands on each AuNC, the biotin concentration in the filtrate after purification was measured and subtracted from the initial concentration of the biotinylated peptide used. The amount of biotin in the filtrate was determined using the Pierce Biotin Quan titration Kit (28005), following the manufacturer's protocol (Thermo Fisher Scientific). Because the sulfur content from ICP reflects the total amount of GSH and peptides attached, the number of GSH molecules could then be estimated.

### Functionalization of human serum albumin

Human serum albumin (3.4 mg, Merck, 126658) was reconstituted in 1360 µl PBS for 15 to 20 min. As much as 60 µl of the linker DBCO-PEG(4)-NHS ester (34.4 mM in anhydrous DMSO by Apollo Scientific) was added to albumin. The solution was left to shake at 500 rpm for 2 h at room temperature. Purification of functionalized albumin was done by centrifugal ultrafiltration using Amicon Ultra centrifugal filter units (30 kDa, 3,195 $g$, 10 min, Sigma-Aldrich) by washing the solution with PBS four times. A final solution of 500 µl in the retentate containing DBCO-functionalized albumin was collected and stored at 4 °C.

### Nanoprobe synthesis

ALBANC nanoprobe was assembled via DBCO-azide click chemistry between azidopeptide-functionalized AuNC and DBCO-functionalized albumin. In a standard conjugation procedure, 93 µl DBCO-functionalized albumin (concentration 7.62 mg ml$^{-1}$ or 0.10 mM in PBS) was mixed with 211 µl of AuNC (concentration 297 µM). This mixture was then incubated for 20 h at 37 °C with gentle agitation (450 rpm). To separate unbound AuNCs from the AuNC–albumin complexes (ALBANC nanoprobe), centrifugal ultrafiltration was employed using Amicon Ultra centrifugal filter units (Ultra-15, 30 kDa, Sigma-Aldrich) by washing with PBS four times (3,195 $g$, 10 min/wash). The nanoprobe was retained in the concentrate, whereas the unbound AuNCs passed into the filtrate. Post-ultrafiltration, the concentrated nanoprobes were suspended again in PBS (with a final concentration of 120 µM by [AuNC]) and then passed through a sterile filter (Millex-GV Filter, Millipore, pore size 0.22 µm). The nanoprobe was stored in 4 °C. A similar protocol was used to prepare the non-cleavable (control) nanoprobe.

### Characterization

Zeta potential and DLS measurements were recorded using a Zetasizer Nano Range instrument (Malvern Analytical) with Zetasizer Software (v7.13). UV-Vis absorption spectra and fluorescence measurements were obtained with an CLARIOstar Plus plate reader (Tecan Life Sciences) with Spark Control software (v2.3), whereas Fourier-transform infrared spectroscopy (FT-IR) spectroscopy was performed using a Bruker Tensor 27 spectrometer with samples pressed into KBr pellets using a PerkinElmer Spectrum One FT-IR Spectrometer (v10). For the characterization using electron microscopy, specimens were placed onto copper grids coated with carbon (Electron Microscopy Sciences), followed by imaging with a Talos F200X G2 electron microscope (Thermo Scientific) at an operating voltage of 200 kV with Thermo Scientific Velox Software (v2.9). To prepare the samples for TEM analysis, AuNC samples were first desalted using Amicon centrifugal filter (10 kDa, Sigma-Aldrich). After desalting, 5 µl of the sample was applied to the grid and left to dry overnight before imaging. For the nanoprobe, the carbon grids were first glow discharged. Nanoprobe samples (2.5 µl) were then applied onto the grid for 60 s and then blotted off to remove excess. Then, 2% uranyl acetate stain was applied for 40 s and then blotted off to remove excess. Samples were then left to dry for 1 h. LC-MS and MALDI were performed using a Waters' Xevo G2-S bench top QTOF with Waters Xevo G2-S QTOF

system with Waters MassLynx (v4.2 SCN971). ImageJ (v1.53k) was used for image analysis, including protease array pixel density quantification and TEM image analysis.

### ICP-MS analysis

AuNC samples, including urine samples from in vivo experiments, were first digested using aqua regia (Trace Metal Grade hydrochloric acid, Fisher Chemical and nitric acid, VWR) for 18 h. These samples were then diluted further into a matrix (4% hydrochloric acid and 4% nitric acid). The quantification of gold content in samples was conducted using an iCAP7400 Duo ICP spectrometer (Thermo Fisher Scientific) with Qtegra software (version 2.8.3170.392).

### Peroxidase assay

The assay was optimized to maximize the absorbance produced by AuNC- catalyzed oxidation of TMB with minimum background signal from self-oxidation of TMB. In general, peroxidase assay was performed by mixing AuNC (varying concentration, 25 µl in synthetic urine), 25 µl PBS and 100 µl TMB ELISA substrate solution (Thermo Fisher Scientific, 34028) in a 96-well plate (Corning). Absorbance at 652 nm was monitored for 15 min. For LoD determination, the absorbance at 652 nm was recorded at 10-s intervals over a 15-min period. A linear regression analysis was then performed to determine the initial reaction velocity ($A_{652 nm}$/s) during the initial 150 s. LoD was defined as the mean background signal plus 3 s.d.

### Alloy formation assays

The assay was optimized to maximize the absorbance produced by AuNC-Ag alloy formed with minimum background signal from silver nanoparticles that could also form during the assay. AuNC samples (either in synthetic urine or mice urine) were first washed with deionized water twice by ultracentrifugation (Amicon 10 kDa cutoff, Sigma-Aldrich). Freshly prepared aqueous solutions of AgNO$_3$ (2.5 mM, 25 µl) and CTAC (2.81 mM, 5 µl) were mixed with 60 µl AuNC samples (of varying concentrations) in a 96-well plate (Nunclon). Following this, to start the alloy formation, a freshly prepared aqueous solution of L-ascorbic acid sodium salt (1.25 mM, 100 µl) was added. The mixture was left to react for 15 min at 25 °C. At the endpoint of the assay, absorbance at 414 nm was recorded. For LoD tests, AuNC was first incubated in synthetic urine for 3 h at 37 °C. After washing with deionized water twice, varying concentrations of AuNC (60 µl) were mixed with 25 µl AgNO$_3$ (2.5 mM, 25 µl) and 5 µl CTAC (2.81 mM) in a 96-well plate. To start the assay, 100 µl freshly prepared L-ascorbic acid solution (1.25 mM) was added. Absorbance at 414 nm was read after 20 min. LoD was defined as the mean background signal plus 3 s.d.

### Pharmacokinetic and biodistribution studies

For pharmacokinetic studies, approximately 100 µl of blood was drawn intravenously from each female C57BL/6 mouse as a reference before injection. The mice were then intravenously injected with Alexa Fluor 750 (AF750)-labeled nanoprobes (15 µmol, 200 µl), and blood samples were taken at 30, 90, 480 and 1,440 min after injection. Nanoprobe concentrations in the blood were directly quantified using a fluorimeter. The quantification results were presented as a biexponential decay curve to estimate the blood half-life ($t_{1/2}$) values.

In the biodistribution studies, female C57BL/6 mice (10–11 weeks old) were administered AF750-labeled nanoprobes or AF750-labeled AuNCs (15 µM, 200 µl) intravenously. The mice were euthanized at 2 h (for both nanoprobes and AuNC) and 15 days (for nanoprobes only) post-injection. The accumulation of these substances in the main organs (liver, kidneys, lungs, spleen, and heart) was measured using the IVIS Spectrum Xenogen machine (Caliper Life Sciences) and quantified with Living Image Software (PerkinElmer, v4.7.3). Regions of interest were drawn around the organs, and the total radiance flux was subtracted (Total Flux [p/s]). Additionally, for mice injected with

AuNCs, urine samples were collected at the specified time points and analyzed using ICP-MS and colorimetric assays.

## Statistics and reproducibility

Statistical analyses were carried out using GraphPad Prism version 10. Data are presented as mean ± s.d. unless otherwise specified. Sample sizes were determined based on pilot data and previous publications of our group[26,89] estimating variance and effect size for each assay, in consultation with an experienced biostatistician. Group sizes were consistent with those commonly used in our previous research and were sufficient to detect biologically meaningful differences with statistical significance at $P < 0.05$. All key findings were reproduced in independent experiments to ensure robustness. Sample sizes for each experiment are provided in the corresponding figure legends. Group assignment was conducted using simple randomization procedures. Data normality was evaluated with the Shapiro-Wilk test, and homogeneity of variances was examined using the F-test. For datasets that followed a normal distribution with equal variances, significance was assessed using a two-tailed unpaired Student's t-test unless stated otherwise. When variances were unequal, Welch's correction was applied. Non-normally distributed data were analyzed using the two-sided Mann-Whitney U test. Comparisons among more than two groups were performed using one-way ANOVA, whereas two-way ANOVA was applied to datasets involving two independent variables, such as cell confluence or longitudinal tumor growth. Following ANOVA, Tukey's, Dunnett's or Šidák post hoc multiple comparison tests were used to evaluate differences between group means (as specified). Researchers conducting micro-CT and ICP analyses were blinded to group allocation during data processing. For mice study involving tumors, mice were matched with age and starting tumor volume. Due to the variability of in vivo tumor growth, animals were randomly allocated to experimental groups once initial tumor burden was determined, such that each experimental group were normalized by comparable tumor volumes before treatment. No samples, animals or data points were excluded under the predefined experimental criteria. Statistical significance was defined as $P < 0.05$.

## Reporting summary

Further information on research design is available in the Nature Portfolio Reporting Summary linked to this article.

# Data availability

All data associated with this study are present in this paper or Supplementary Information and are available from the corresponding authors upon request.

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

## Acknowledgements

We thank C. Martins (Cambridge MRC Cancer Centre) and E. Kerr (Queen's University Belfast) for providing lung cancer cell line L1475. We would like to thank H. Greer of the Electron Microscopy facility and Y. Hamied of the Department of Chemistry, University of Cambridge, for their support with TEM imaging. We thank S. Morsli (Early Cancer Institute) for his assistance with in vivo experiments. We would also like to thank N. Howard of the Microanalysis facility and A. Boodhun, R. Canales and D. Matak-Vinkovic of the Mass Spectrometry facility and Y. Hamied of the Department of Chemistry, University of Cambridge, for their continued support. The Fruk laboratory is supported by CRUK Early Detection Primer Award EDDPMA-May 23/100051 and EPSRC interdisciplinary research center (EPSRC IRC grant EP/S009000/1). M.H. was supported by the Gates Cambridge Scholarship. M.D. was funded by the CRUK Cambridge Centre Clinical Research Training Fellowship. M.G.E. and T.E. acknowledge funding from the EPSRC Centre for Doctoral Training in Sensor Technologies for a Healthy and Sustainable Future (EP/S023046/1). A.G.B. was supported by the W. D. Armstrong Studentship. The Munoz-Espin laboratory (including J.A.H.) is supported by a CRUK Program Foundation Award (C62187/A29760), a CRUK Early Detection OHSU Project Award (C62187/A26989) and a MRC New Investigator Research Grant (NIRG) (MR/R000530/1). D.M-E. was funded by the CRUK Cambridge Major Centre Award and University of Cambridge. J.G. is funded by Darley/Sands Downing College fellowship (G1099261). TEM was funded through the EPSRC Underpinning Multi-user Equipment Call (EP/P030467/1). R.C.R. is supported by the Cancer Research UK Cambridge Centre (CTRQQR-2021\100012) and the NIHR Cambridge Biomedical Research Centre (NIHR2033312). The views expressed are those of the authors and not necessarily those of the NIHR or the Department of Health and Social Care.

## Author contributions

L.F. and D.M.-E. supervised the project and secured funding. L.F., D.M.-E. and M.H. conceived the project and designed the experiments. M.H. performed most of the experiments and data analysis, with assistance from M.G.E. and A.G.B. J.G. and M.D. performed the in vivo experiments using murine models. J.A.H. performed computational analyses of scRNA-seq datasets. R.C.R. provided clinical samples of NSCLC and clinical oversight. L.F., M.H. and D.M.-E. wrote the manuscript. All authors read and commented on the manuscript.

## Competing interests

A.G.B., L.F. and D.M.-E. are involved in Senesys.Bio spin off company working on the development of a new generation of senolytics. The remaining authors declare no competing interests.

## Additional information

**Extended data** is available for this paper at https://doi.org/10.1038/s43587-026-01116-z.

**Correspondence and requests for materials** should be addressed to Ljiljana Fruk or Daniel Muñoz-Espín.

## (1) ALBANC nanoprobe administration

## (2) MMP-7-mediated cleavage

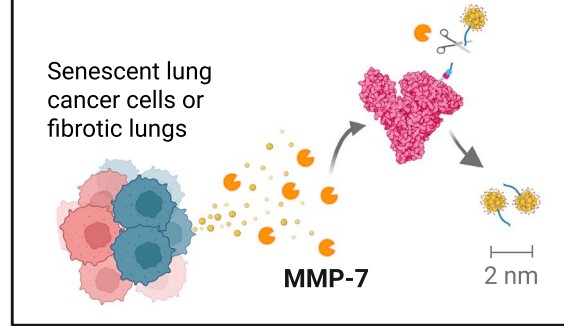

### Senescent lung cancer cells secrete MMP-7

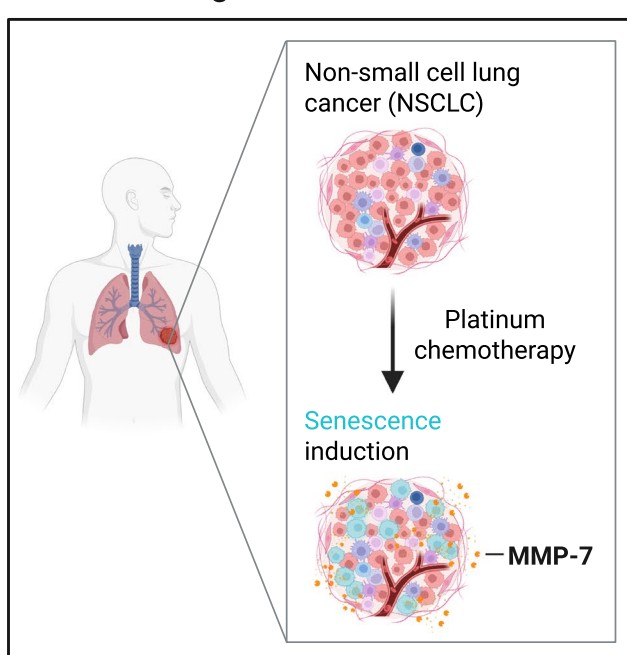

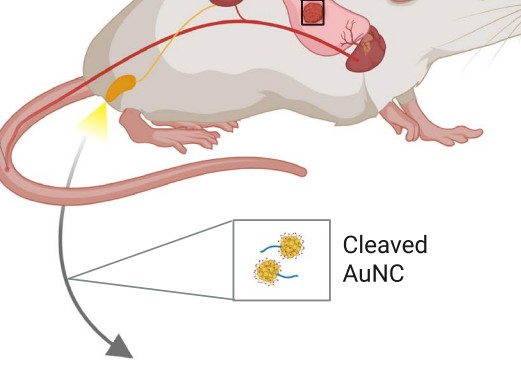

### (3) Colorimetric urine test

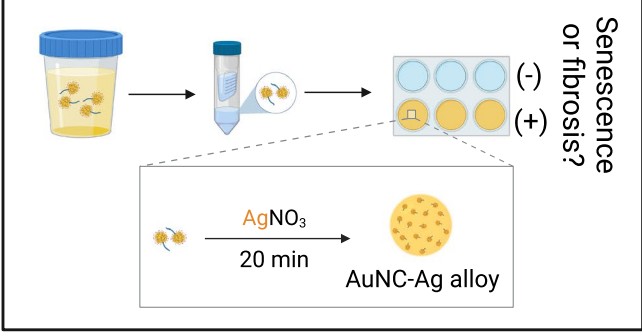

**Extended Data Fig. 1 | Design and mechanism of ALBANC (albumin-linked Au nanocluster) nanoprobes for *in vivo* detection of chemotherapy-induced senescence and pulmonary fibrosis.** ALBANC consists of human serum albumin, as a protein carrier, linked to gold nanoclusters (AuNC), as the urinary reporters, *via* MMP-7-cleavable peptides. After *in vivo* administration, cleavage of nanoprobes by MMP-7, secreted from therapy-induced senescent cells in lung tumor, or in the case of pulmonary fibrosis, from fibrotic tissues, releases AuNC that undergoes glomerular excretion through the kidneys and becomes concentrated in the urine. The secreted AuNC can be detected using simple assays, enabling colorimetric and spectroscopic detection of diseases in which MMP-7 level is elevated, including senescence and pulmonary fibrosis. Created in BioRender. Fruk, L. (2026) https://BioRender.com/5gfodlo.

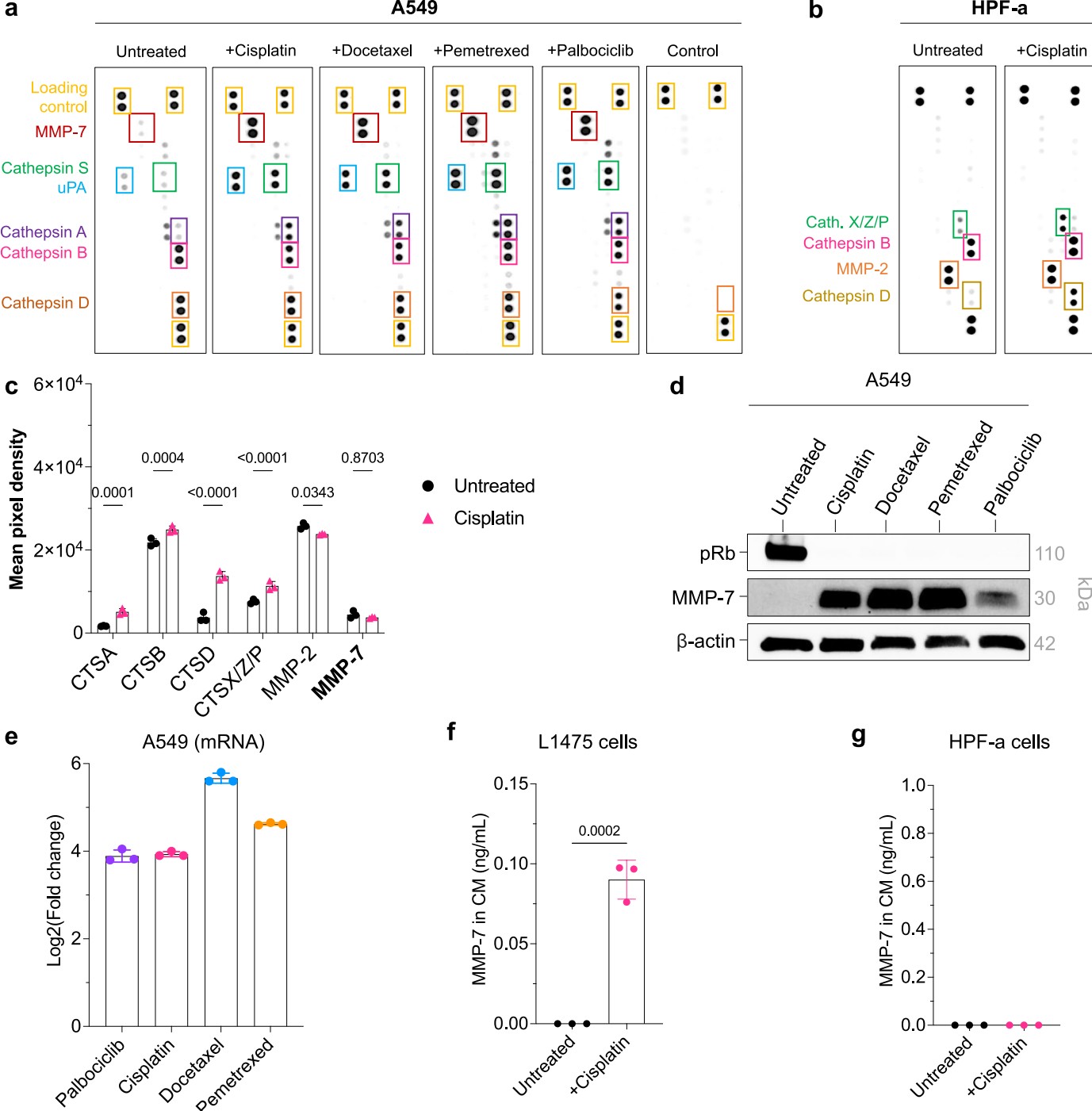

**Extended Data Fig. 2 | Senescent human and murine cancer cells abundantly secrete MMP-7 into the conditioned media, while fibroblast cells do not.** **a**. Photograph of representative uncropped human protease array panels of conditioned media samples from untreated and treated A549 cells, as well as only media without cells (control). All blots are exposed for 2 min. Experiments involved n = 4 independent biological replicates per group, corresponding to the plot in Fig. 1c. **b**. Photograph of representative uncropped human protease array panels of conditioned media samples from untreated and treated HPF-a cells. Both blots are exposed for 3 min. Experiments involved n = 3 independent biological replicates per group, corresponding to the plot in Extended Data Fig. 1c. **c**. Pixel intensity z-score of secreted proteases corresponding to (**b**) (n = 3 independent biological replicates per group, mean ± s.d., ordinary

two-way ANOVA with Šídák's multiple comparisons). **d**. Western blot analysis of pro-MMP-7 level (28 kDa) from lysates of senescent and non-senescent A549 cells (n = 3 independent biological replicates yielding similar results) **e**. Log2 fold change gene expression levels of *MMP7* in treated A549 cells relative to untreated cells (n = 3 independent biological replicates per group, mean ± s.d.). Concentration of MMP-7 in the conditioned media of untreated (non-senescent) and treated (senescent) (**f**) L1475 cells (n = 3 independent biological replicates per group, mean ± s.d.; unpaired two-tailed t-test) and (**g**) HPF-a cells (n = 3 independent biological replicates per group, mean ± s.d., statistical analysis not performed because both groups demonstrated identically undetectable levels of MMP-7). pRb: hyperphosphorylated Retinoblastoma.

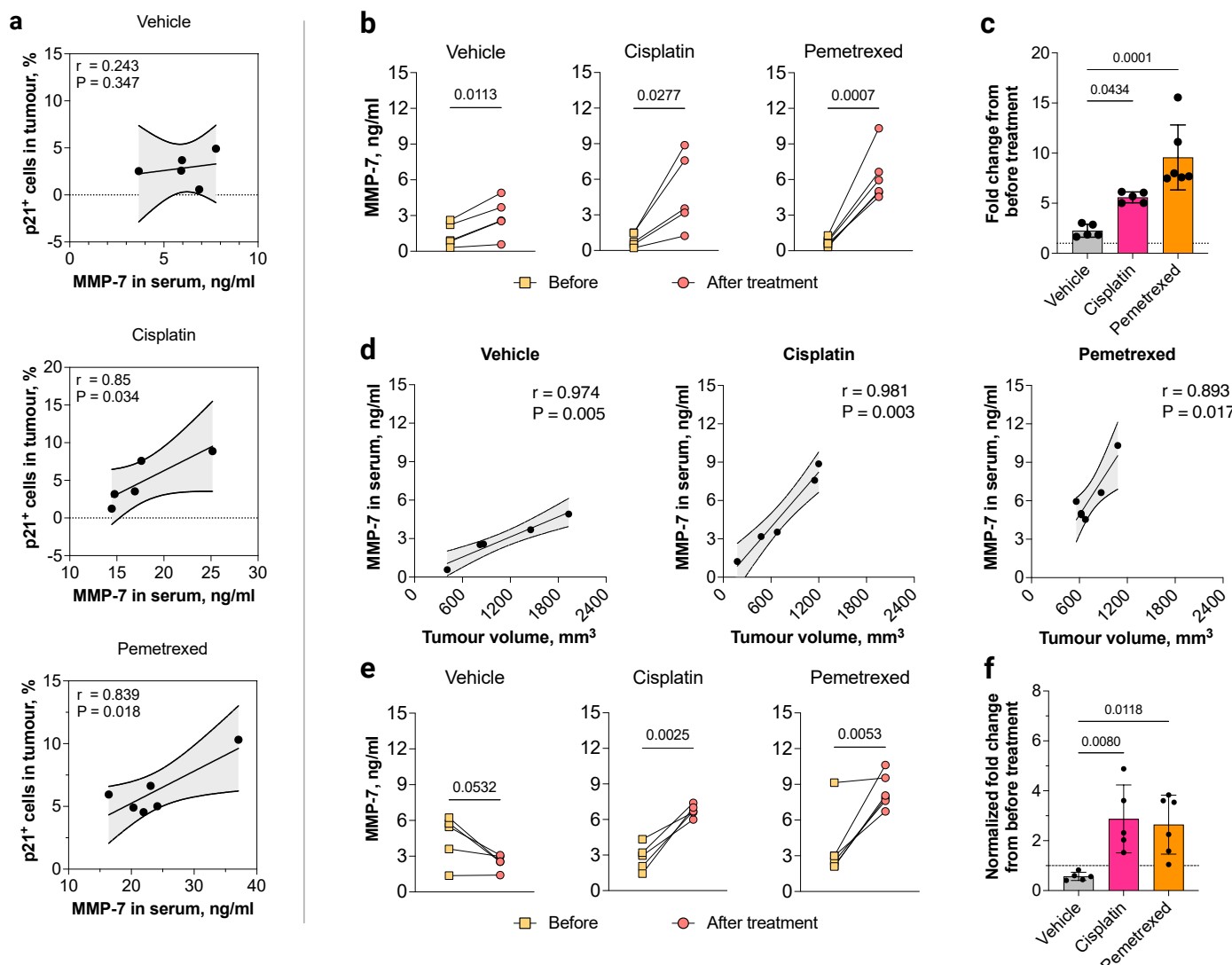

**Extended Data Fig. 3 | Higher p21+ levels in tumors correlate with circulating MMP-7 levels and treatment-induced changes in serum MMP-7. a**, Plot of p21+ cells in the tumor versus MMP-7 level in the serum for vehicle, cisplatin, and pemetrexed-treated mice (n = 5 independent mice for vehicle / cisplatin group, and n = 6 independent mice for pemetrexed group, one-tailed Pearson correlation). Solid lines indicate linear regression fits (least-squares fit; estimated mean relationship between variables), and dashed lines denote the 95% confidence intervals of the regression line. **b.** Serum concentration of MMP-7, before and after treatment (n = 5 independent mice for vehicle / cisplatin group, and n = 6 independent mice for pemetrexed group; paired two-tailed t-test). **c.** Fold change of serum concentration of MMP-7 before and after treatment (n = 5 independent mice for vehicle / cisplatin group, and n = 6 independent mice for pemetrexed group, mean ± s.d., ordinary one-way ANOVA

with Dunnett's multiple comparisons). **d.** Correlation plot of MMP-7 in the serum versus tumor volume (n = 5 independent mice for vehicle / cisplatin group, and n = 6 independent mice for pemetrexed group, two-tailed Pearson correlation). Solid lines indicate linear regression fits (least-squares fit; estimated mean relationship between variables), and dashed lines denote the 95% confidence intervals of the regression line. **e.** Concentration of MMP-7 in the serum, before and after treatment, normalized to tumor volume before and after treatment (n = 5 independent mice for vehicle / cisplatin group, and n = 6 independent mice for pemetrexed group; paired two-tailed t-test). **f.** Fold change of concentration of MMP-7 in the serum before and after treatment normalized to tumor volume (n = 5 independent mice for vehicle / cisplatin group, and n = 6 independence mice for pemetrexed group; mean ± s.d., ordinary one-way ANOVA with Dunnett's test multiple comparisons).

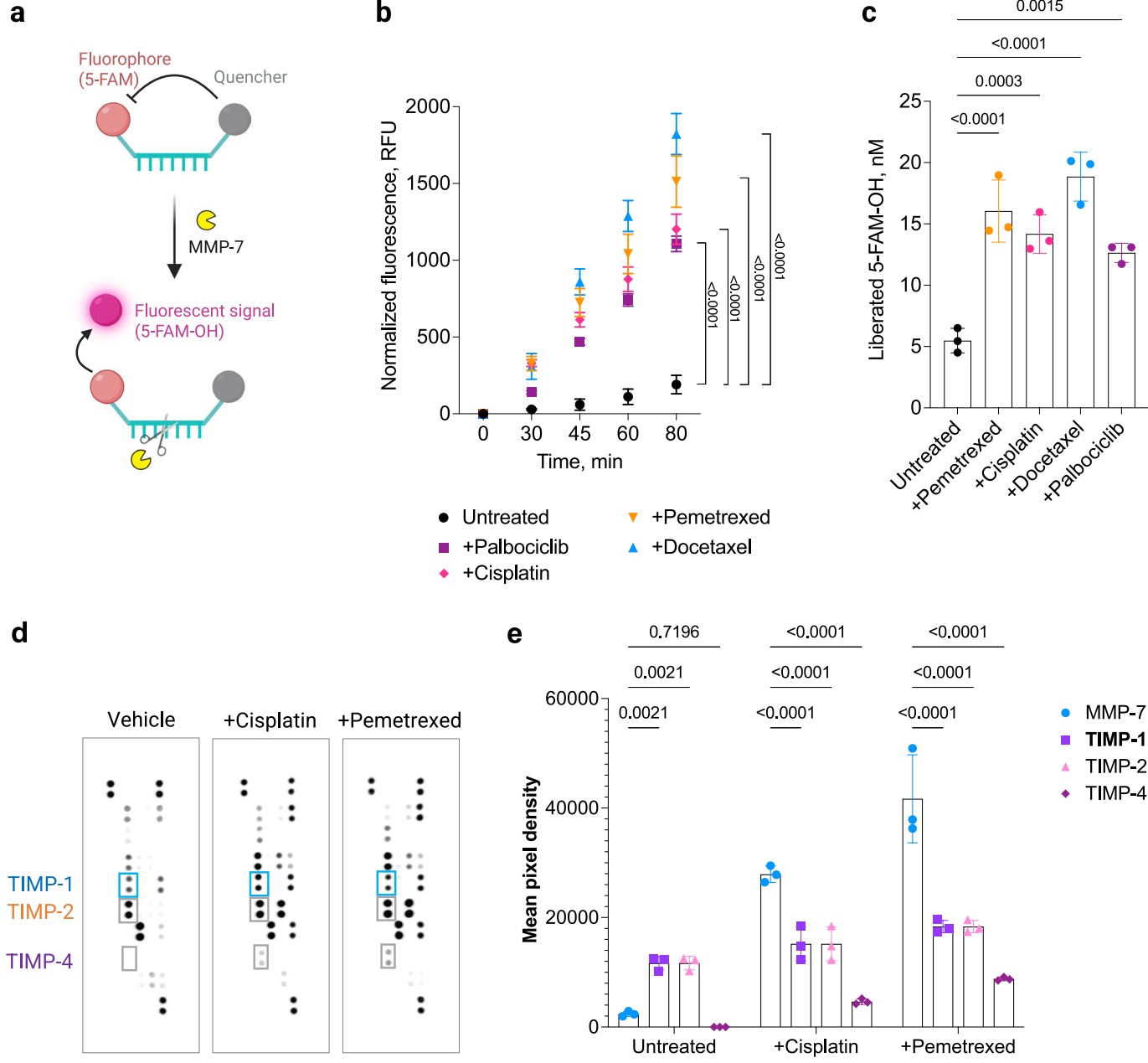

**Extended Data Fig. 4 | MMP-7 in the conditioned media of senescent A549 cells was enzymatically active. a.** Schematic illustration of the commercial MMP-7 activity assay used in this study. Created in BioRender. Fruk, L. (2026) https://BioRender.com/boen294. **b.** Secreted MMP-7 was active as enzyme, as measured by fluorescence-based assay (n = 3 independent biological replicates per group, mean ± s.d., two-way ANOVA with Dunnett's multiple comparisons). **c.** Graph of fluorophore liberated due to MMP-7 activity (n = 3 independent biological replicates per group, mean ± s.d., one-way ANOVA with Dunnett's multiple comparisons). **d.** Photograph of uncropped human protease inhibitor array panels of CM samples from untreated and treated A549 cells. All blots were exposed for 2 min. **e.** Pixel intensity z-score of secreted TIMP corresponding to (**d**) (n = 3 independent biological replicates per group, mean ± s.d., two-way ANOVA with Tukey's multiple comparisons).

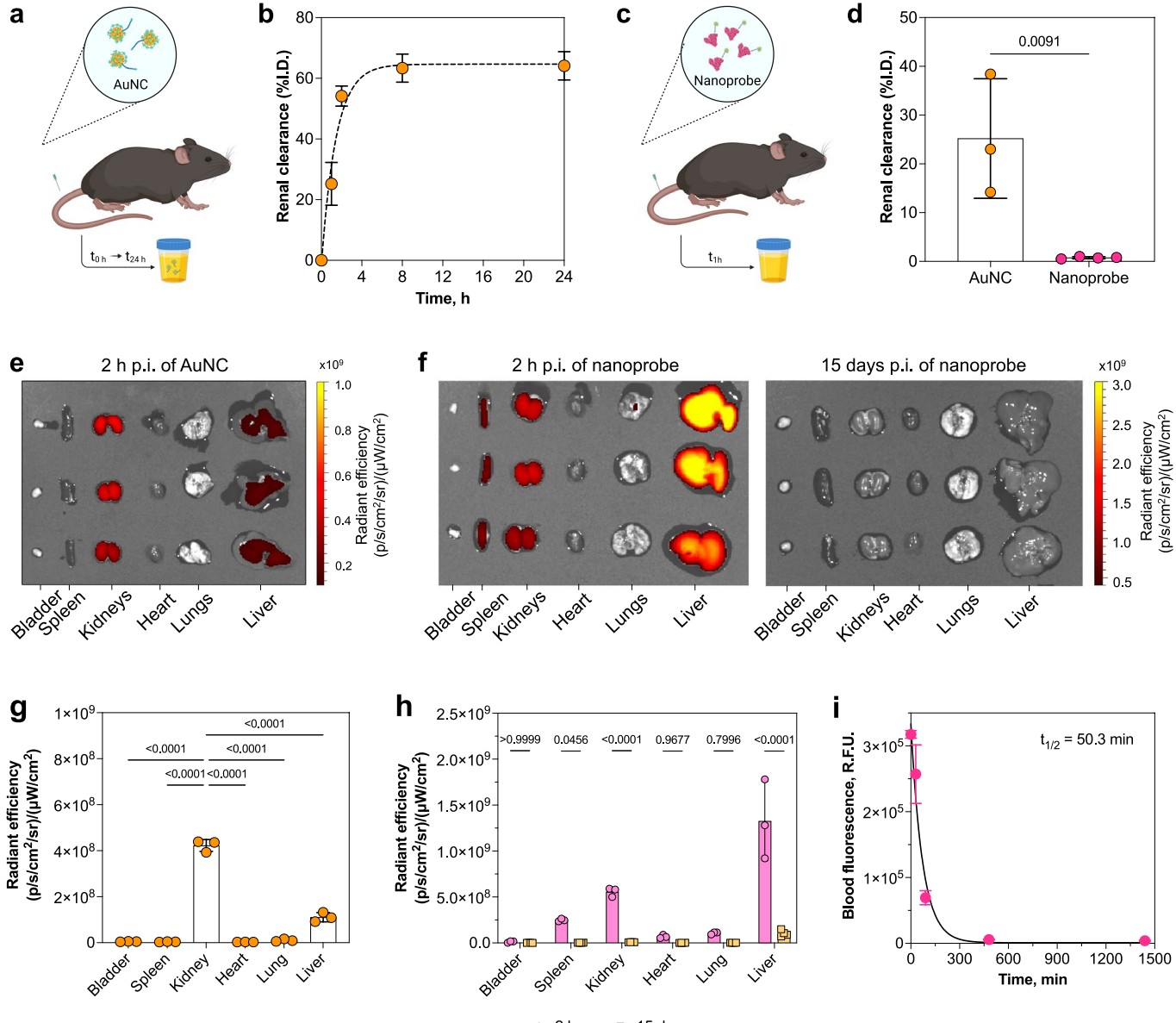

**Extended Data Fig. 5 | Renal clearance, biodistribution, and pharmacokinetics of AuNC and nanoprobes. a**. Schematic illustration of renal clearance study, where AuNCs were intravenously (i.v.) injected into C57BL/6 mice, and urine was collected within 24 h post-injection (p.i.). Created in BioRender. Fruk, L. (2026) https://BioRender.com/d8n1n72. Renally cleared AuNC content in the urine was measured using ICP–MS, plotted in (**b**) as % injected dose (I.D.) (n = 3 independent mice per group, mean ± s.e.m., exponential plateau fit). **c**. Schematic illustration of renal clearance study, where nanoprobes were i.v. injected into C57BL/6 mice, and urine was collected within 2 h p.i. Created in BioRender. Fruk, L. (2026) https://BioRender.com/ppgx0j8. Urine was analyzed using ICP–MS for Au content, plotted in (**d**), n = 3 independent mice for AuNC,

n = 4 independent mice for nanoprobes, mean ± s.e.m, unpaired two-tailed t-test). Organs were harvested at 2 h and 15 days p.i. IVIS images of main organs of mice 2 h p.i. of AuNC (**e**) and of mice 2 h and 15 days p.i. of nanoprobes (**f**), with corresponding fluorescence emission quantifications for each organ plotted in (**g**) for AuNC and (**h**) for nanoprobes (n = 3 independent mice for AuNC, n = 4 independent mice for nanoprobes, mean ± s.e.m., ordinary one-way ANOVA with Dunnett's multiple comparisons and two-way ANOVA with Šídák's multiple comparisons, respectively). **i**. Pharmacokinetic study of NIR-labeled nanoprobes in C57BL/6 mice. The serum concentration of nanoprobe was fit to a nonlinear fit (n = 4 independent mice, mean ± s.e.m, one-phase decay).

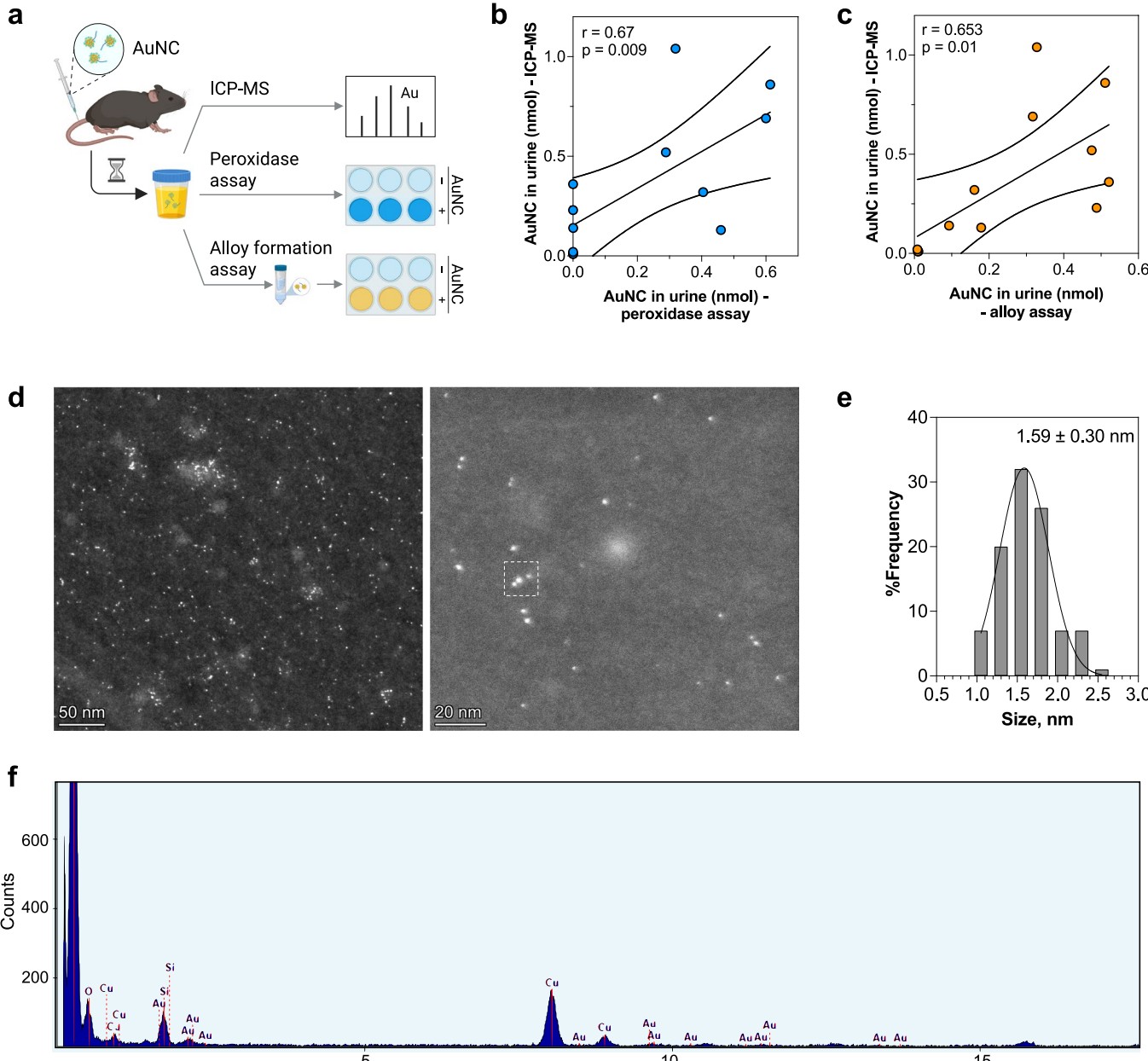

**Extended Data Fig. 6 | Colorimetric assays enabled sensitive detection of renally cleared AuNC in the urine. a**. Schematic of methods to detect the renally cleared AuNCs from mice. ICP-MS, peroxidase assay, and alloy formation assays were performed on the same urine samples to compare their performance in detecting AuNC. Created in BioRender. Fruk, L. (2026) https://BioRender.com/28w3bay. Correlation between the estimated AuNC contents measured in the urine by (**b**) peroxidase assay and (**c**) alloy formation assay versus total Au content measured by ICP-MS (n = 12 independent mice, one-tailed Pearson correlation). Solid lines indicate linear regression fits (least-squares fit; estimated mean relationship between variables), and dashed lines denote the 95% confidence intervals of the regression line. **d**. TEM images of renally cleared AuNCs and (**e**) their corresponding size distribution (n = 200 independent particles, Gaussian fit). Three independent experiments were performed yielding similar results. **f**. EDX spectra of the urine samples (white box area in (**d**)). Plot taken directly from the equipment. Three independent urine samples from n = 3 independent mice were analyzed with similar results. Scale bar = 20 or 50 nm as specified.

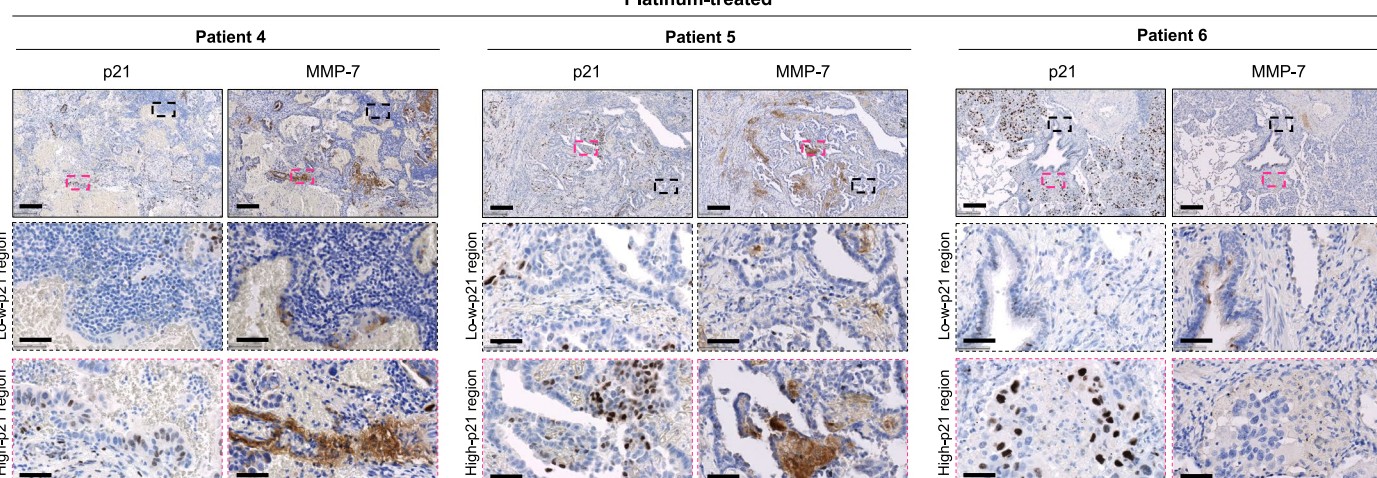

**Extended Data Fig. 7 | Platinum-treated tumor specimens showed a higher expression of MMP-7 and p21 compared to treatment-naive tumor specimens.** Representative histological images of NSCLC biopsy samples from treatment-naïve patients and platinum chemotherapy, analyzed for p21 and MMP-7 (n = 3 independent patients per group). Scale bars = 200 μm (top row) or 50 μm (bottom rows). The square indicates the area that is zoomed in and shown in detail below.

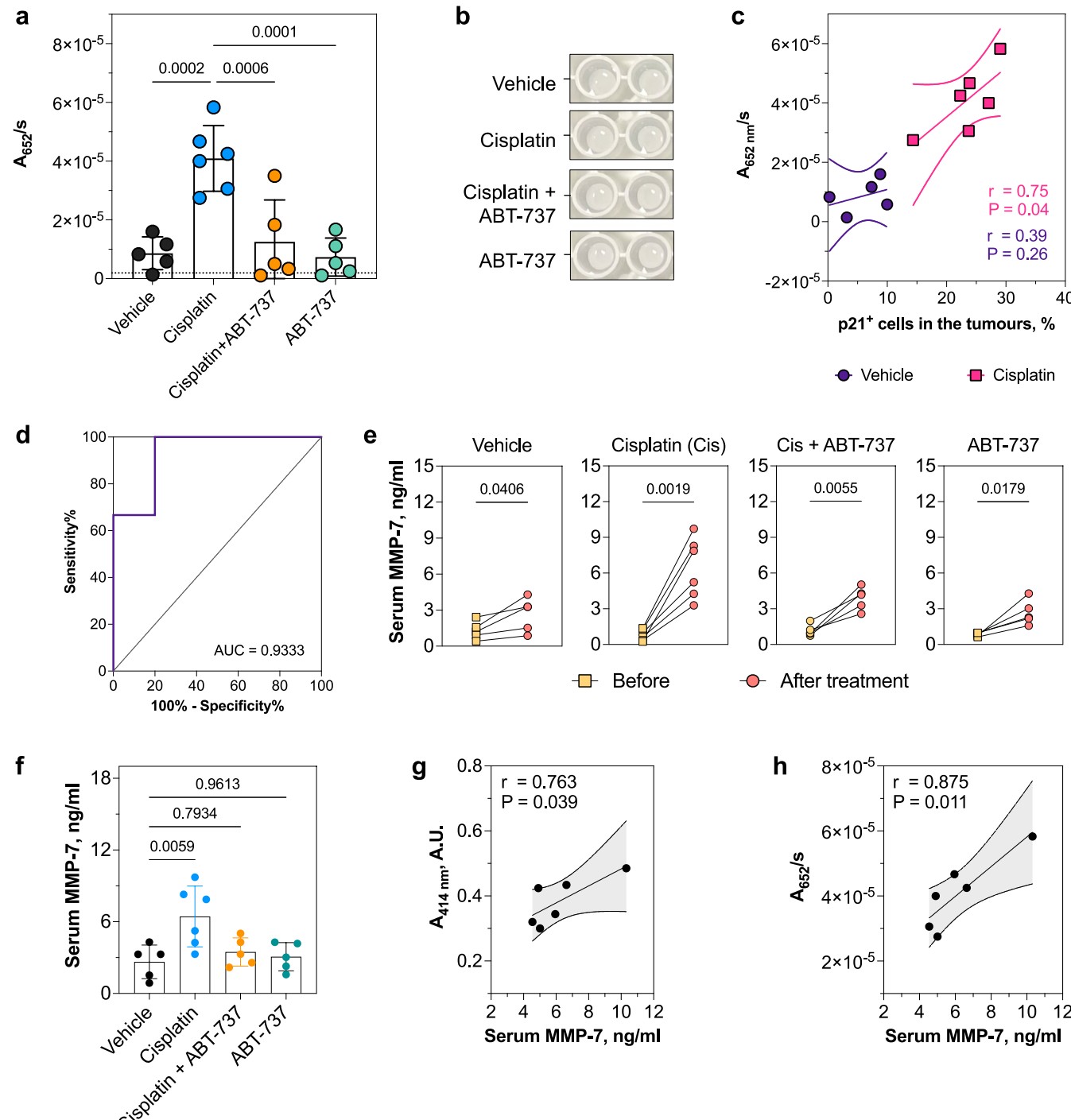

**Extended Data Fig. 8 | Peroxidase assay enables spectroscopic urinary detection of chemotherapy-induced senescence in lung cancer, and urinary signals from both alloy assay and peroxidase assay correlate with circulating MMP-7. a.** Initial kinetics ($A_{652nm}$/s) from peroxidase assays of urine samples collected from xenograft mice 2 h p.i. with nanoprobes (n = 5 independent mice per group, except for cisplatin group with n = 6 independent mice, mean ± s.e.m, ordinary one-way ANOVA with Dunnett's multiple comparison test). **b.** Photograph of peroxidase assay on urine samples from xenograft mice injected with nanoprobes. **c.** Correlation analysis between the percentage of MMP-7 in the tumors and in the serum from the xenograft mice, either cisplatin-treated or vehicle-treated group, and intensity of signals of the urine samples detected from peroxidase assay, $A_{652nm}$/s. Solid lines indicate linear regression fits (least-squares fit; estimated mean relationship between variables), and dashed lines denote the 95% confidence intervals of the regression line, n = 5 independent mice for vehicle group and n = 6 independent mice for cisplatin group. Statistics performed with one-tailed Pearson correlation. **d.** ROC analysis

showing the diagnostic specificity and sensitivity of nanoprobes and peroxidase assay in detecting senescence between cisplatin-treated and vehicle-treated mice groups (AUC = 0.9333, 95% CI = 0.7787 – 1), n = 5 independent mice for vehicle group and n = 6 independent mice for cisplatin group, r = Pearson's coefficient. The solid line represents the ROC curve, the dashed diagonal line represents the performance of a random classifier (AUC = 0.5). **e.** Serum concentration of MMP-7, before and after treatment (n = 5 independent mice per group, except for cisplatin group with n = 6 independent mice, paired two-tailed t-test). **f.** Serum concentration of MMP-7 after treatment (n = 5 independent mice per group, except for cisplatin group with n = 6 independent mice, mean ± s.d., ordinary one-way ANOVA with Dunnett's multiple comparisons). Correlation analysis between serum MMP-7 concentration and intensity of signals of the urine samples detected from (**g**) alloy assay and (**h**) peroxidase assay (n = 6 independent mice, statistics performed with one-tailed Pearson correlation). Solid lines indicate linear regression fits (least-squares fit; estimated mean relationship between variables), and dashed lines denote the 95% CI of the regression line.

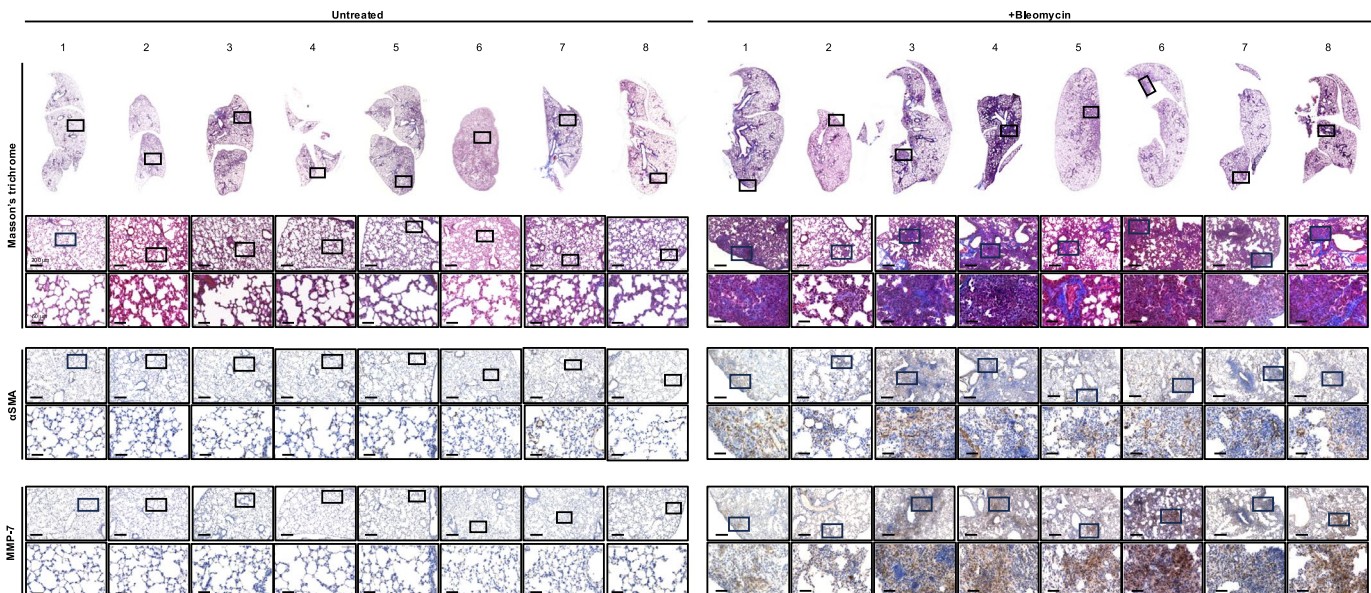

**Extended Data Fig. 9 | Histological validation of pulmonary fibrosis in bleomycin-treated mice.** Lungs from either untreated or bleomycin-treated mice (at the endpoint), stained for fibrosis markers (Masson's trichrome and α-SMA) and MMP-7 expression. Scale bar = 200 or 50 μm as specified (n = 8 independent mice per group). The square indicates the area that is zoomed in and shown in detail below.

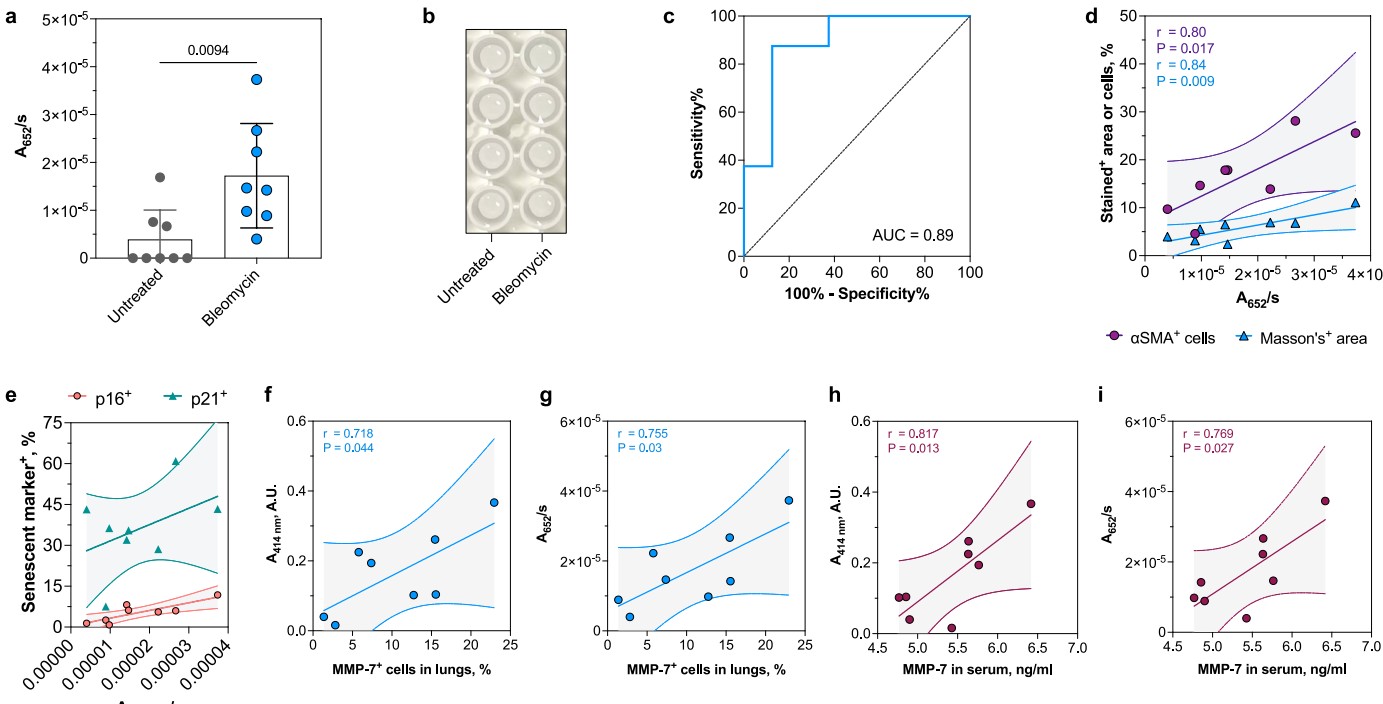

**Extended Data Fig. 10 | Peroxidase assay enables spectroscopic urinary detection of pulmonary fibrosis, and urinary signals from both alloy assay and peroxidase assay correlate with tissue and circulating MMP-7 levels.** Initial kinetic values ($A_{652nm}$/s) from peroxidase assay of urine samples collected from untreated or bleomycin-treated mice 2 h p.i. with nanoprobes (n = 8 independent mice per group, mean ± s.e.m, unpaired two-tailed t-test). **b**. Photograph of the peroxidase assay on urine samples from untreated or bleomycin-treated mice injected with nanoprobes. **c**. ROC analysis showing the diagnostic specificity and sensitivity of nanoprobes and alloy formation assay in detecting fibrosis between bleomycin-treated and untreated mice groups (AUC = 0.89, 95% CI = 0.7215 – 1, n = 8 independent mice). The solid line represents the ROC curve, the dashed diagonal line represents the performance of a random classifier (AUC = 0.5). **d**. Correlation analysis between the percentage of Masson's⁺ area and $\alpha$-SMA⁺ cells in the lungs and intensity of signals of the urine samples detected from peroxidase assay. Solid lines indicate linear regression fits (least-squares fit; estimated mean relationship between variables; and dashed lines denote the 95% confidence intervals of the regression line; n = 8 independent mice, two-tailed Pearson's correlation. **e**. Correlation analysis between the percentage of senescence marker, p21⁺ and p16⁺, cells with the urinary signal from peroxidase assay. Solid lines indicate linear regression fits (least-squares fit; estimated mean relationship between variables), and dashed lines denote the 95% confidence intervals of the regression line; n = 8 independent mice, two-tailed Pearson's correlation. Pearson's coefficient (r) and P-values: r = 0.4308, P = 0.2866 for p21; r = 0.8374, P = 0.0095 for p16. Correlation analysis between the percentage of MMP-7⁺ cells in the lungs (**f, g**) and MMP-7 in the serum (**h, i**) and intensity of signals of the urine samples detected from alloy formation assay (**f, h**) and peroxidase assay (**g, i**). Solid lines indicate linear regression fits (least-squares fit; estimated mean relationship between variables), and dashed lines denote the 95% confidence intervals of the regression line; n = 8 independent mice, two-tailed Pearson's correlation for (**f**)-(**i**). r = Pearson's coefficient.

# Reporting Summary

## Statistics

For all statistical analyses, confirm that the following items are present in the figure legend, table legend, main text, or Methods section.

| n/a | Confirmed | |
|---|---|---|
| ☐ | ☒ | The exact sample size (*n*) for each experimental group/condition, given as a discrete number and unit of measurement |
| ☐ | ☒ | A statement on whether measurements were taken from distinct samples or whether the same sample was measured repeatedly |
| ☐ | ☒ | The statistical test(s) used AND whether they are one- or two-sided<br>*Only common tests should be described solely by name; describe more complex techniques in the Methods section.* |
| ☐ | ☒ | A description of all covariates tested |
| ☐ | ☒ | A description of any assumptions or corrections, such as tests of normality and adjustment for multiple comparisons |
| ☐ | ☒ | A full description of the statistical parameters including central tendency (e.g. means) or other basic estimates (e.g. regression coefficient) AND variation (e.g. standard deviation) or associated estimates of uncertainty (e.g. confidence intervals) |
| ☐ | ☒ | For null hypothesis testing, the test statistic (e.g. *F*, *t*, *r*) with confidence intervals, effect sizes, degrees of freedom and *P* value noted<br>*Give P values as exact values whenever suitable.* |
| ☒ | ☐ | For Bayesian analysis, information on the choice of priors and Markov chain Monte Carlo settings |
| ☒ | ☐ | For hierarchical and complex designs, identification of the appropriate level for tests and full reporting of outcomes |
| ☒ | ☐ | Estimates of effect sizes (e.g. Cohen's *d*, Pearson's *r*), indicating how they were calculated |

*Our web collection on statistics for biologists contains articles on many of the points above.*

## Software and code

Policy information about availability of computer code

| Data collection | Processed and annotated single-cell RNA sequencing data from Huang et al (PMID: 39729352) was downloaded from https://figshare.com/articles/dataset/scRNA-sequencing_raw_data_of_LUAD/24797265 as stated from the original publication. |
|---|---|
| Data analysis | A Seurat object compiling all detected cells by Huang et al (PMID: 39729352) was created from the available count and meta data matrices using the CreateSeuratObject function of the Seurat package (v5.3). The epithelial cell cluster was obtained from the complete dataset using the subset function within Seurat (v5.3). Data from epithelial cells were re-processed using the standard Seurat pipeline and integrated using the Canonical Correlation Analysis (CCA) method. Uniform manifold approximation and projection (UMAP) was used to visualise the dataset in a two-dimensional space. Differential expression testing was performed using the FindMarkers function selecting "MAST" as test type. Significant upregulated genes between lung adenocarcinoma patients treated with neoadjuvant chemotherapy (NCT) (n=5) and naïve lung adenocarcinoma patients (Control, n=4) were defined by showing Log2Fold ≥1 and significant adjusted p-value or false discovery rate (FDR) <0.05. Gene set enrichment analysis was performed using the clusterProfiler R package (v4.14.6) against the Hallmark, the C2 canonical pathway collection (C2.cp.v5.1) and Human SenMayo (PMID: 35974106) gene sets that were downloaded from the Molecular Signatures Database (https://www.gsea-msigdb.org/gsea/msigdb). |

For manuscripts utilizing custom algorithms or software that are central to the research but not yet described in published literature, software must be made available to editors and reviewers. We strongly encourage code deposition in a community repository (e.g. GitHub). See the Nature Portfolio guidelines for submitting code & software for further information.

## Data

Policy information about availability of data

All manuscripts must include a data availability statement. This statement should provide the following information, where applicable:
- Accession codes, unique identifiers, or web links for publicly available datasets
- A description of any restrictions on data availability
- For clinical datasets or third party data, please ensure that the statement adheres to our policy

All data associated with this study are present in the paper, the Materials and Methods section, or the Supplementary Materials.

# Field-specific reporting

Please select the one below that is the best fit for your research. If you are not sure, read the appropriate sections before making your selection.

☒ Life sciences   ☐ Behavioural & social sciences   ☐ Ecological, evolutionary & environmental sciences

For a reference copy of the document with all sections, see nature.com/documents/nr-reporting-summary-flat.pdf

# Life sciences study design

All studies must disclose on these points even when the disclosure is negative.

| | |
|---|---|
| Sample size | Describe how sample size was determined, detailing any statistical methods used to predetermine sample size OR if no sample-size calculation was performed, describe how sample sizes were chosen and provide a rationale for why these sample sizes are sufficient. |
| Data exclusions | Describe any data exclusions. If no data were excluded from the analyses, state so OR if data were excluded, describe the exclusions and the rationale behind them, indicating whether exclusion criteria were pre-established. |
| Replication | Describe the measures taken to verify the reproducibility of the experimental findings. If all attempts at replication were successful, confirm this OR if there are any findings that were not replicated or cannot be reproduced, note this and describe why. |
| Randomization | Describe how samples/organisms/participants were allocated into experimental groups. If allocation was not random, describe how covariates were controlled OR if this is not relevant to your study, explain why. |
| Blinding | Describe whether the investigators were blinded to group allocation during data collection and/or analysis. If blinding was not possible, describe why OR explain why blinding was not relevant to your study. |

# Reporting for specific materials, systems and methods

We require information from authors about some types of materials, experimental systems and methods used in many studies. Here, indicate whether each material, system or method listed is relevant to your study. If you are not sure if a list item applies to your research, read the appropriate section before selecting a response.

### Materials & experimental systems

| n/a | Involved in the study |
|---|---|
| ☐ | ☒ Antibodies |
| ☐ | ☒ Eukaryotic cell lines |
| ☐ | ☐ Palaeontology and archaeology |
| ☐ | ☒ Animals and other organisms |
| ☐ | ☒ Human research participants |
| ☒ | ☐ Clinical data |
| ☐ | ☐ Dual use research of concern |

### Methods

| n/a | Involved in the study |
|---|---|
| ☒ | ☐ ChIP-seq |
| ☒ | ☐ Flow cytometry |
| ☒ | ☐ MRI-based neuroimaging |

## Antibodies

| | |
|---|---|
| Antibodies used | Antibody, Host Species, Manufacturer, Catalogue Number, Clone (where applicable)<br>Phosphorylated Rb (pRb), Rabbit, Cell signalling, D20B12<br>p21, Rabbit, Abcam, ab109520 and ab302893<br>MMP-7, Rabbit, Abcam, ab207299<br>Ki-67, Rabbit, Cell Signaling, 12202, D3B5<br>p16, Rabbit, Proteintech, 10883-1-AP<br>αSMA, Rabbit, Cell signaling, D4K9N<br>β-actin, Rabbit, Proteintech, 20536-1-AP |

HRP-conjugated AffiniPure Anti-Rabbit IgG (H+L), Donkey, Jackson ImmunoResearch, 711-035-152

| Validation | Antibody, Catalogue Number, Validation<br>Phosphorylated Rb (pRb), D20B12, validated by IHC, WB, IF; referenced by >552 publications.<br>p21, ab109520 and ab302893, validated by IHC, WB, IF; referenced by >530 publications.<br>MMP-7, ab207299, validated by IHC, WB, IF; referenced by >23 publications.<br>Ki-67, 12202, validated by IHC; referenced by >320 publications.<br>p16, 10883-1-AP, validated by WB, IHC, IF/ICC, FC (Intra), IP, ELISA referenced by >480 publications.<br>αSMA, D4K9N, validated by IHC, WB, IF; referenced by >850 publications.<br>β-actin, 20536-1-AP, validated by IHC, WB, IF; referenced by >3968 publications.<br>HRP-conjugated AffiniPure Anti-Rabbit IgG (H+L), 711-035-152, validated by IHC, WB, ELISA; referenced by >1328 publications. |

# Eukaryotic cell lines

Policy information about cell lines

| Cell line source(s) | A549: ATCC (Catalog. no. CCL-185).<br>SK-Mel-103: Sigma Aldrich (Catalog. no. SCC439).<br>MDA-MB-231: ATCC (Catalog. no. HTB-26).<br>PC-3: ATTC (Catalog. no. CRL-1435)<br>L1475(luc): generated from KrasG12D/WT;p53Fx/Fx mice.<br>HPF-a: ScienCell research laboratories (Catalog. no. 3310). |
| Authentication | A549 and HPF-a cells were authenticated by STR profiling by the Cancer Research UK Cambridge Institute. L1475(luc) cell line was validated as described in Turrell et al., 2017. |
| Mycoplasma contamination | All cell lines were routinely tested for mycoplasma infection by Universal Mycoplasma Detection Kit (ATCC, 30-1012K). |
| Commonly misidentified lines (See ICLAC register) | No commonly misidentified lines were used in the study. |

# Palaeontology and Archaeology

| Specimen provenance | *Provide provenance information for specimens and describe permits that were obtained for the work (including the name of the issuing authority, the date of issue, and any identifying information). Permits should encompass collection and, where applicable, export.* |
| Specimen deposition | *Indicate where the specimens have been deposited to permit free access by other researchers.* |
| Dating methods | *If new dates are provided, describe how they were obtained (e.g. collection, storage, sample pretreatment and measurement), where they were obtained (i.e. lab name), the calibration program and the protocol for quality assurance OR state that no new dates are provided.* |

☐ Tick this box to confirm that the raw and calibrated dates are available in the paper or in Supplementary Information.

| Ethics oversight | *Identify the organization(s) that approved or provided guidance on the study protocol, OR state that no ethical approval or guidance was required and explain why not.* |

Note that full information on the approval of the study protocol must also be provided in the manuscript.

# Animals and other organisms

Policy information about studies involving animals; ARRIVE guidelines recommended for reporting animal research

| Laboratory animals | Female athymic nude mice (Crl:NU(NCr)-Foxn1nu, 11 weeks) were used for MMP-7 validation and A549 xenograft tumour studies. Female C57BL/6 mice (11–12 weeks old) were used for renal clearance, toxicity, and pharmacokinetic, and lung fibrosis studies. Female C57BL/6 mice, either 2 months or 19 months old, were used to compare expression of MMP-7 in young and aged mice. CC57BL/6 mice (12.3-week-old) were used for 'healthy tissues with cisplatin mice experiments'. Female C57BL/6J mice were used for L1475(luc) orthotopic mice experiments. Complete descriptions can be found in relevant figure legends and Methods. |
| Wild animals | The study did not involve wild animals. |
| Field-collected samples | The study did not involve field-collected samples. |
| Ethics oversight | All mice protocols were approved for Ethical Conduct by the Home Office England and Central Biomedical Services (CBS) of the University of Cambridge, regulated under the Animals (Scientific Procedures) Act 1986, as stated in the International Guiding Principles for Biomedical Research involving animals, which fully comply with the current Home Office legislation. |

Note that full information on the approval of the study protocol must also be provided in the manuscript.

# Human research participants

Policy information about <u>studies involving human research participants</u>

| | |
|---|---|
| Population characteristics | The available patient information is provided in the Supplementary Table 1. |
| Recruitment | Human biopsies and ethical regulations. Human lung adenocarcinoma samples were collected from the Royal Papworth Hospital Research Tissue Bank (RPHRTB) after being reviewed by the RPHRTB project review committee (Project Number T02722). Lung tissue samples from patients with idiopathic pulmonary fibrosis were collected from the RPHRTB under Project Numbers T02147 and T02259.<br>RPHRTB has a derogation under the UK Human Tissue Authority (HTA) to supply samples (HTA number 12212) that are surplus to therapeutic necessity and were acquired with Research Ethics Committee-approved RPHRTB permission. Patients signed the RPHRTB general consent form, approving the use of their biopsies for research purposes and sample transfer was covered by a valid Material Transfer Agreement. Written consent was obtained for all tissue samples using Papworth Hospital Research Tissue Bank's ethical approval (East of England - Cambridge East Research Ethics Committee). Further clinical information on the adenocarcinoma samples is available in Supplementary Table 1. |
| Ethics oversight | Human lung adenocarcinoma samples were collected from the Royal Papworth Hospital Research Tissue Bank (RPHRTB) after being reviewed by the RPHRTB project review committee (Project Number T02722). RPHRTB has a derogation under the UK Human Tissue Authority (HTA) to supply samples (HTA number 12212) that are surplus to therapeutic necessity and were acquired with Research Ethics Committee-approved RPHRTB permission. Patients signed the RPHRTB general consent form, approving the use of their biopsies for research purposes and sample transfer was covered by a valid Material Transfer Agreement. Written consent was obtained for all tissue samples using Papworth Hospital Research Tissue Bank's ethical approval (East of England - Cambridge East Research Ethics Committee). |

Note that full information on the approval of the study protocol must also be provided in the manuscript.

# Dual use research of concern

Policy information about <u>dual use research of concern</u>

## Hazards

Could the accidental, deliberate or reckless misuse of agents or technologies generated in the work, or the application of information presented in the manuscript, pose a threat to:

| No | Yes | |
|---|---|---|
| ☒ | ☐ | Public health |
| ☒ | ☐ | National security |
| ☒ | ☐ | Crops and/or livestock |
| ☒ | ☐ | Ecosystems |
| ☒ | ☐ | Any other significant area |

## Experiments of concern

Does the work involve any of these experiments of concern:

| No | Yes | |
|---|---|---|
| ☒ | ☐ | Demonstrate how to render a vaccine ineffective |
| ☒ | ☐ | Confer resistance to therapeutically useful antibiotics or antiviral agents |
| ☒ | ☐ | Enhance the virulence of a pathogen or render a nonpathogen virulent |
| ☒ | ☐ | Increase transmissibility of a pathogen |
| ☒ | ☐ | Alter the host range of a pathogen |
| ☒ | ☐ | Enable evasion of diagnostic/detection modalities |
| ☒ | ☐ | Enable the weaponization of a biological agent or toxin |
| ☒ | ☐ | Any other potentially harmful combination of experiments and agents |

