## [Peer Review File · Nature Aging]

Urinary Detection of Therapy-Induced Senescence and Fibrosis Using Injectable Albumin-based Nanoprobe

Corresponding Author: Professor Ljiljana Fruk

Version 0:

Reviewer comments:

Reviewer #1

(Remarks to the Author)

In the current manuscript Hartono and collaborators developed a urinary Nanosensor for the non-invasive detection of cellular senescence in the contexts of (i) therapy-induced senescence in lung cancer and (ii) pulmonary fibrosis. This nanosensor is activated by the MMP-7 protease, which the authors found to be highly secreted by senescent cells as part of the SASP in the referred contexts, resulting in the renal clearance of a reporter (gold nanoclusters) that we can be measured in the urine by colorimetric reactions and also by spectroscopic techniques with very high sensitivity. The authors have validated this urinary nanosensor in in vitro models, and also in lung cancer xenografts murine models treated with cisplatin, and in bleomycin-induced pulmonary fibrosis mouse models.

The work is very interesting, well presented and particularly with future application potential. Yet, some issues in the work must be attended to improve it.

1. Since the Urinary Nanosensor is activated by MMP-7, the authors should generate knock-out MMP-7 lung cancer cells (A549-MMP7KO) to demonstrate in vitro that the activation of the nanosensor depends on MMP-7 secreted by senescent lung cancer cells but no other proteases or SASP components.
2. In addition to chemotherapy-treated A549 lung cancer cells, the authors should test a number of other cancer cell types (such as prostate, melanoma, and breast cancer cells) to determine whether the secretion of MMP-7 is a defining feature of therapy-induced senescent cancer cells in many tissues, or whether it is restricted to senescent lung cancer cells. In the latter case, this will define if this is a property of senescent lung cancer cells, making the described Urinary Nanosensor a very specific and sophisticated tool to be used in Lung Disease.
3. The authors have validated the nanosensor in bleomycin-induced pulmonary fibrosis in mice. Interestingly, MMP-7 has been related to a bad prognosis in human IPF. The authors could validate senescence biomarkers in the fibrotic lungs of their murine model at the histological level (e.g. Sa-b-gal activity, p21/p16 staining, negative staining for ki67) to associate senescence induction with MMP-7 expression and the activation of the nanosensor.
4. The authors are encouraged to discuss, even briefly, senescence clearance methods particularly targeted ones with none to minimal side effects (eg PMID 39730824), that could be coupled with detection platforms like the one presented in this manuscript to monitor the efficacy of the senolytic treatment.

Reviewer #2

(Remarks to the Author)

Hartono et al. report the development of ALBANC, an albumin-based nanosensor linked with gold nanoclusters (AuNCs) via an MMP-7–cleavable peptide, designed for urinary detection of cancer senescence and fibrosis. Following intravenous injection, ALBANC accumulates in the lungs, where upregulated MMP-7 cleaves the linker, releasing AuNCs that circulate and are renally excreted into urine. For readout, urinary AuNCs are used as nucleation sites to form Au@Ag alloys upon addition of AgNO₃, enhancing colorimetric detection sensitivity compared to conventional peroxidase-mimicking systems such as TMB assays. The platform is demonstrated in murine models of chemotherapy-induced tumor senescence and bleomycin-induced pulmonary fibrosis.

This study falls short in terms of novelty and translational relevance. The core concept is not new, and the overall system design closely resembles a previously published approach (Nat. Nanotechnol. 2019, 14, 883–890). The proposed silver-alloying strategy, though improving sensitivity, adds complexity to the detection workflow, thereby diminishing its practicality for point-of-care applications. As such, the manuscript does not represent a meaningful advance in the field of urinary diagnostics and is not suitable for publication in such a high-impact journal.

Major Concerns:

1. Lack of Novelty in Material Design and Experimental Approach

The ALBANC platform is highly similar to the AuNC-NAV system previously reported, which employed neutravidin-linked AuNCs via MMP-cleavable peptides for urinary detection. The in vivo mechanism—intravenous injection, protease-mediated cleavage in the tumor, renal clearance, and urinary detection—is nearly identical. The only conceptual variation lies in the colorimetric readout using AgNO₃-induced Au@Ag alloy formation, but this introduces additional steps including buffer exchange and chemical reagent addition, limiting its utility for simple diagnostics. Furthermore, Au/Ag-based signal amplification is a well-established technique in in vitro diagnostics (e.g., *Langmuir* 2018, 34, 13897–13904; *JACS* 2023, 145, 919–928).

2. Limited Understanding of the Field and Incomplete Literature Review

The authors appear to have limited familiarity with the field of artificial urinary nanosensors. This is evident in the superficial Introduction, which overlooks key references such as *Nat Rev Cancer* 2021, 21, 655–668; *Nat. Mater.* 2022, 21, 598–607; *Nat. Nanotechnol.* 2023, 18, 798–807; and *Nat Rev Bioeng* 2024, 2, 425–441. This lack of context has led to an overestimation of the novelty and significance of the current work.

3. Questionable Specificity of MMP-7 as a Senescence Marker

MMP-7 is not a specific biomarker for cellular senescence. It is upregulated in a wide range of cancers—including renal, colon, and prostate cancers—even in the absence of senescence (e.g., *Clin Cancer Res* 2008, 14, 5503–5511; *Cancer Cell* 2005, 7, 485–496). Fig. 1f of the manuscript confirms this, showing that 20% of untreated A549 cells are already MMP-7 positive. Moreover, MMP-7 expression is associated with non-malignant conditions such as infection, inflammation, and tissue injury (e.g., *Sci. Transl. Med.* 2017, 9, eaan8462), undermining the specificity of ALBANC for senescence detection.

4. Low Practicality for Point-of-Care Use

The system's reliance on ultrafiltration and multiple chemical reagents for Ag growth significantly reduces its practicality compared to previously reported urine-based assays that allow for direct readouts. While ALBANC may offer higher sensitivity than TMB-based methods, it remains a colorimetric approach, which typically has higher background signals than fluorescence or Raman-based assays. These optical methods are compatible with portable detectors and increasingly accessible in low-resource settings.

5. Limited Clinical Relevance of Animal Models

The senescence model involves chemotherapy and ABT-737 co-treatment, the latter of which is not FDA-approved and lacks selectivity, also inducing apoptosis in proliferating cancer cells (e.g., *Cancer Res* 2008, 68, 2321–2328). This undermines the specificity and interpretability of ALBANC's signal.

In the fibrosis model, the sensor detects established disease 14 days post-bleomycin. However, the clinical challenge lies in early detection, where interventions are most effective. The manuscript provides no data on whether ALBANC can detect early-stage fibrosis, limiting its translational impact.

6. Technical Oversights

The manuscript does not report the size of AuNC@Ag particles formed in urine or discuss whether size evolution affects colorimetric absorption. This is a critical omission, as particle size can directly influence optical properties and detection reproducibility.

7. Unfounded Claims of Platform Adaptability

The authors claim the platform can be adapted to detect other proteases, presenting this as a novel feature. However, this modularity has been extensively demonstrated in prior literature and does not represent a meaningful innovation in this context.

Reviewer #3

(Remarks to the Author)

In this manuscript, Hartono et al. developed an albumin-linked Au nanocluster (ALBANC) nanoprobe to detect therapy-induced senescence in lung cancer through non-invasive urine analysis. ALBANC consists of gold nanoclusters (AuNCs) conjugated to a human serum albumin via an MMP-7-cleavable peptide linker. The authors identified MMP-7 as a senescence-associated protease upregulated in chemotherapy-treated lung cancer cells using protease array analysis. Upon cleavage by MMP-7, AuNCs are released and renally excreted, allowing for urinary detection. The authors demonstrate that ALBANC responds to MMP-7 activity both in vitro and in vivo and show its utility in detecting chemotherapy-induced senescence and lung fibrosis in mouse models. They also developed a nanoparticle growth-based assay to detect AuNCs in urine and compared it to the conventional peroxidase assay. Although the new assay showed improved sensitivity, both assays performed similarly in detecting senescence and fibrosis in vivo. Overall, the study is comprehensive and well executed, but the novelty of the work is somewhat limited beyond the alloy-forming assay and use of albumin as a carrier.

Major points

1. The authors demonstrate that MMP-7 is upregulated in senescent lung cancer cells following chemotherapy both in vitro and in vivo, but not in a fibroblast cell line. However, the specificity and robustness of MMP-7 as a chemotherapy-induced senescence marker remain unclear. MMP-7 needs to be further validated as a biomarker for chemotherapy-induced senescence. For example, the authors could:
 - a. Evaluate MMP-7 expression in more non-cancerous cell lines, in healthy tissues following chemotherapy, or in aged mice.
 - b. Perform RNA analysis on publicly available data from drug-treated tumors to analyze the correlation between MMP-7 and

senescence in different contexts.

c. Discuss the limitations and future directions needed to validate MMP-7 as a robust biomarker for chemotherapy-induced senescence.

2. The authors suggest that MMP-7, as part of the senescence-associated secretory phenotype (SASP) profile, offers improved specificity over SA- β -gal for identifying chemotherapy-induced senescence. How would this platform distinguish between age-associated senescence from chemotherapy-induced senescence?

3. The authors developed an alloy-forming assay that lowered the limit of detection for AuNCs compared to conventional peroxidase assay. However, it would be more convincing to show that this increased sensitivity translates into improved performance in detecting senescence or fibrosis using in vivo samples. For example, can the alloy-forming assay detect senescence or fibrosis at earlier time points or in cases where the peroxidase assay fails to detect a signal?

Minor points

1. The figures in the supplementary material are not referenced in order. Reordering them to follow the order of appearance in the main text would improve readability.

2. There are several inconsistencies and mismatches between the main text and the supplementary material that should be carefully reviewed. For example:

a. Line 153 in the main text references Supplementary Fig. 3c-e, but Fig. appears unrelated to the statement.

b. The figures and captions in Supplementary Fig. 17c-e are out of order.

c. Line 222 states "Nanoprobe assembly was confirmed by gel electrophoresis (Fig. 3d) and dynamic light scattering (DLS) (Supplementary Fig. 17c), which indicated a size increase after conjugation." However, Supplementary Fig. 17c shows only a single peak without a shift in size.

d. Line 313 references Supplementary Fig. 32d-f, which does not exist.

3. In line 308, the authors state that the colorimetric assays "quantitatively" detect renally cleared AuNCs, which is imprecise given the reported correlation coefficients of 0.67 and 0.65. This claim should be revised to more accurately reflect the data.

Version 1:

Reviewer comments:

Reviewer #1

(Remarks to the Author)

The authors have properly addressed all issues raised. Therefore I recommend publication.

Reviewer #2

(Remarks to the Author)

This study has two major limitations. First, there is insufficient evidence demonstrating the specificity of MMP-7 for cancer-associated senescence (also expression in non-treated cancer, infection, and inflammation). Second, the artificial urinary probe design lacks substantial novelty beyond the alloy-forming strategy and the chemical linkage. These concerns were also raised by Reviewer 3.

To address the first limitation, the authors additionally tested their ALBANC probes in gene knockout cell lines, other cancer cell lines beyond lung cancer, aging mice, and patient specimens. The results appear to support the authors' claim that MMP-7 is highly expressed in chemotherapy-treated cancer cells and in fibrotic lung cancers. However, in the Discussion section, the authors also acknowledge that MMP-7 is not an optimal biomarker with high specificity for senescence as MMP-7 cannot reliably distinguish cancer-associated senescence from inflammation or infection, and additional biomarkers would be required to improve diagnostic accuracy.

To address the second limitation, the authors provided additional experiments demonstrating that the alloy-forming method achieves a lower limit of detection than traditional peroxidase-based assays, which is a meaningful improvement. They also claim that chemically linked ALBANC probes offer enhanced biosafety, reproducibility, and translational robustness compared with neutravidin-based conjugates formed through physical interactions (page 3). However, such advantages are not supported by direct experimental evidence.

Reviewer #3

(Remarks to the Author)

The authors were responsive to my comments and overall I am satisfied with the revised manuscript. I do want to comment on novelty as this was surfaced by the other reviewers as well. In my opinion, novelty can be divided into biological, technological, and translational advances. Here the authors have supportive data to target MMP-7 as a potentially new biological target of drug-induced senescence. They also transferred the in vivo sensing scaffold to albumin, which increases the translatability compared to the use of neutravidin in previous papers (although there are still likely too many moving parts for translation). While the general concept of urinary sensing via the application of protease sensors has been previously

described, the authors particular focus on drug-induced senescence is novel and would expose this type of approach to a broader audience than the previous studies. For these reasons, I am supportive of publication.

Response to Reviewers
**“Urinary Detection of Therapy-Induced Senescence and Fibrosis Using
Injectable Albumin-Based Nanoprobe” (NATAGING-T09593)**

Reviewer #1:

In the current manuscript Hartono and collaborators developed a urinary Nanosensor for the non-invasive detection of cellular senescence in the contexts of (i) therapy-induced senescence in lung cancer and (ii) pulmonary fibrosis. This nanosensor is activated by the MMP-7 protease, which the authors found to be highly secreted by senescent cells as part of the SASP in the referred contexts, resulting in the renal clearance of a reporter (gold nanoclusters) that can be measured in the urine by colorimetric reactions and also by spectroscopic techniques with very high sensitivity. The authors have validated this urinary nanosensor in in vitro models, and also in lung cancer xenografts murine models treated with cisplatin, and in bleomycin-induced pulmonary fibrosis mouse models. The work is very interesting, well presented and particularly with future application potential. Yet, some issues in the work must be attended to improve it.

We thank the reviewer for the enthusiastic and insightful comments.

1. *Since the Urinary Nanosensor is activated by MMP-7, the authors should generate knock-out MMP-7 lung cancer cells (A549-MMP7KO) to demonstrate in vitro that the activation of the nanosensor depends on MMP-7 secreted by senescent lung cancer cells but no other proteases or SASP components.*

We thank the reviewer for raising this important point. To directly address it, we generated MMP-7-deficient A549 cells (MMP-7^{-/-} A549) and induced senescence with cisplatin (new Fig. 4g, new Supplementary Fig. 33). Unlike wild-type (WT) A549 cells, MMP-7^{-/-} cells did not upregulate intracellular or secreted MMP-7 (new Fig. 4h) upon senescence induction. Moreover, conditioned media from senescent MMP-7^{-/-} cells failed to cleave the ALBANC nanoprobe, in contrast to wild-type cells (new Fig. 4i). These results confirm that ALBANC activation is MMP-7-dependent and rule out contributions from off-target protease activity.

2. *In addition to chemotherapy-treated A549 lung cancer cells, the authors should test a number of other cancer cell types (such as prostate, melanoma, and breast cancer cells) to determine whether the secretion of MMP-7 is a defining feature of therapy-induced senescent cancer cells in many tissues, or whether it is restricted to senescent lung cancer cells. In the latter case, this will define if this is a property of senescent lung cancer cells, making the described Urinary Nanosensor a very specific and sophisticated tool to be used in Lung Disease.*

We thank the reviewer for this helpful suggestion. We conducted experiments in melanoma (SK-MEL-103), prostate (PC-3), and breast cancer (MDA-MB-231) cell lines treated with clinically relevant senescence-inducing agents (Supplementary Fig. 10). Each cell type exhibited distinct SASP protease signatures (e.g., MMP-13 in PC-3, cathepsin A in SK-MEL-103, MMP-3 in MDA-MB-231), but none upregulated MMP-7 significantly (new Supplementary Fig. 11). These findings demonstrate that MMP-7 upregulation is lineage-specific to lung cancer cells subjected to senescence-inducing chemotherapy.

3. *The authors have validated the nanosensor in bleomycin-induced pulmonary fibrosis in mice. Interestingly, MMP-7 has been related to a bad prognosis in human IPF. The authors could validate senescence biomarkers in the fibrotic lungs of their murine model at the*

histological level (e.g. Sa-b-gal activity, p21/p16 staining, negative staining for ki67) to associate senescence induction with MMP-7 expression and the activation of the nanosensor.

We thank the reviewer for this helpful suggestion. We performed additional histological analyses in the bleomycin model demonstrating increased p16 and p21 expression within fibrotic lesions, with clear co-localization of these senescence markers and regions of elevated MMP-7 expression (new Fig. 7). Quantitative analyses further revealed positive correlations between fibrosis markers (collagen, α -SMA), senescence markers, and MMP-7 (new Supplementary Fig. 44). Together, these data strengthen the link between MMP-7 activity, senescence, and fibrotic remodelling.

4. *The authors are encouraged to discuss, even briefly, senescence clearance methods particularly targeted ones with none to minimal side effects (eg PMID 39730824), that could be coupled with detection platforms like the one presented in this manuscript to monitor the efficacy of the senolytic treatment.*

We thank the reviewer for this useful suggestion. We have now expanded the Discussion to address this point (lines 659-663): “Furthermore, it serves as a valuable tool for monitoring responses to senolytic therapies. This is particularly significant given the number of senolytic agents currently in development and clinical trials, as well as the lack of effective tools to evaluate their efficacy. ALBANC readouts could be paired with emerging targeted senescence-clearance/senolytic strategies to iteratively optimize dosing, maximize efficacy, and minimize side effects.

Reviewer #2 (Remarks to the Author):

Hartono et al. report the development of ALBANC, an albumin-based nanosensor linked with gold nanoclusters (AuNCs) via an MMP-7–cleavable peptide, designed for urinary detection of cancer senescence and fibrosis. Following intravenous injection, ALBANC accumulates in the lungs, where upregulated MMP-7 cleaves the linker, releasing AuNCs that circulate and are renally excreted into urine. For readout, urinary AuNCs are used as nucleation sites to form Au@Ag alloys upon addition of AgNO₃, enhancing colorimetric detection sensitivity compared to conventional peroxidase-mimicking systems such as TMB assays. The platform is demonstrated in murine models of chemotherapy-induced tumor senescence and bleomycin-induced pulmonary fibrosis.

This study falls short in terms of novelty and translational relevance. The core concept is not new, and the overall system design closely resembles a previously published approach (Nat. Nanotechnol. 2019, 14, 883–890). The proposed silver-alloying strategy, though improving sensitivity, adds complexity to the detection workflow, thereby diminishing its practicality for point-of-care applications. As such, the manuscript does not represent a meaningful advance in the field of urinary diagnostics and is not suitable for publication in such a high-impact journal.

We thank the reviewer for their careful evaluation of our manuscript and address the points raised below.

Major Concerns:

1. Lack of Novelty in Material Design and Experimental Approach

The ALBANC platform is highly similar to the AuNC-NAV system previously reported, which employed neutravidin-linked AuNCs via MMP-cleavable peptides for urinary detection. The in vivo mechanism—intravenous injection, protease-mediated cleavage in the tumor, renal clearance, and urinary detection—is nearly identical. The only conceptual variation lies in the

colorimetric readout using AgNO₃-induced Au@Ag alloy formation, but this introduces additional steps including buffer exchange and chemical reagent addition, limiting its utility for simple diagnostics. Furthermore, Au/Ag-based signal amplification is a well-established technique in *in vitro* diagnostics (e.g., *Langmuir* 2018, 34, 13897–13904; *JACS* 2023, 145, 919–928).

We respectfully disagree with the Reviewer's assessment of limited novelty. Although both systems rely on renal clearance and urinary detection of AuNCs, ALBANC differs significantly from the previously reported AuNC-Neutravidin platform in both material design and functional scope. Specifically, ALBANC employs albumin rather than neutravidin as the carrier protein and uses azide-DCBO click chemistry to achieve covalent AuNC conjugation, in contrast to the non-covalent biotin-neutravidin assembly used previously. The design confers enhanced biostability and translational robustness.

While Au/Ag-based signal amplification has been reported in other diagnostic contexts, this study is, to our knowledge, the first to implement silver growth-based amplification for the detection of renally cleared AuNCs in urine. Substantial optimisation was required to adapt this chemistry to the urinary environment, resulting in a approx. 250-fold sensitivity improvement over conventional peroxidase-mimicking assays. Given that silver enhancement is already widely used in point-of-care formats such as lateral flow assays, we view this approach as facilitating, rather than limiting, clinical translation.

Importantly, the principal innovations of this work extend beyond nanomaterial design and align with the scope of Nature aging. These include i) identification of MMP-7 as a context-dependent biomarker of therapy-induced senescence in lung cancer and senescence-associated pulmonary fibrosis, and ii) development of a non-invasive, translatable platform for early detection and monitoring of senescence and fibrosis in diseases of high unmet clinical need.

2. Limited Understanding of the Field and Incomplete Literature Review

The authors appear to have limited familiarity with the field of artificial urinary nanosensors. This is evident in the superficial Introduction, which overlooks key references such as Nat Rev Cancer 2021, 21, 655–668; Nat. Mater. 2022, 21, 598–607; Nat. Nanotechnol. 2023, 18, 798–807; and Nat Rev Bioeng 2024, 2, 425–441. This lack of context has led to an overestimation of the novelty and significance of the current work.

We thank the reviewer for this comment. We note that the original manuscript already cited 11 key references on artificial urinary nanosensors (References 28-39 in the original manuscript), including “Nat Rev Cancer 2021, 21, 655–668” suggested by the reviewer (Citation No. 39 in the original manuscript). Nonetheless, to further strengthen the contextual framework, we have expanded the Introduction to include the additional reviewers highlighted by Reviewer, as well as a recent relevant primary study (Rojas-Vazquez, *Nat Comm* 2024, 15, 775). These additions better situate our work within the existing literature while clarifying its distinct biological and translational contributions.

3. Questionable Specificity of MMP-7 as a Senescence Marker

MMP-7 is not a specific biomarker for cellular senescence. It is upregulated in a wide range of cancers—including renal, colon, and prostate cancers—even in the absence of senescence (e.g., Clin Cancer Res 2008, 14, 5503–5511; Cancer Cell 2005, 7, 485–496). Fig. 1f of the manuscript confirms this, showing that 20% of untreated A549 cells are already MMP-7 positive. Moreover, MMP-7 expression is associated with non-malignant conditions such as infection, inflammation, and tissue injury (e.g., Sci. Transl. Med. 2017, 9, eaan8462), undermining the specificity of ALBANC for senescence detection.

We thank the reviewer for raising this important point. We fully agree that no single, universal biomarker uniquely defines cellular senescence. Even widely used markers such as p16 or SA β -Gal are not sufficient across all senescent context and can be detected in non-senescent settings, including cancer, inflammation and tissue injury. Accordingly, we do not present MMP-7 as a universal or senescence-specific biomarker marker.

Instead, our study explicitly positions MMP-7 as a context-dependent marker of senescence, particularly in the setting of chemotherapy-induced senescence in lung tumours and senescence-associated pulmonary fibrosis. This conclusion is supported by several key observations:

1. Selective upregulation in senescent lung cancer cells: MMP-7 was markedly upregulated following cisplatin-induced senescence in human (A549) and murine lung cancer cells, but not in non-lung cancer cell lines or non-senescent contexts (Fig 2; Supplementary Fig. 1, 2, 3, 4; new Supplementary Fig. 10, 11).
2. Validation *in vivo* and in patients: MMP-7 accumulation associated with senescence markers was confirmed in xenograft and orthotopic lung tumour models and in samples from lung cancer patients treated with platinum-based (new Fig. 5a-i; new Supplementary Fig. 38, 39).
3. Lack of induction by systemic damage or ageing: Cisplatin treatment in non-tumour - bearing mice did not elevate MMP-7 levels, arguing against a nonspecific response to inflammation or tissue injury (new Supplementary Fig. 13). Likewise, naturally aged mice exhibited increased p16 expression without corresponding MMP-7 upregulation (new Supplementary Fig. 12).
4. Functional role within ALBANC: Within the ALBANC platform, MMP-7 serves as a biochemical actuator enabling nanoprobe cleavage and urinary signal generation, rather than as a standalone diagnostic marker. The readout therefore reflects localised protease activity in a defined pathological context.

Thus, our approach intentionally departs from “pan-senescence” detection strategies and instead exploits cell-type and context-specific senescence features, which we argue is essential for both accurate detection and translational relevance. We now explicitly discuss these limitations and the context dependence of MMP-7 in the revised Discussion (lines 696-707)

4. Low Practicality for Point-of-Care Use

The system's reliance on ultrafiltration and multiple chemical reagents for Ag growth significantly reduces its practicality compared to previously reported urine-based assays that allow for direct readouts. While ALBANC may offer higher sensitivity than TMB-based methods, it remains a colorimetric approach, which typically has higher background signals than fluorescence or Raman-based assays. These optical methods are compatible with portable detectors and increasingly accessible in low-resource settings.

We thank the reviewer for these thoughtful concerns and the opportunity to clarify the point-of-care applicability of ALBANC.

First, we note that the silver amplification-based colorimetric readout exhibits very low background, as shown in Fig. 3f and 3j, enabling robust detection of AuNCs at low picomolar concentrations. Importantly, ALBANC is designed as an endpoint assay, which simplifies implementation relative to kinetic enzymatic readouts that require precise timing and continuous monitoring. Endpoint formats are inherently more compatible with low-complexity and decentralized workflows.

While we agree that fluorescence and Raman-based approaches are powerful, their practical deployment in PoC settings remains constrained. Raman detection of AuNCs at relevant concentrations typically relies on surface-enhanced Raman that requires substrates that are costly and difficult to standardise, while Raman reporters require additional surface functionalisation that may interfere with protease cleavage or probe stability in urine. Fluorescence-based urine assays have similarly reported limited sensitivity without extensive sample processing, as noted in prior senescence-detection studies (*Rojas-VAzquez et al., Nat. Commun., Ref 40*)

In contrast, ALBANC relies on a simple optical density measurement in a well-established colorimetric format that is inexpensive, reproducible, and compatible with portable readers, including handheld photometers or smartphone -based devices commonly used in decentralised diagnostics. Although ultrafiltration and reagent addition are currently part of the workflow, these steps are readily adaptable to cartridge- or lateral flow-style formats, where filtration and silver deposition can be integrated into a single-use device. Several commercial colorimetric diagnostic systems already include multiple chemical steps in a user-friendly design, showing that complexity can be abstracted from the end user.

In line with the Reviewer's suggestion, we have now explicitly acknowledged the current limitations for PoC deployment and future integration strategies in the Discussion (lines 708-722)

5. Limited Clinical Relevance of Animal Models

The senescence model involves chemotherapy and ABT-737 co-treatment, the latter of which is not FDA-approved and lacks selectivity, also inducing apoptosis in proliferating cancer cells (e.g., Cancer Res 2008, 68, 2321–2328). This undermines the specificity and interpretability of ALBANC's signal.

We thank the reviewer for raising this important point. We agree that ABT-737 is not an FDA-approved, however, there are currently no approved senolytic therapies. Importantly, ABT-737 is not used here as a translational candidate, but rather as a tool compound with well-established senolytic activity to functionally validate the specificity of ALBANC for detecting therapy-induced senescence *in vivo*. The goal of these experiments is not to model a clinical regimen, but to demonstrate that partial senescent-cell clearance results in a corresponding reduction in the ALBANC urinary signal (new Fig. 6 and new Fig. 8), thereby supporting probe specificity.

ABT-737 is a BH3 mimetic that inhibits anti-apoptotic proteins Bcl-2, Bcl-xL and Bcl-w and has been extensively validated as a senolytic across multiple senescence-driven disease models, including lung cancer, pulmonary injury and ageing-related pathologies, by several independent groups (including our own):

- Platinum Senescence in Lung Cancer. Gonzalez-Gualda et al. Nature Aging. In Press.
- Senescent Macrophages in Lung Cancer. Haston et al. Cancer Cell 2023. PMID 37267953.
- Senescence in PanINs. Kolodkin-Gal et al. Gut 2022. PMID 33649045.
- Liver regeneration model. Ritchka et al. Genes & Dev 2020. PMID 32139422.
- LMNA^{+G609G} Progeroid Senescence. Ovadya et al. Nat Comm 2018. PMID 30575733.
- Senescence in Lung Irradiation. Yosef et al. Nat Comm 2016. PMID 27048913.

Its use in this context is therefore well supported by the literature.

Unfortunately, genetic senescence-ablation models (e.g. our p16FDR mice, PMID 37267953) were not suitable alternatives in our study, as therapy-induced senescence in our lung cancer

and fibrosis models is predominantly p21-dependent and confined to transplanted tumour cells rather than the host stroma.

However, to further strengthen translational relevance and address this concern, we have added multiple complementary validations in the revised manuscript:

- i) confirmation of ALBANC performance in **a more physiologically-relevant orthotopic lung cancer model** using KRasG12D/WT;p53^{-/-} (KP) L1475(luc) cell line with platinum chemotherapy (new Supplementary Fig. 9). The results recapitulate the results obtained from A549 xenograft model.
- ii) demonstration of ALBANC sensitivity **at early stages of bleomycin-induced pulmonary fibrosis** in mice (new Fig 8i-m; new Supplementary Fig. 47).
- iii) validation of **MMP-7 accumulation in clinical samples and transcriptomic datasets** from lung adenocarcinoma patients treated with platinum-based therapy (new Fig. 5; new Supplementary Fig. 38; new Supplementary Fig. 39), and
- iv) confirmation of **MMP-7 association with senescence and fibrosis** in samples from patients with idiopathic pulmonary fibrosis (new Fig. 7).

Together, these data reinforce the clinical relevance of the models used and support the applicability of ALBANC for detecting therapy-induced senescence and fibrosis in translational settings.

6. Technical Oversights

The manuscript does not report the size of AuNC@Ag particles formed in urine or discuss whether size evolution affects colorimetric absorption. This is a critical omission, as particle size can directly influence optical properties and detection reproducibility.

We appreciate the reviewer's point and would like to note that the size of the alloyed AuNC@Ag nanoparticles was in fact reported in the manuscript: "*TEM analysis of the alloy nanoparticles after the assay duration showed a size distribution of 30.9 ± 7.8 nm.*" We agree that particle size can influence plasmonic absorption and have now explicitly acknowledged this principle in the revised text.

Importantly, size effects are controlled for in our assay design:

- i) The readout is performed as a standardized endpoint measurement, minimising variability arising from growth kinetics.
- ii) Quantification relies on calibration curves generated from known concentrations of AuNC seeds processed under identical conditions, ensuring that absorbance changes reflect seed concentration rather than random variation in growth kinetics, and
- iii) The starting AuNC population is highly monodisperse, providing a uniform template for silver overgrowth and reproducible optical properties.

These points are now explicitly discussed in the revised manuscript (lines 622-626)

7. Unfounded Claims of Platform Adaptability

The authors claim the platform can be adapted to detect other proteases, presenting this as a novel feature. However, this modularity has been extensively demonstrated in prior literature and does not represent a meaningful innovation in this context.

We thank the reviewer for this comment and agree that the modularity in protease-responsive nanoparticle systems has been demonstrated in prior work. We do not present modularity itself as a novel concept. Rather, we **highlight the practical implications of modularity** within the ALBANC framework.

Specifically, what distinguishes ALBANC is the implementation of protease-responsive modularity in a highly sensitive, non-invasive urinary readout suitable for in vivo detection of senescence-associated tissue states. While prior studies have largely demonstrated modular protease sensing in vitro or in formats requiring imaging or invasive sampling, our work shows that such modularity can be implemented in a urine-based diagnostic system with translational relevance. This is exemplified by successful detection of two distinct disease contexts in vivo: chemotherapy-treated lung cancer and pulmonary fibrosis, using the same core platform.

Thus, our contribution is not the conceptual novelty of modular protease sensing per se, but the demonstration that this versatility (i.e by exchanging the protease-cleavable linker) can be leveraged in a functional, translatable urine nanosensor platform, which could be readily adapted to alternative proteases as dictated by disease-and context-specific biology.

Reviewer #3:

In this manuscript, Hartono et al. developed an albumin-linked Au nanocluster (ALBANC) nanoprobe to detect therapy-induced senescence in lung cancer through non-invasive urine analysis. ALBANC consists of gold nanoclusters (AuNCs) conjugated to a human serum albumin via an MMP-7-cleavable peptide linker. The authors identified MMP-7 as a senescence-associated protease upregulated in chemotherapy-treated lung cancer cells using protease array analysis. Upon cleavage by MMP-7, AuNCs are released and renally excreted, allowing for urinary detection. The authors demonstrate that ALBANC responds to MMP-7 activity both in vitro and in vivo and show its utility in detecting chemotherapy-induced senescence and lung fibrosis in mouse models. They also developed a nanoparticle growth-based assay to detect AuNCs in urine and compared it to the conventional peroxidase assay. Although the new assay showed improved sensitivity, both assays performed similarly in detecting senescence and fibrosis in vivo. Overall, the study is comprehensive and well executed, but the novelty of the work is somewhat limited beyond the alloy-forming assay and use of albumin as a carrier.

We thank the reviewer for their time and constructive evaluation of our manuscript and appreciate their assessment that the study is comprehensive and well executed. With respect to the use of albumin as a carrier and questions regarding novelty, we refer the reviewer to our detailed response to Reviewer 2 (Point 1), where we clarify the material design advances and translational rationale of this choice.

Major points

1. The authors demonstrate that MMP-7 is upregulated in senescent lung cancer cells following chemotherapy both in vitro and in vivo, but not in a fibroblast cell line. However, the specificity and robustness of MMP-7 as a chemotherapy-induced senescence marker remain unclear. MMP-7 needs to be further validated as a biomarker for chemotherapy-induced senescence. For example, the authors could:
a. Evaluate MMP-7 expression in more non-cancerous cell lines, in healthy tissues following chemotherapy, or in aged mice.

We thank the reviewer for this helpful suggestion. In response, we performed additional in vivo validations to further assess the specificity of MMP-7 as a chemotherapy-induced senescence marker. We show that MMP-7 expression does not increase in aged (19-month-old) C57BL/6 mouse tissues (lung, kidney, heart) despite elevated p16, compared with young (2-month-old mouse tissues) mice (new Supplementary Fig. 12). In addition, MMP-7 levels did not increase in healthy, non-tumour-bearing mice treated with cisplatin, despite increased p21 in lungs (new Supplementary Fig.13).

Together, these data indicate that MMP-7 is not a general marker of age- or chemotherapy-related senescence *in normal tissues, but is preferentially upregulated in chemotherapy-treated lung tumors. Please also see our response to Reviewer 2 (Point 3) for related discussion.*

b.Perform RNA analysis on publicly available data from drug-treated tumors to analyze the correlation between MMP-7 and senescence in different contexts.

We thank the reviewer for this insightful suggestion. To further strengthen our clinical and contextual validation, we re-analysed a publicly available single-cell RNAseq dataset from lung adenocarcinoma (LUAD) patients, comparing treatment-naïve tumours with tumours exposed to neoadjuvant platinum-based chemotherapy (new Fig 5e-i). Importantly, expression of MMP7 and CDKN2A was restricted to malignant epithelial cells within LUAD tumours (new Fig. 5e).

In agreement with our previous results, chemotherapy-treated tumours exhibited increased expression of CDKN1A, CDKN2A and MMP7 (new Fig. 5h). Differential expression analysis further revealed significant enrichment of senescence-associated pathways, as well extracellular matrix (ECM) remodelling, cytokine-to-receptor interactions and drug metabolism signatures in malignant epithelial cells following chemotherapy (new Fig. 5i). Together, these analyses support the conclusion that platinum-based chemotherapy induces tumour senescence accompanied by elevated MMP-7 expression in a clinically relevant context.

c.Discuss the limitations and future directions needed to validate MMP-7 as a robust biomarker for chemotherapy-induced senescence.

We thank the reviewer for this useful suggestion. We have expanded the Discussion to explicitly address both the limitations and future directions required to validate MMP-7 as a robust of chemotherapy-induced senescence. Specifically, we now acknowledge that MMP-7 is implicated in other pathological contexts, including cancer progression, infection and inflammation, and therefore cannot be considered a universal or standalone senescence marker. We further outline key future studies needed to strengthen its validation, including comparative analyses across diverse senescent and non-senescent conditions, longitudinal studies to define temporal relationship between MMP-7 levels and senescence dynamics, and mechanistic investigations to identify the cellular sources of MMP-7 under chemotherapeutic stress.

Finally, we discuss the need for cross-tissue transcriptomic and proteomic profiling to benchmark MMP-7 against other SASP-associated proteases and to define when multiplexed detection panels will be required to achieve a robust, context-aware senescence signature (lines 696-722)

2.The authors suggest that MMP-7, as part of the senescence-associated secretory phenotype (SASP) profile, offers improved specificity over SA- β -gal for identifying chemotherapy-induced senescence. How would this platform distinguish between age-associated senescence from chemotherapy-induced senescence?

We thank the reviewer for this important question. As addressed in point (1), our new experiments directly distinguish age-associated from chemotherapy-induced senescence. Specifically, MMP-7 levels were not significantly elevated in naturally aged mouse tissues

despite increased p16 expression (new Supplementary Fig. 12), nor were they induced in healthy, non-tumour-bearing mice following cisplatin treatment (new Supplementary Fig. 13). In contrast, robust MMP-7 upregulation was consistently observed in chemotherapy-treated tumours. Together, these data indicate that MMP-7 preferentially reports chemotherapy-induced senescence in tumor contexts rather than age-associated senescence.

3. The authors developed an alloy-forming assay that lowered the limit of detection for AuNCs compared to conventional peroxidase assay. However, it would be more convincing to show that this increased sensitivity translates into improved performance in detecting senescence or fibrosis using in vivo samples. For example, can the alloy-forming assay detect senescence or fibrosis at earlier time points or in cases where the peroxidase assay fails to detect a signal?

We thank the reviewer for this helpful suggestion. To directly address this point, we performed additional in vivo experiments in a bleomycin-induced fibrosis model at an early time point (day 7), corresponding to incipient fibrotic remodelling. At this stage, fibrotic changes were validated histologically in bleomycin-treated mice (new Fig. 8i). Importantly, the alloy-forming assay detected elevated urinary signals that correlated with p21 and α -SMA expression (new Fig. 7j-k), while the conventional peroxidase-based assay failed to distinguish treated from control animals at this time point (new Supplementary Fig. 47).

These results demonstrate that the enhanced analytical sensitivity of the alloy assay confers a clear functional advantage *in vivo*, enabling earlier detection of senescence-associated fibrosis that is not achievable with peroxidase-based readouts.

Minor points

1. The figures in the supplementary material are not referenced in order. Reordering them to follow the order of appearance in the main text would improve readability.

We thank the reviewer for this helpful suggestion. We have carefully checked the order of the supplementary figures, and they are now referenced in order of appearance in the main text.

2. There are several inconsistencies and mismatches between the main text and the supplementary material that should be carefully reviewed. For example: a. Line 153 in the main text references Supplementary Fig. 3c-e, but Fig. appears unrelated to the statement.

We thank the reviewer for spotting a typographical error in the original line “ELISA confirmed the increased concentration of MMP-7 in the conditioned media from senescent cells, which was accompanied by elevated intracellular protein levels of MMP-7 and MMP-7 encoding mRNA (Supplementary Fig. 3c-e).” This now has been revised to lines 151-153 “ELISA confirmed the increased concentration of MMP-7 in the conditioned media from senescent cells, which was accompanied by elevated intracellular protein levels of MMP-7 and MMP-7 encoding mRNA (Supplementary Fig. 3d-e).” Supplementary Fig. 3d shows the intracellular protein levels of MMP-7, and Supplementary Fig. 3e shows the level of MMP-7 mRNA.

b. The figures and captions in Supplementary Fig. 17c-e are out of order.

We thank the reviewer for spotting this. The original order in the caption of the previous Supplementary Fig. 17 “c. LC-MS spectrum of nanoprobe, showing peaks corresponding to unbound DBCO-albumin and the nanoprobe peak in which albumin is conjugated to one AuNC. d. Histogram plotting the hydrodynamic size of AuNC and nanoprobe in PBS.

Experiment was repeated independently 3 times with similar results. **e.** Surface charge of AuNC and nanoprobe. Experiment was repeated independently 3 times with similar results.” has now been changed to (now Supplementary Fig. 20) “**c.** Histogram plotting the hydrodynamic size of AuNC and nanoprobe in PBS. Experiment was repeated independently 3 times with similar results. **d.** Surface charge of AuNC and nanoprobe. Experiment was repeated independently 3 times with similar results. **e.** LC-MS spectrum of nanoprobe, showing peaks corresponding to unbound DBCO-albumin and the nanoprobe peak in which albumin is conjugated to one AuNC.”

c. Line 222 states “Nanoprobe assembly was confirmed by gel electrophoresis (Fig. 3d) and dynamic light scattering (DLS) (Supplementary Fig. 17c), which indicated a size increase after conjugation.” However, Supplementary Fig. 17c shows only a single peak without a shift in size.

We thank the reviewer for this spot-on comment. We have now changed Supplementary Fig. 20c (which was previously Supplementary Fig. 17c) to include the DLS histogram for both the nanoprobe and AuNC. This figure now clearly reflects an increase in hydrodynamic size from AuNC to the nanoprobe.

d. Line 313 references Supplementary Fig. 32d-f, which does not exist.

We thank the reviewer for noticing this typographical error in the previous line 313 “TEM analysis confirmed the presence of AuNCs cleared from the mice, with EDX spectra identifying the particles consisting of elemental gold (Supplementary Fig. 32d-f).” We have changed this to Supplementary Fig. 37d-f, which presents the data mentioned in the text.

3. In line 308, the authors state that the colorimetric assays “quantitatively” detect renally cleared AuNCs, which is imprecise given the reported correlation coefficients of 0.67 and 0.65. This claim should be revised to more accurately reflect the data.

We thank the reviewer for this suggestion. We agree with this, therefore we have changed this line from “We also showed that the developed colorimetric assays could quantitatively detect renally cleared AuNCs in urine of healthy C57BL/6 mice. Both assays showed a positive correlation with the Au content measured by ICP-MS, with Pearson correlation coefficients of 0.67 and 0.65, respectively” to (now line 377-380) into “We also showed that the developed colorimetric assays could **semi**-quantitatively detect renally cleared AuNCs in urine of healthy C57BL/6 mice. Both assays showed a positive correlation with the Au content measured by ICP-MS, with Pearson correlation coefficients of 0.67 and 0.65, respectively (Supplementary Fig. 37a-c).”

Response to reviewers

Reviewer #1:

The authors have properly addressed all issues raised. Therefore I recommend publication.
We thank the reviewer for their time and insightful comments through the revision process.

Reviewer #2:

This study has two major limitations. First, there is insufficient evidence demonstrating the specificity of MMP-7 for cancer-associated senescence (also expression in non-treated cancer, infection, and inflammation). Second, the artificial urinary probe design lacks substantial novelty beyond the alloy-forming strategy and the chemical linkage. These concerns were also raised by Reviewer 3.

To address the first limitation, the authors additionally tested their ALBANC probes in gene knockout cell lines, other cancer cell lines beyond lung cancer, aging mice, and patient specimens. The results appear to support the authors' claim that MMP-7 is highly expressed in chemotherapy-treated cancer cells and in fibrotic lung cancers. However, in the Discussion section, the authors also acknowledge that MMP-7 is not an optimal biomarker with high specificity for senescence as MMP-7 cannot reliably distinguish cancer-associated senescence from inflammation or infection, and additional biomarkers would be required to improve diagnostic accuracy.

To address the second limitation, the authors provided additional experiments demonstrating that the alloy-forming method achieves a lower limit of detection than traditional peroxidase-based assays, which is a meaningful improvement. They also claim that chemically linked ALBANC probes offer enhanced biosafety, reproducibility, and translational robustness compared with neutravidin-based conjugates formed through physical interactions (page 3). However, such advantages are not supported by direct experimental evidence.

Regarding the first limitation, it is important to note that there is no universal or specific biomarker of senescence. Even widely used markers have clear limitations. For instance, SA- β -gal activity is not exclusive to senescent cells, as many non-senescent cells show basal lysosomal β -galactosidase activity, and some cancer or differentiated cells (e.g., osteoclasts) can display high activity independent of senescence. Likewise, p16 and/or p21 expression and IL-6 secretion have been reported in quiescent or arrested (non-senescent) cells. Therefore, these markers are not specific when used alone.

In our study, we found that MMP-7 strongly correlates with therapy-induced senescence (TIS) in lung cancer and with senescence associated with pulmonary fibrosis, but not with ageing-associated p16⁺ senescence in lung, heart, or kidney tissues. We cannot exclude that MMP-7 may be expressed in other physiological or pathological contexts in specific tissues, and we acknowledge this limitation in the Discussion.

Importantly, data from the Human Protein Atlas (<https://www.proteinatlas.org/ENSG00000137673-MMP7/tissue>) indicate that MMP-7 protein is undetectable in 37 of 44 tissues, detected at low levels in 5 tissues, and at medium levels only in gallbladder and kidney (Figure below). Thus, MMP-7 shows limited basal expression across tissues. Based on these data and our findings, we conclude that MMP-7 is a valuable, but not unique, context-dependent biomarker of TIS in lung cancer and pulmonary fibrosis, and should be assessed in combination with other senescence markers, as done in our study.

Relative expression of the four different protease candidates across different tissues and organs. The data was extracted from publicly available Human Protein Atlas database. MMP-7 shows the lowest overall expression across 44 tissues. Samples were obtained from male and female (age 63-73).

To address the reviewer's comments regarding the second limitation, we revised the manuscript to remove unsupported claims about biosafety, reproducibility, and robustness:

- Removed statements claiming enhanced biostability, reproducibility, and translational robustness relative to neutravidin-based conjugates (introduction Ln 96-97)
- Replaced strong claims about "overcoming limitations" and "precision surveillance" with more neutral language describing a sensitive urinary readout that complements histological assays and supports longitudinal monitoring in our disease models. (Ln 98-100)
- Rephrased the reproducibility section (Ln 477-478) to describe the standardized endpoint assay, batch calibration with known AuNC concentrations, and reported nanoparticle size (30.9 ± 7.8 nm), without extending conclusions beyond the presented data.
- Replaced "no in vivo toxicity was observed, confirming high biocompatibility" with "No detectable levels of toxicity were observed over the study period, consistent with tolerability under these conditions." (Ln 495-496)

Finally, we thank the reviewer for their constructive comments and suggestions throughout the revision process, which have helped strengthen and clarify the manuscript.

Reviewer #3:

The authors were responsive to my comments and overall I am satisfied with the revised manuscript. I do want to comment on novelty as this was surfaced by the other reviewers as well. In my opinion, novelty can be divided into biological, technological, and translational advances. Here the authors have supportive data to target MMP-7 as a potentially new biological target of drug-induced senescence. They also transferred the in vivo sensing scaffold to albumin, which increases the translatability compared to the use of neutravidin in previous papers (although there are still likely too many moving parts for translation). While the general concept of urinary sensing via the application of protease sensors has been previously described, the authors particular focus on drug-induced senescence is novel and would expose this type of approach to a broader audience than the previous studies. For these reasons, I am supportive of publication.

We thank the reviewer for their time, careful evaluation of the manuscript, and constructive comments.